# Flowing Datasets with Wasserstein over Wasserstein Gradient Flows

Clément Bonet [* 1]    Christophe Vauthier [* 2]    Anna Korba [1]

## Abstract

Many applications in machine learning involve data represented as probability distributions. The emergence of such data requires radically novel techniques to design tractable gradient flows on probability distributions over this type of (infinite-dimensional) objects. For instance, being able to flow labeled datasets is a core task for applications ranging from domain adaptation to transfer learning or dataset distillation. In this setting, we propose to represent each class by the associated conditional distribution of features, and to model the dataset as a mixture distribution supported on these classes (which are themselves probability distributions), meaning that labeled datasets can be seen as probability distributions over probability distributions. We endow this space with a metric structure from optimal transport, namely the Wasserstein over Wasserstein (WoW) distance, derive a differential structure on this space, and define WoW gradient flows. The latter enables to design dynamics over this space that decrease a given objective functional. We apply our framework to transfer learning and dataset distillation tasks, leveraging our gradient flow construction as well as novel tractable functionals that take the form of Maximum Mean Discrepancies with Sliced-Wasserstein based kernels between probability distributions.

## 1. Introduction

Probability measures provide a powerful way to represent many data types. For instance, they allow to naturally represent documents (Kusner et al., 2015), genes (Bellazzi et al., 2021), point clouds (Qi et al., 2017; Geuter et al., 2025), images (Sodini et al., 2025), or single-cell data (Persad et al.,

2023; Haviv et al., 2024b). Remarkably, it has been shown that one can embed any finite dataset with little or no distortion (Andoni et al., 2018; Kratsios et al., 2023) in the Wasserstein space, *i.e.*, the space of probability distributions (*e.g.*, over a Euclidean space) equipped with the Wasserstein-2 distance from Optimal Transport (OT). This has motivated the use of this space to embed many types of data ranging from words (Vilnis & McCallum, 2015) to knowledge graphs (He et al., 2015; Wang et al., 2022), graphs (Bojchevski & Günnemann, 2018; Petric Maretic et al., 2019), or neuroscience data (Bonet et al., 2023). Therefore, it is essential to develop tools to work on the space of probability measures over probability measures, also known as random measures. In particular, they provide a natural way to represent labeled datasets as mixtures (Alvarez-Melis & Fusi, 2020).

A natural distance on this space is the Wasserstein over Wasserstein distance (WoW) (Nguyen, 2016; Catalano & Lavenant, 2024), also known as the Hierarchical OT distance, which lifts the Wasserstein distance between probability distributions as a ground cost, to define a Wasserstein distance between random measures. The latter has been used for generative modeling applications (Dukler et al., 2019), domain adaptation tasks (El Hamri et al., 2022), comparing documents (Yurochkin et al., 2019) or multilevel clustering (Ho et al., 2017). It has also been used to compare Gaussian mixtures (Chen et al., 2018; Delon & Desolneux, 2020; Wilson et al., 2024) or generic mixtures (Dusson et al., 2023; Chen & Zhang, 2024). However, its poor sample complexity has motivated the development of alternative distance measures, such as those based on Integral Probability Metrics (Catalano & Lavenant, 2024). Nonetheless, this space possesses a rich Riemannian structure, enabling the definition of concepts like geodesics. This has been leveraged recently by Haviv et al. (2024a) to perform generative modeling over the space of probability distributions with Flow Matchings.

While this space naturally supports a range of machine learning tasks, optimization methods tailored to it have received limited attention. Yet, this is important for multiple applications, including variational inference with a Gaussian mixture family (Lambert et al., 2022; Huix et al., 2024), computing barycenters (Delon & Desolneux, 2020), or flowing datasets (Alvarez-Melis & Fusi, 2021), *e.g.*, for domain adaptation, transfer learning (Alvarez-Melis & Fusi, 2021; Hua et al., 2023) or dataset distillation (Wang et al., 2018).

---

[*]Equal contribution  [1]ENSAE, CREST, IP Paris  [2]Université Paris-Saclay, Laboratoire de Mathématique d'Orsay. Correspondence to: Clément Bonet <clement.bonet@ensae.fr>, Christophe Vauthier <christophe.vauthier@universite-paris-saclay.fr>.

*Proceedings of the 42nd International Conference on Machine Learning*, Vancouver, Canada. PMLR 267, 2025. Copyright 2025 by the author(s).

In this paper, we propose to leverage the Riemannian structure of random measures equipped with the WoW distance, by defining and simulating gradient flows, *i.e.*, paths of random measures that follow the steepest descent of a given objective functional.

**Related works.** An elegant and popular way to perform optimization over probability distributions (over a manifold) is to leverage the Riemannian structure of the Wasserstein space (Otto, 2001), and to use Wasserstein gradient flows (Ambrosio et al., 2008; Santambrogio, 2017). Several time discretizations of these flows have been studied (Jordan et al., 1998; Salim et al., 2020; Bonet et al., 2024), and they have been applied to simulate the flow dynamics of multiple objectives such as the Kullback-Leibler divergence (Wibisono, 2018; Salim et al., 2020; Diao et al., 2023), the Maximum Mean Discrepancy (MMD) (Arbel et al., 2019; Altekrüger et al., 2023; Hertrich et al., 2024a;b) and variants thereof (Glaser et al., 2021; Chen et al., 2024; Neumayer et al., 2024; Chazal et al., 2024) or the Sliced-Wasserstein distance (Liutkus et al., 2019; Du et al., 2023; Bonet et al., 2025). Yet, all these works focus on the case where the probability distributions are defined over a finite-dimensional manifold, *e.g.* $\mathbb{R}^d$. In practice, simulating these flows often boils down to simulating a particle system in $\mathbb{R}^d$. Hence, these works do not address probability distributions defined on infinite-dimensional spaces, such as the space of probability measures, which is the focus of this work.

The closest works to ours are the ones of Alvarez-Melis & Fusi (2021) and Hua et al. (2023). These papers cast labeled datasets as measures over a product space of the features and the conditional distributions (*i.e.*, the distributions of the features of a given class). However, they circumvent the issue of designing gradient flows on this space by modeling the conditional probabilities as Gaussian distributions, hence parametrized by a mean and covariance, which are finite-dimensional objects. While this enables them to leverage standard Wasserstein gradient flows, this Gaussian modeling of mixture components is a strong assumption that may not capture the true shape of many labeled datasets in practice.

**Contributions.** In this work, we introduce a principled framework for optimizing functionals over the space of probability measures on probability measures, leveraging the Riemannian structure of this space to develop Wasserstein over Wasserstein (WoW) gradient flows. We provide a theoretical construction of the flows, and then a practical implementation through time discretization using a forward Euler scheme. We also propose a novel functional objective, that writes as an MMD with kernel between distributions based on the Sliced-Wasserstein distance, and whose gradient flow simulation is tractable. We then apply this scheme to flow datasets viewed as random measures; specifically, as

mixtures of probability distributions corresponding to the class-conditional distributions. We focus on image datasets, and show that the flow enables structured transitions of classes toward other classes, with applications to transfer learning and dataset distillation.

**Notations.** For a Riemannian manifold $\mathcal{M}$, $d : \mathcal{M} \times \mathcal{M} \to \mathbb{R}_+$ is its geodesic distance. For $x \in \mathcal{M}$, we denote by $T_x\mathcal{M}$ the tangent space at $x$, and by $\|\cdot\|_x$ the Riemannian metric. We define by $T\mathcal{M} = \{(x, v),\ x \in \mathcal{M}$ and $v \in T_x\mathcal{M}\}$ the tangent bundle. We define for $(x, v) \in T\mathcal{M}$ the projections $\pi^{\mathcal{M}}(x, v) = x$ and $\pi^{\mathrm{v}}(x, v) = v$. $\exp : T\mathcal{M} \to \mathcal{M}$ is the exponential map. For $x \in \mathcal{M}$, if $\exp_x : T_x\mathcal{M} \to \mathcal{M}$ is invertible, we note $\log_x$ its inverse. $\nabla$ and $\mathrm{div}$ refer to the Riemannian gradient and divergence on $\mathcal{M}$. For a metric space $(X, d)$, $\mathcal{P}_2(X)$ denotes the space of probability distributions on $X$ with second finite moments, *i.e.*, $\mathcal{P}_2(X) = \{\mu \in \mathcal{P}(X),\ \int d(x, o)^2 \mathrm{d}\mu(x) < \infty\}$ with $o \in X$ some arbitrary origin. For any $\mu \in \mathcal{P}_2(\mathcal{M})$, $L^2(\mu, T\mathcal{M})$ is the set of functions $v : \mathcal{M} \to T\mathcal{M}$ such that $\int \|v(x)\|_x^2 \mathrm{d}\mu(x) < \infty$. For a measurable map $T : \mathcal{M} \to \mathcal{M}$, we note by $T_{\#}\mu$ the pushforward measure. Id denotes the identity map on $\mathcal{M}$. $\mathcal{P}_{2,\mathrm{ac}}(\mathcal{M}) \subset \mathcal{P}_2(\mathcal{M})$ is the space of measures absolutely continuous *w.r.t.* the volume measure on $\mathcal{M}$. For $\mu, \nu \in \mathcal{P}(X)$, we denote $\mu \ll \nu$ if $\mu$ is absolutely continuous *w.r.t.* $\nu$. $\Pi(\mu, \nu) = \{\gamma \in \mathcal{P}(X \times X),\ \pi^1_{\#}\gamma = \mu,\ \pi^2_{\#}\gamma = \nu\}$ with $\pi^i : (x_1, x_2) \mapsto x_i$, is the set of couplings, and $\Pi_o(\mu, \nu)$ the set of optimal couplings.

## 2. Background

We begin by introducing some background on Optimal Transport (OT) and on Wasserstein Gradient Flows. For theoretical purposes, we provide background on the geometry of $(\mathcal{P}_2(\mathcal{M}), \mathrm{W}_2)$ with $\mathcal{M}$ a Riemannian manifold, as in the next section, we will rely on results which hold on compact Riemannian manifolds (without boundary). Nonetheless, the applications will be done for $\mathcal{M} = \mathbb{R}^d$. The reader may refer to Appendix A for more details.

**Optimal Transport.** The Wasserstein distance between $\mu, \nu \in \mathcal{P}_2(\mathcal{M})$ is defined as

$$\mathrm{W}_2^2(\mu, \nu) = \inf_{\tilde{\gamma} \in \Pi(\mu, \nu)} \int d(x, y)^2\ \mathrm{d}\tilde{\gamma}(x, y). \tag{1}$$

The metric space $(\mathcal{P}_2(\mathcal{M}), \mathrm{W}_2)$ has a Riemannian structure (Otto, 2001; Erbar, 2010). In particular, if the log map is well defined $\mu$-almost everywhere (a.e.), (constant-speed) geodesics between $\mu, \nu$ are defined as $\mu_t = \left(\exp_{\pi^1} \circ (t \log_{\pi^1} \circ \pi^2)\right)_{\#} \tilde{\gamma}$ with $\tilde{\gamma} \in \Pi_o(\mu, \nu)$ an optimal coupling. If $\mu \in \mathcal{P}_{2,\mathrm{ac}}(\mathcal{M})$, there is a map T, namely the OT map, such that $T_{\#}\mu = \nu$ and $\tilde{\gamma} = (\mathrm{Id}, T)_{\#}\mu$ by McCann's theorem for a wide range of manifolds (McCann, 2001; Figalli, 2007). In particular, $T = \exp_{\mathrm{Id}} \circ (-\nabla \varphi_{\mu, \nu})$

with $\varphi_{\mu,\nu}$ a Kantorovich potential between $\mu$ and $\nu$, and geodesics become $\mu_t = \exp_{\mathrm{Id}} \circ (-t\nabla\varphi_{\mu,\nu})_{\#}\mu$. At any $\mu \in \mathcal{P}_{2,\mathrm{ac}}(\mathcal{M})$, we can define a tangent space $T_{\mu}\mathcal{P}_2(\mathcal{M}) = \overline{\{\nabla\varphi, \; \varphi \in C_c^{\infty}(\mathcal{M})\}}^{L^2(\mu,T\mathcal{M})}$ with $C_c^{\infty}(\mathcal{M})$ the space of smooth compactly supported functions on $\mathcal{M}$ (Erbar, 2010; Gigli, 2011). This is a Hilbert space endowed with the $L^2(\mu, T\mathcal{M})$ inner product. The exponential map on $\mathcal{P}_2(\mathcal{M})$ is then defined as $\exp_{\mu}(v) = (\exp_{\mathrm{Id}} \circ v)_{\#}\mu$ for $\mu \in \mathcal{P}_2(\mathcal{M})$, $v \in T_{\mu}\mathcal{P}_2(\mathcal{M})$. For instance, when $\mathcal{M} = \mathbb{R}^d$ with $d(x,y)^2 = \|x-y\|_2^2$, then $\exp_x(y) = x + y$ and $\log_x(y) = y - x$ for all $x, y \in \mathbb{R}^d$.

Let $\mathcal{P}_2(T\mathcal{M}) := \{\gamma \in \mathcal{P}(T\mathcal{M}), \; \int (d(x,o)^2 + \|v\|_x^2)\mathrm{d}\gamma(x,v) < \infty\}$ where $o \in \mathcal{M}$ is any reference point. Following (Gigli, 2011), we define for every $\mu, \nu \in \mathcal{P}_2(\mathcal{M})$,

$$\exp_{\mu}^{-1}(\nu) := \Big\{\gamma \in \mathcal{P}_2(T\mathcal{M}), \; \pi_{\#}^{\mathcal{M}}\gamma = \mu, \; \exp_{\#}\gamma = \nu, \int \|v\|_x^2 \mathrm{d}\gamma(x,v) = \mathrm{W}_2^2(\mu,\nu)\Big\} \quad (2)$$

the set of plans $\gamma \in \mathcal{P}_2(T\mathcal{M})$ such that $(\pi^{\mathcal{M}}, \exp)_{\#}\gamma$ is an OT plan between $\mu$ and $\nu$. This allows one to avoid using the logarithm map, which might not be well defined everywhere, *e.g.* being multivalued. This space carries more information than the set of optimal couplings as it precises which geodesic was chosen to move the mass, as $\mu_t = (\exp_{\pi^{\mathcal{M}}} \circ (t\pi^v))_{\#}\gamma$ are constant speed geodesics between $\mu$ and $\nu$ (Gigli, 2011, Theorem 1.11). On $\mathcal{P}_2(\mathbb{R}^d)$, this translates as $\exp_{\mu}^{-1}(\nu) = \{(\pi^1, \pi^2 - \pi^1)_{\#}\tilde{\gamma}, \; \tilde{\gamma} \in \Pi_o(\mu,\nu)\}$ (Gigli, 2004; Hertrich et al., 2024a). We show in the next proposition, whose proof can be found in Appendix C.1, that we can build a surjective map from $\exp_{\mu}^{-1}(\nu)$ to $\Pi_o(\mu,\nu)$.

**Proposition 2.1.** *Let $\mu, \nu \in \mathcal{P}_2(\mathcal{M})$. A surjective map from $\exp_{\mu}^{-1}(\nu)$ to $\Pi_o(\mu,\nu)$ is given by $\gamma \mapsto (\pi^{\mathcal{M}}, \exp)_{\#}\gamma$. In particular, $\exp_{\mu}^{-1}(\nu)$ is not empty, and if $\gamma \in \exp_{\mu}^{-1}(\nu)$, then $d(x, \exp_x(v)) = \|v\|_x$ for $\gamma$-a.e. $(x,v) \in T\mathcal{M}$.*

*Additionally, if $\mathcal{M}$ is compact and connected, and $\mu \in \mathcal{P}_{2,\mathrm{ac}}(\mathcal{M})$, then there exists a unique $\gamma \in \exp_{\mu}^{-1}(\nu)$, of the form $\gamma = (\mathrm{Id}, -\nabla\varphi_{\mu,\nu})_{\#}\mu$.*

**Wasserstein Gradient Flows.** Let $\mathcal{F} : \mathcal{P}_2(\mathcal{M}) \to \mathbb{R}$ be a lower semi-continuous functional. We briefly introduce the differential structure on $(\mathcal{P}_2(\mathcal{M}), \mathrm{W}_2)$, *i.e.*, probability measures on manifolds, inspired by (Erbar, 2010) and (Lanzetti et al., 2025).

Let $\mu \in \mathcal{P}_2(\mathcal{M})$. We say that $\nabla_{\mathrm{W}_2}\mathcal{F}(\mu) \in L^2(\mu, T\mathcal{M})$ is a Wasserstein gradient of $\mathcal{F}$ at $\mu$ if for any $\nu \in \mathcal{P}_2(\mathcal{M})$ and any $\gamma \in \exp_{\mu}^{-1}(\nu)$, we have the Taylor expansion

$$\mathcal{F}(\nu) = \mathcal{F}(\mu) + \int \langle \nabla_{\mathrm{W}_2}\mathcal{F}(\mu)(x), v \rangle_x \, \mathrm{d}\gamma(x,v)$$
$$+ o(\mathrm{W}_2(\mu,\nu)). \quad (3)$$

If such a gradient exists, then we say that $\mathcal{F}$ is Wasserstein differentiable at $\mu$. There is a unique gradient belonging to $T_{\mu}\mathcal{P}_2(\mathcal{M})$ and we restrict to this gradient. Informally, the Wasserstein gradient of $\mathcal{F}$ can be computed as $\nabla_{\mathrm{W}_2}\mathcal{F}(\mu) = \nabla\frac{\delta\mathcal{F}}{\delta\mu}(\mu)$, with $\frac{\delta\mathcal{F}}{\delta\mu}(\mu)$ the first variation defined, when it exists, as the unique function (up to an additive constant) such that, for $\chi$ satisfying $\int \mathrm{d}\chi = 0$, $\frac{\mathrm{d}}{\mathrm{d}t}\mathcal{F}(\mu + t\chi)\big|_{t=0} = \int \frac{\delta\mathcal{F}}{\delta\mu}(\mu) \, \mathrm{d}\chi$ (Ambrosio et al., 2008, Lemma 10.4.1). Examples of differentiable functionals include potential energies $\mathcal{V}(\mu) = \int V\mathrm{d}\mu$ and interaction energies $\mathcal{W}(\mu) = \iint W(x,y)\mathrm{d}\mu(x)\mathrm{d}\mu(y)$ for $V : \mathcal{M} \to \mathbb{R}$ and $W : \mathcal{M} \times \mathcal{M} \to \mathbb{R}$ twice differentiable with bounded Hessian, for which $\nabla_{\mathrm{W}_2}\mathcal{V}(\mu) = \nabla V$ and $\nabla_{\mathrm{W}_2}\mathcal{W}(\mu)(x) = \int (\nabla_1 W(x,y) + \nabla_2 W(y,x))\mathrm{d}\mu(y)$. Moreover, if the functional $\mathcal{F} : \mathcal{P}_2(\mathbb{R}^d) \to \mathbb{R}$ has a closed-form over discrete measures, *i.e.*, there exists $F : (\mathbb{R}^d)^n \to \mathbb{R}$ such that $\mathcal{F}(\frac{1}{n}\sum_{i=1}^n \delta_{x_i}) = F(x_1, \ldots, x_n)$, then we can use backpropagation on $F$ and find the Wasserstein gradient of $\mathcal{F}$ using the relation $\nabla_{\mathrm{W}_2}\mathcal{F}(\frac{1}{n}\sum_{i=1}^n \delta_{x_i})(x_i) = n\nabla_i F(x_1, \ldots, x_n)$ (see Proposition A.9).

A Wasserstein gradient flow of a differentiable functional $\mathcal{F}$ is a curve $t \mapsto \mu_t$ which is a (weak) solution of the continuity equation $\partial_t \mu_t = \mathrm{div}(\mu_t \nabla_{\mathrm{W}_2}\mathcal{F}(\mu_t))$. A possible discretization is the Riemannian Wasserstein gradient descent (Bonnabel, 2013; Bonet et al., 2025), defined as $\mu_{k+1} = \exp_{\mathrm{Id}}(-\tau\nabla_{\mathrm{W}_2}\mathcal{F}(\mu_k))_{\#}\mu_k$. For discrete distributions $\mu_k = \frac{1}{n}\sum_{i=1}^n \delta_{x_i^k}$, it translates as, for all $k \geq 0$, $i \in \{1, \ldots, n\}$, $x_i^{k+1} = \exp_{x_i^k}(-\tau\nabla_{\mathrm{W}_2}\mathcal{F}(\mu_k)(x_i^k))$. For $\mathcal{M} = \mathbb{R}^d$, this is simply $x_i^{k+1} = x_i^k - \tau\nabla_{\mathrm{W}_2}\mathcal{F}(\mu_k)(x_i^k)$, which corresponds to Wasserstein gradient descent.

## 3. Wasserstein over Wasserstein Space

We introduce in this section the Wasserstein over Wasserstein space $(\mathcal{P}_2(\mathcal{P}_2(\mathcal{M})), \mathrm{W}_{\mathrm{W}_2})$, *i.e.*, the space of probability distributions over probability distributions $\mathcal{P}_2(\mathcal{P}_2(\mathcal{M}))$, endowed with the OT distance with the squared Wasserstein distance on $\mathcal{P}_2(\mathcal{M})$ as groundcost. We first state some properties of this distance, and then introduce a differential structure on this space which will be used in the next sections to develop suitable optimization methods. In the following, $\mathcal{M}$ is a compact and connected manifold. The proofs can be found in Appendix C.

### 3.1. OT Distance and Riemannian Structure

The WoW distance is defined as the OT problem with the squared Wasserstein distance on $\mathcal{P}_2(\mathcal{M})$ as groundcost, *i.e.*, for $\mathbb{P}, \mathbb{Q} \in \mathcal{P}_2(\mathcal{P}_2(\mathcal{M}))$,

$$\mathrm{W}_{\mathrm{W}_2}(\mathbb{P}, \mathbb{Q})^2 = \inf_{\Gamma \in \Pi(\mathbb{P},\mathbb{Q})} \int \mathrm{W}_2^2(\mu,\nu) \, \mathrm{d}\Gamma(\mu,\nu). \quad (4)$$

This defines a distance (Nguyen, 2016). Analogously to $\mathcal{P}_2(\mathcal{M})$ and Brenier-McCann's theorem, it has been shown

that there exists an OT map from $\mathbb{P}$ to $\mathbb{Q}$ under absolute continuity of $\mathbb{P}$ with respect to a suitable reference measure $\mathbb{P}_0 \in \mathcal{P}_2(\mathcal{P}_2(\mathcal{M}))$ (Emami & Pass, 2025), which has no atom and satisfies an integration by part formula (Dello Schiavo, 2020). We refer to Appendix B for more details.

Now, let us denote for any $\gamma \in \mathcal{P}_2(T\mathcal{M})$, the projections $\phi^{\mathcal{M}}(\gamma) = \pi^{\mathcal{M}}_{\#}\gamma$, $\phi^{\exp}(\gamma) = \exp_{\#}\gamma$ and $\phi^{\mathrm{v}}(\gamma) = \pi^{\mathrm{v}}_{\#}\gamma$. For any $\mathbb{P}, \mathbb{Q} \in \mathcal{P}_2(\mathcal{P}_2(\mathcal{M}))$, let us also define

$$\exp^{-1}_{\mathbb{P}}(\mathbb{Q}) := \{\Gamma \in \mathcal{P}_2(\mathcal{P}_2(T\mathcal{M})), \phi^{\mathcal{M}}_{\#}\Gamma = \mathbb{P}, \; \phi^{\exp}_{\#}\Gamma = \mathbb{Q},$$
$$\iint \|v\|^2_x \mathrm{d}\gamma(x,v)\mathrm{d}\Gamma(\gamma) = \mathrm{W}^2_{\mathrm{W}_2}(\mathbb{P},\mathbb{Q})\}. \quad (5)$$

Relying on Proposition 2.1, we can define for any $\mathbb{P}, \mathbb{Q} \in \mathcal{P}_2(\mathcal{P}_2(\mathcal{M}))$ a surjective map from $\exp^{-1}_{\mathbb{P}}(\mathbb{Q})$ to $\Pi_o(\mathbb{P}, \mathbb{Q})$.

**Proposition 3.1.** *Let* $\mathbb{P}, \mathbb{Q} \in \mathcal{P}_2(\mathcal{P}_2(\mathcal{M}))$. *Then,* $\Gamma \mapsto (\phi^{\mathcal{M}}, \phi^{\exp})_{\#}\Gamma$ *is a surjective map from* $\exp^{-1}_{\mathbb{P}}(\mathbb{Q})$ *to* $\Pi_o(\mathbb{P}, \mathbb{Q})$. *In particular,* $\exp^{-1}_{\mathbb{P}}(\mathbb{Q})$ *is not empty and if* $\Gamma \in \exp^{-1}_{\mathbb{P}}(\mathbb{Q})$, $\gamma \in \exp^{-1}_{\pi^{\mathcal{M}}_{\#}\gamma}(\exp_{\#}\gamma)$ *for* $\Gamma$*-a.e.* $\gamma$.

*Additionally, if* $\mathbb{P} \ll \mathbb{P}_0$, *there exists a unique* $\Gamma \in \exp^{-1}_{\mathbb{P}}(\mathbb{Q})$, *of the form* $(\mu \mapsto (\mathrm{Id}, -\nabla\varphi_{\mu,\mathrm{T}(\mu)})_{\#}\mu)_{\#}\mathbb{P}$ *with* $\mathrm{T}$ *the unique transport map from* $\mathbb{P}$ *to* $\mathbb{Q}$ *and* $\varphi_{\mu,\mathrm{T}(\mu)}$ *a Kantorovich potential between* $\mu, \mathrm{T}(\mu) \in \mathcal{P}_2(\mathcal{M})$.

The proof of Proposition 3.1 can be found in Appendix C.2. The previous construction enables us to formalize the Riemannian structure of $(\mathcal{P}_2(\mathcal{P}_2(\mathcal{M})), \mathrm{W}_{\mathrm{W}_2})$, without having to define a notion of logarithm map on $\mathcal{P}_2(\mathcal{M})$, which might be ill-defined when the OT plan is not unique. Between $\mathbb{P}, \mathbb{Q} \in \mathcal{P}_2(\mathcal{P}_2(\mathcal{M}))$, we can define for $\Gamma \in \exp^{-1}_{\mathbb{P}}(\mathbb{Q})$ a geodesic $t \mapsto \mathbb{P}_t = \exp_{\phi^{\mathcal{M}}} \circ (t\phi^{\mathrm{v}})_{\#}\Gamma$, which satisfies for all $s, t \in [0,1]$, $\mathrm{W}_{\mathrm{W}_2}(\mathbb{P}_s, \mathbb{P}_t) = |t - s|\mathrm{W}_{\mathrm{W}_2}(\mathbb{P}, \mathbb{Q})$, see Appendix B. For $\mathbb{P} \ll \mathbb{P}_0$, using Proposition 3.1, the curve simplifies as $\mathbb{P}_t = \exp_{\mathrm{Id}} \circ (-t\nabla\varphi_{\mathrm{Id},\mathrm{T}})_{\#}\mathbb{P}$. Moreover, for $\mathcal{M} = \mathbb{R}^d$, this reads as $\mathbb{P}_t = (\mu \mapsto (\mathrm{Id} - t\nabla\varphi_{\mu,\mathrm{T}(\mu)})_{\#}\mu)_{\#}\mathbb{P}$.

### 3.2. Differential Structure

We now provide a differential structure to $(\mathcal{P}_2(\mathcal{P}_2(\mathcal{M})), \mathrm{W}_{\mathrm{W}_2})$, following the one of (Ambrosio et al., 2008; Erbar, 2010; Lanzetti et al., 2025) for $(\mathcal{P}_2(\mathcal{M}), \mathrm{W}_2)$. In this section, let $\mathbb{F} : \mathcal{P}_2(\mathcal{P}_2(\mathcal{M})) \to \mathbb{R}$ be a lower semi-continuous functional. We define formally the Hilbert space $L^2(\mathbb{P}, T\mathcal{P}_2(\mathcal{M}))$ of functions from $\mathcal{P}_2(\mathcal{M})$ to $T\mathcal{P}_2(\mathcal{M})$ in Appendix B.1. First, we define the notions of (extended) sub- and super-differential.

**Definition 3.2.** $\xi \in L^2(\mathbb{P}, T\mathcal{P}_2(\mathcal{M}))$ belongs to the subdifferential $\partial^-\mathbb{F}(\mathbb{P})$ of $\mathbb{F}$ at $\mathbb{P}$ if for all $\mathbb{Q} \in \mathcal{P}_2(\mathcal{P}_2(\mathcal{M}))$,

$$\mathbb{F}(\mathbb{Q}) \geq \mathbb{F}(\mathbb{P}) + \sup_{\Gamma} \iint \langle \xi(\pi^{\mathcal{M}}_{\#}\gamma)(x), v\rangle_x \, \mathrm{d}\gamma(x,v)\mathrm{d}\Gamma(\gamma)$$
$$+ o(\mathrm{W}_{\mathrm{W}_2}(\mathbb{P}, \mathbb{Q})), \quad (6)$$

where the $\Gamma$ in the $\sup$ are selected in $\exp^{-1}_{\mathbb{P}}(\mathbb{Q})$. Similarly, $\xi \in L^2(\mathbb{P}, T\mathcal{P}_2(\mathcal{M}))$ belongs to the super-differential $\partial^+\mathbb{F}(\mathbb{P})$ of $\mathbb{F}$ at $\mathbb{P}$ if $-\xi \in \partial^-(-\mathbb{F})(\mathbb{P})$.

If the functional admits a sub- and super-differential, which coincide, we can define a gradient.

**Definition 3.3.** $\mathbb{F}$ is Wasserstein differentiable at $\mathbb{P} \in \mathcal{P}_2(\mathcal{P}_2(\mathcal{M}))$ if $\partial^+\mathbb{F}(\mathbb{P}) \cap \partial^-\mathbb{F}(\mathbb{P}) \neq \emptyset$. In this case, we say that $\xi \in \partial^-\mathbb{F}(\mathbb{P}) \cap \partial^+\mathbb{F}(\mathbb{P})$ is a WoW gradient of $\mathbb{F}$ at $\mathbb{P}$, and it satisfies for any $\mathbb{Q} \in \mathcal{P}_2(\mathcal{P}_2(\mathcal{M}))$, $\Gamma \in \exp^{-1}_{\mathbb{P}}(\mathbb{Q})$,

$$\mathbb{F}(\mathbb{Q}) = \mathbb{F}(\mathbb{P}) + \iint \langle \xi(\pi^{\mathcal{M}}_{\#}\gamma)(x), v\rangle_x \, \mathrm{d}\gamma(x,v)\mathrm{d}\Gamma(\gamma)$$
$$+ o(\mathrm{W}_{\mathrm{W}_2}(\mathbb{P}, \mathbb{Q})). \quad (7)$$

In the following, we note $\nabla_{\mathrm{W}_{\mathrm{W}_2}}\mathbb{F}(\mathbb{P})$ such a gradient.

We can also define a notion of strong sub- and superdifferential, as well as gradient, by allowing the coupling $\Gamma$ to be non-optimal, in contrast with the previous definitions.

**Definition 3.4.** $\xi \in L^2(\mathbb{P}, T\mathcal{P}_2(\mathcal{M}))$ is a strong subdifferential of $\mathbb{F}$ at $\mathbb{P}$ if for all $\mathbb{Q} \in \mathcal{P}_2(\mathcal{P}_2(\mathcal{M}))$, for all $\Gamma \in \mathcal{P}_2(\mathcal{P}_2(T\mathcal{M}))$ s.t. $\phi^{\mathcal{M}}_{\#}\Gamma = \mathbb{P}$, $\phi^{\exp}_{\#}\Gamma = \mathbb{Q}$,

$$\mathbb{F}(\mathbb{Q}) \geq \mathbb{F}(\mathbb{P}) + \iint \langle \xi(\pi^{\mathcal{M}}_{\#}\gamma)(x), v\rangle_x \, \mathrm{d}\gamma(x,v)\mathrm{d}\Gamma(\gamma)$$
$$+ o\left(\sqrt{\iint \|v\|^2_x \mathrm{d}\gamma(x,v)\mathrm{d}\Gamma(\gamma)}\right). \quad (8)$$

Strong superdifferentials and gradients are defined similarly.

The latter definition is particularly useful when perturbing a measure along a non-optimal direction, as in the case of the forward Euler schemes we will compute in the next section.

We now turn to examples of functional on $\mathcal{P}_2(\mathcal{P}_2(\mathcal{M}))$ that take the form of free energies. Given $\mathcal{F} : \mathcal{P}_2(\mathcal{M}) \to \mathbb{R}$, we define a potential energy $\mathbb{V} : \mathcal{P}_2(\mathcal{P}_2(\mathcal{M})) \to \mathbb{R}$ as $\mathbb{V}(\mathbb{P}) = \int \mathcal{F}(\mu)\mathrm{d}\mathbb{P}(\mu)$. Analogously to classical Wasserstein gradients, its WoW gradient is obtained as $\nabla_{\mathrm{W}_{\mathrm{W}_2}}\mathbb{V}(\mathbb{P})(\mu) = \nabla_{\mathrm{W}_2}\mathcal{F}(\mu)$. Given a kernel $\mathcal{W} : \mathcal{P}_2(\mathbb{R}^d) \times \mathcal{P}_2(\mathbb{R}^d) \to \mathbb{R}$, we define interaction energies as $\mathbb{W}(\mathbb{P}) = \iint \mathcal{W}(\mu, \nu) \, \mathrm{d}\mathbb{P}(\mu)\mathrm{d}\mathbb{P}(\nu)$, and their WoW gradients are obtained as $\nabla_{\mathrm{W}_{\mathrm{W}_2}}\mathbb{W}(\mathbb{P})(\mu) = \int (\nabla_{\mathrm{W}_2,1}\mathcal{W}(\mu, \nu) + \nabla_{\mathrm{W}_2,2}\mathcal{W}(\nu, \mu)) \, \mathrm{d}\mathbb{P}(\nu)$. We refer to Appendix B.4 for more details.

Let us now define cylinder functions, which provide a class of Wasserstein differentiable functionals (von Renesse & Sturm, 2009; Dello Schiavo, 2020; Fornasier et al., 2023).

**Definition 3.5.** A functional $\mathcal{F} : \mathcal{P}_2(\mathcal{M}) \to \mathbb{R}$ is a cylinder if there exists $k \geq 0$, $F \in C^\infty_c(\mathbb{R}^k)$ and $V_1, \ldots, V_k \in C^\infty_c(\mathcal{M})$ such that, for all $\mu \in \mathcal{P}_2(\mathcal{M})$,

$$\mathcal{F}(\mu) = F\left(\int V_1\mathrm{d}\mu, \ldots, \int V_k\mathrm{d}\mu\right). \quad (9)$$

In this case, we note $\mathcal{F} \in \mathrm{Cyl}\big(\mathcal{P}_2(\mathcal{M})\big)$. Similarly, for $I$ an interval, we note $\mathcal{F} \in \mathrm{Cyl}\big(I \times \mathcal{P}_2(\mathcal{M})\big)$, if $\mathcal{F}(t, \mu) = F\big(t, \int V_1 \mathrm{d}\mu, \ldots, \int V_k \mathrm{d}\mu\big)$ for every $t \in I$ and $\mu \in \mathcal{P}_2(\mathcal{M})$, this time for some $F \in C_c^\infty(I \times \mathbb{R}^k)$.

Using the chain rule, any $\mathcal{F} \in \mathrm{Cyl}\big(\mathcal{P}_2(\mathcal{M})\big)$ is Wasserstein differentiable and for all $\mu \in \mathcal{P}_2(\mathcal{M})$,

$$\nabla_{\mathrm{W}_2}\mathcal{F}(\mu) = \sum_{i=1}^k \frac{\partial}{\partial x_i} F\left(\int V_1 \mathrm{d}\mu, \cdots, \int V_k \mathrm{d}\mu\right) \nabla V_i. \tag{10}$$

This provides the main building block for defining a tangent space, in which we will show that WoW gradients reside.

**Definition 3.6.** The tangent space at $\mathbb{P} \in \mathcal{P}_2\big(\mathcal{P}_2(\mathcal{M})\big)$ is

$$T_{\mathbb{P}}\mathcal{P}_2\big(\mathcal{P}_2(\mathcal{M})\big) = \overline{\{\nabla_{\mathrm{W}_2}\varphi, \ \varphi \in \mathrm{Cyl}\big(\mathcal{P}_2(\mathcal{M})\big)\}} \tag{11}$$

where the closure is taken in the space $L^2(\mathbb{P}, T\mathcal{P}_2(\mathcal{M}))$.

We now justify the definition of this tangent space. We show the existence of velocity fields $(v_t)_t$ belonging to the latter, associated to any absolutely continuous curves $(\mathbb{P}_t)_t$, such that the pair $(v_t, \mathbb{P}_t)_t$ satisfy a continuity equation. We recall that a curve $(\mathbb{P}_t)_{t \in [0,1]}$ is absolutely continuous if there exists $g \in L^1([0,1])$ such that $\mathrm{W}_{\mathrm{W}_2}(\mathbb{P}_s, \mathbb{P}_t) \leq \int_s^t g(u)\mathrm{d}u$, and its metric derivative is $|\mathbb{P}'|(t) = \lim_{h \to 0} \frac{1}{h}\mathrm{W}_{\mathrm{W}_2}(\mathbb{P}_{t+h}, \mathbb{P}_t)$, which exists a.e. (Ambrosio et al., 2008, Th. 1.1.2).

**Proposition 3.7.** *Let $(\mathbb{P}_t)_{t \in I}$ be an absolutely continuous curve on $\mathcal{P}_2\big(\mathcal{P}_2(\mathcal{M})\big)$. Then, for a.e. $t \in I$, there exists $v_t \in T_{\mathbb{P}_t}\mathcal{P}_2\big(\mathcal{P}_2(\mathcal{M})\big)$ such that $\|v_t\|_{L^2(\mathbb{P}_t, T\mathcal{P}_2(\mathcal{M}))} \leq |\mathbb{P}'|(t)$ and for all $\varphi \in \mathrm{Cyl}(I \times \mathcal{P}_2(\mathcal{M}))$,*

$$\iint \big(\partial_t \varphi_t(\mu) + \langle \nabla_{\mathrm{W}_2}\varphi_t(\mu), v_t(\mu)\rangle_{L^2(\mu)}\big) \, \mathrm{d}\mathbb{P}_t(\mu)\mathrm{d}t = 0. \tag{12}$$

The proof of Proposition 3.7 is deferred to Appendix C.3. We leave the investigation of the converse implication to future work, *i.e.* that satisfying the (weak) continuity equation (12) implies absolute continuity of the curve $(\mathbb{P}_t)_t$. We then have the following properties, which show that elements of the tangent space are strong gradients and are unique. Their proofs are deferred to Appendix C.4 and Appendix C.5.

**Proposition 3.8.** *Let $\xi \in \partial^- \mathbb{F}(\mathbb{P}) \cap T_{\mathbb{P}}\mathcal{P}_2\big(\mathcal{P}_2(\mathcal{M})\big)$. Then $\xi$ is a strong subdifferential of $\mathbb{F}$ at $\mathbb{P}$.*

**Proposition 3.9.** *There is at most one element in $\partial^- \mathbb{F}(\mathbb{P}) \cap \partial^+ \mathbb{F}(\mathbb{P}) \cap T_{\mathbb{P}}\mathcal{P}_2\big(\mathcal{P}_2(\mathcal{M})\big)$.*

As $L^2(\mathbb{P}, T\mathcal{P}_2(\mathcal{M}))$ and the tangent space are Hilbert spaces, one can always decompose a WoW gradient with a part in $T_{\mathbb{P}}\mathcal{P}_2\big(\mathcal{P}_2(\mathcal{M})\big)$ and another part orthogonal to it. We show in Appendix C.5 that under technical assumptions,

this orthogonal part has a null contribution in the Taylor expansion given in (7). Thus, in this case, we can restrict ourselves to the unique WoW gradient belonging to the tangent space, in particular to write optimization schemes.

## 4. WoW Gradient Flows

In this section, we aim at minimizing $\mathbb{F} : \mathcal{P}_2\big(\mathcal{P}_2(\mathbb{R}^d)\big) \to \mathbb{R}$ some functional. We first show the existence of the WoW gradient flow of this functional as the limit of the JKO scheme (Jordan et al., 1998) for $\mathbb{F}$ convex along generalized geodesics (Ambrosio et al., 2008). Then, building on the differentiable structure of the space introduced earlier, we propose a forward (explicit) scheme that is computationally more efficient in practice than the implicit JKO scheme, and tractable for relevant functionals.

### 4.1. Optimization Schemes on $\mathcal{P}_2\big(\mathcal{P}_2(\mathbb{R}^d)\big)$

**JKO Scheme.** Let $\mathbb{P}_0 \in \mathcal{P}_2\big(\mathcal{P}_{2,\mathrm{ac}}(\mathbb{R}^d)\big)$. The JKO sheme of $\mathbb{F}$ is defined, for all $k \geq 0$ and $\tau > 0$, as

$$\mathbb{P}_{k+1} \in \underset{\mathbb{P} \in \mathcal{P}_2(\mathcal{P}_{\mathrm{ac}}(\mathbb{R}^d))}{\mathrm{argmin}} \ \frac{1}{2\tau}\mathrm{W}_{\mathrm{W}_2}(\mathbb{P}, \mathbb{P}_k)^2 + \mathbb{F}(\mathbb{P}). \tag{13}$$

Its Wasserstein gradient flow is defined as the limit when $\tau \to 0$. Leveraging (Ambrosio et al., 2008, Theorem 4.0.4), we show in the next Proposition the existence of the flow for functionals $\mathbb{F}$ that are $\lambda$-convex along generalized geodesics $\mathbb{P}_t = \big(\big((1-t)\mathrm{T}_{\pi^1}^{\pi^2} + t\mathrm{T}_{\pi^1}^{\pi^3}\big)_\# \pi^1\big)_\# \Gamma$ between $\mathbb{Q}, \mathbb{O} \in \mathcal{P}_2\big(\mathcal{P}_2(\mathbb{R}^d)\big)$, where $\Gamma \in \Pi(\mathbb{P}, \mathbb{Q}, \mathbb{O})$ satisfies $\pi_\#^{1,2}\Gamma \in \Pi_o(\mathbb{P}, \mathbb{Q})$, $\pi_\#^{1,3}\Gamma \in \Pi_o(\mathbb{P}, \mathbb{O})$ with $\pi^{1,2} : (x, y, z) \mapsto (x, y)$, $\pi^{1,3} : (x, y, z) \mapsto (x, z)$ and $\mathbb{P} \in \mathcal{P}_2\big(\mathcal{P}_{2,\mathrm{ac}}(\mathbb{R}^d)\big)$. Since $\mathbb{P} \in \mathcal{P}_2\big(\mathcal{P}_{2,\mathrm{ac}}(\mathbb{R}^d)\big)$, there is always an OT map starting from $\mu \sim \mathbb{P}$ towards any $\nu \in \mathcal{P}_2(\mathbb{R}^d)$, which we write $\mathrm{T}_\mu^\nu$.

**Proposition 4.1.** *Let $\lambda \geq 0$. Let $\mathbb{F} : \mathcal{P}_2\big(\mathcal{P}_2(\mathbb{R}^d)\big) \to \mathbb{R}$ be proper, coercive, lower-semi continuous and $\lambda$-convex along generalized geodesics, i.e., satisfying for all $t \in [0, 1]$,*

$$\mathbb{F}(\mathbb{P}_t) \leq (1-t)\mathbb{F}(\mathbb{P}_0) + t\mathbb{F}(\mathbb{P}_1) - \frac{\lambda t(1-t)}{2}\mathrm{W}_{\mathrm{W}_2}^2(\mathbb{P}_0, \mathbb{P}_1), \tag{14}$$

*for $\mathbb{P}_t = \big(\big((1-t)\mathrm{T}_{\pi^1}^{\pi^2} + t\mathrm{T}_{\pi^1}^{\pi^3}\big)_\# \pi^1\big)_\# \Gamma$, $\Gamma \in \Pi(\mathbb{P}, \mathbb{Q}, \mathbb{O})$ that satisfies $\pi_\#^{1,2}\Gamma \in \Pi_o(\mathbb{P}, \mathbb{Q})$, $\pi_\#^{1,3}\Gamma \in \Pi_o(\mathbb{P}, \mathbb{O})$ and $\mathbb{P} \in \mathcal{P}_2\big(\mathcal{P}_{2,\mathrm{ac}}(\mathbb{R}^d)\big)$. Then, the gradient flow of $\mathbb{F}$ exists and is unique.*

The proof of Proposition 4.1 can be found in Appendix C.6. Examples of $\lambda$-convex $\mathbb{F}$ on $\mathcal{P}_2\big(\mathcal{P}_2(\mathbb{R}^d)\big)$ include potential energies for any $\mathcal{F}$ $\lambda$-convex along generalized geodesics on $\mathcal{P}_2(\mathbb{R}^d)$, and interaction energies for $\lambda = 0$ and $\mathcal{W}$ jointly convex along generalized geodesics, see Appendix B.5.

**Forward Scheme.** Given the existence of the WoW gradient of $\mathbb{F}$, as established in the previous section, we propose an alternative to the implicit JKO scheme: a forward scheme, commonly referred to as Wasserstein gradient descent, defined as follows

$$\forall k \geq 0, \; \mathbb{P}_{k+1} = \exp_{\mathbb{P}_k}\big(-\tau \nabla_{\mathrm{W}_{\mathrm{W}_2}} \mathbb{F}(\mathbb{P}_k)\big). \quad (15)$$

At the "distribution particle" level in $\mathcal{P}_2(\mathcal{M})$, this means that for each distribution $\mu_k \sim \mathbb{P}_k$, we update it as

$$\mu_{k+1} = \exp_{\mu_k}\big(-\tau \nabla_{\mathrm{W}_{\mathrm{W}_2}} \mathbb{F}(\mathbb{P}_k)(\mu_k)\big). \quad (16)$$

In practice, we will mostly focus on distributions of the form $\mathbb{P} = \frac{1}{C}\sum_{c=1}^{C} \delta_{\mu^c}$ with $\mu^c = \frac{1}{n}\sum_{i=1}^{n} \delta_{x_i^c}$, which notably include labeled datasets (assuming for simplicity now that all classes $c = 1, \ldots, C$ contain $n$ examples). Thus, we apply to each particle in $\mathcal{M} = \mathbb{R}^d$ the update

$$\begin{aligned} x_{i,k+1}^c &= \exp_{x_{i,k}^c}\big(-\tau \nabla_{\mathrm{W}_{\mathrm{W}_2}} \mathbb{F}(\mathbb{P}_k)(\mu_k^c)(x_{i,k}^c)\big) \\ &= x_{i,k}^c - \tau \nabla_{\mathrm{W}_{\mathrm{W}_2}} \mathbb{F}(\mathbb{P}_k)(\mu_k^c)(x_{i,k}^c). \end{aligned} \quad (17)$$

We see that there are two levels of interactions for each particle in $\mathcal{M}$: one "intra-class" through the dependence in the distribution $\mu_c$ and one "inter-class" between the distributions $\mu^c \sim \mathbb{P}$ through the dependence in $\mathbb{P}_k$ in the gradient. Thus, we expect to observe an interaction between particles of each distribution $\mu^c$, but also between each distribution $\mu^c$.

### 4.2. Examples of Discrepancies

Classical functionals in the study of Wasserstein gradient flows are obtained as linear combinations of potential energies, interaction energies and internal energies (Santambrogio, 2015). We focus here on potential energies and interaction energies. We leave the study of internal energies on this space for future works. We refer to *e.g.* (von Renesse & Sturm, 2009; Sturm, 2024) for discussions of entropy functionals on this space.

A classical discrepancy to compare probability distributions, which can be written as a sum of a potential energy and an interaction energy, is the Maximum Mean Discrepancy (MMD) (Gretton et al., 2012). Given a positive definite kernel $K : \mathcal{P}_2(\mathbb{R}^d) \times \mathcal{P}_2(\mathbb{R}^d) \to \mathbb{R}$, $\mathbb{P}, \mathbb{Q} \in \mathcal{P}_2(\mathcal{P}_2(\mathbb{R}^d))$, let $\mathbb{F}(\mathbb{P}) = \frac{1}{2}\mathrm{MMD}^2(\mathbb{P}, \mathbb{Q})$ be defined as

$$\begin{aligned} \mathbb{F}(\mathbb{P}) &= \frac{1}{2}\iint K(\mu, \nu) \, \mathrm{d}(\mathbb{P} - \mathbb{Q})(\mu)\mathrm{d}(\mathbb{P} - \mathbb{Q})(\nu) \\ &= \mathbb{V}(\mathbb{P}) + \mathbb{W}(\mathbb{P}) + \mathrm{cst}, \end{aligned} \quad (18)$$

where $\mathbb{V}(\mathbb{P}) = \int \mathcal{V}(\mu)\mathrm{d}\mathbb{P}(\mu)$, $\mathcal{V}(\mu) = -\int K(\mu, \nu)\mathrm{d}\mathbb{Q}(\nu)$, $\mathbb{W}(\mathbb{P}) = \frac{1}{2}\iint K(\mu, \nu) \, \mathrm{d}\mathbb{P}(\mu)\mathrm{d}\mathbb{P}(\nu)$ and the constant only depends on $\mathbb{Q}$ that is fixed. For $K$, we will use kernels based on the Sliced-Wasserstein (SW) (Rabin et al., 2012; Bonneel et al., 2015), defined between $\mu, \nu \in \mathcal{P}_2(\mathbb{R}^d)$ as

$$\mathrm{SW}_2^2(\mu, \nu) = \int_{S^{d-1}} \mathrm{W}_2^2(P_\#^\theta \mu, P_\#^\theta \nu) \, \mathrm{d}\sigma(\theta), \quad (19)$$

with $S^{d-1} = \{\theta \in \mathbb{R}^d, \|\theta\|_2 = 1\}$ the sphere, $P^\theta(x) = \langle x, \theta \rangle$ the coordinate of the projection of $x \in \mathbb{R}^d$ on the line $\theta\mathbb{R}$ for $\theta \in S^{d-1}$, and $\sigma$ the uniform measure on $S^{d-1}$. For instance, positive definite kernels include the Gaussian SW kernel $K(\mu, \nu) = e^{-\mathrm{SW}_2^2(\mu,\nu)/h}$ (Kolouri et al., 2016; Carriere et al., 2017; Meunier et al., 2022). We also experiment with the Riesz SW kernel $K(\mu, \nu) = -\mathrm{SW}_2(\mu, \nu)$ in analogy with the Riesz kernel (sometimes referred to as negative distance kernel), $k(x, y) = -\|x - y\|_2$ on $\mathbb{R}^d \times \mathbb{R}^d$, which is not positive definite, but which has demonstrated very good results in practice (Hertrich et al., 2024b) and does not require tuning a bandwidth $h$.

**WoW gradient of the MMD.** Given $\nu \in \mathcal{P}_2(\mathbb{R}^d)$, if $K_\nu : \mu \mapsto K(\mu, \nu)$ is a Wasserstein differentiable functional, then $\mathbb{F}$ is differentiable, and its WoW gradient at $\mathbb{P} \in \mathcal{P}_2(\mathcal{P}_2(\mathbb{R}^d))$ is of the form, for all $\mu \in \mathcal{P}_2(\mathbb{R}^d)$,

$$\nabla_{\mathrm{W}_{\mathrm{W}_2}} \mathbb{F}(\mathbb{P})(\mu) = \int \nabla_{\mathrm{W}_2} K_\nu(\mu) \, \mathrm{d}(\mathbb{P} - \mathbb{Q})(\nu). \quad (20)$$

For the Gaussian SW kernel $K(\mu, \nu) = e^{-\frac{1}{2h}\mathrm{SW}_2^2(\mu,\nu)}$, denoting $\mathcal{F}(\mu) = \frac{1}{2}\mathrm{SW}_2^2(\mu, \nu)$, its gradient can be obtained by the chain rule as

$$\nabla_{\mathrm{W}_2} K_\nu(\mu) = -\frac{1}{h}e^{-\frac{1}{2h}\mathrm{SW}_2^2(\mu,\nu)}\nabla_{\mathrm{W}_2}\mathcal{F}(\mu), \quad (21)$$

where $\nabla_{\mathrm{W}_2}\mathcal{F}(\mu) = \int_{S^{d-1}} \psi_\theta'(\langle x, \theta \rangle)\theta \, \mathrm{d}\sigma(\theta)$ with $\psi_\theta$ the Kantorovich potential between $P_\#^\theta \mu$ and $P_\#^\theta \nu$ (Bonnotte, 2013, Proposition 5.1.7). In practice, the Sliced-Wasserstein distance, involving an integral over the sphere, is approximated through Monte Carlo. Moreover, for discrete measures $\mathbb{P} = \frac{1}{C}\sum_{c=1}^{C} \delta_{\mu^{c,n}}$ and $\mathbb{Q} = \frac{1}{C}\sum_{c=1}^{C} \delta_{\nu^{c,n}}$ with $\mu^{c,n} = \frac{1}{n}\sum_{i=1}^{n} \delta_{x_i^c}$ and $\nu^{c,n} = \frac{1}{n}\sum_{i=1}^{n} \delta_{y_i^c}$, we use autodifferentiation over $\mathbf{x} := (x_i^c)_{i,c}$ of $F(\mathbf{x}) = \mathbb{F}(\mathbb{P})$, and rescale the Euclidean gradient of $F$ by $n \times C$ to obtain the WoW gradient $\nabla_{\mathrm{W}_{\mathrm{W}_2}} \mathbb{F}(\mathbb{P})(\mu^{c,n})(x_i^c) = nC\nabla_{i,c}F(\mathbf{x})$. This is analogous to the Wasserstein gradient case, and coincides with the WoW gradient for functionals with a closed-form over discrete measures (see Proposition B.7).

## 5. Applications

In this section, we minimize the MMD on $\mathcal{P}_2(\mathcal{P}_2(\mathbb{R}^d))$ to solve various tasks[1]. We represent labeled datasets with $C$ classes as distributions $\mathbb{P} = \frac{1}{C}\sum_{c=1}^{C} \delta_{\mu^{c,n}}$, where each

---

[1]Code available at https://github.com/clbonet/
Flowing_Datasets_with_WoW_Gradient_Flows.

Iter 0        Iter 10        Iter 25        Iter 100

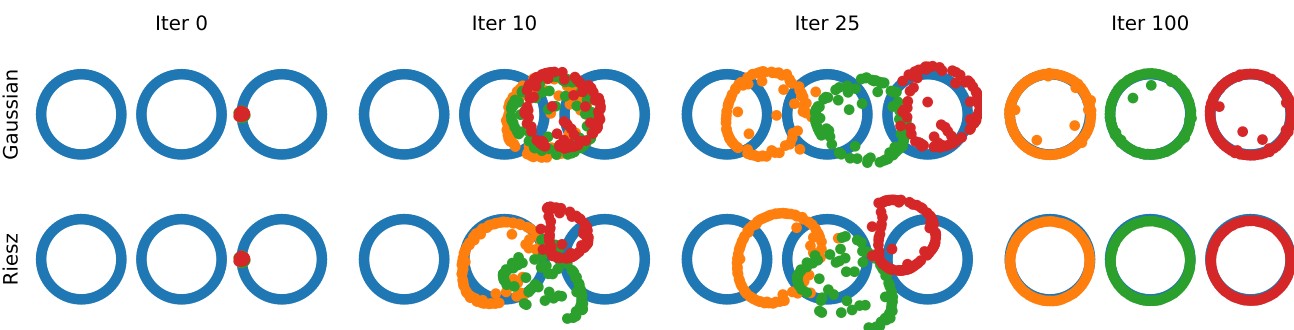

Figure 1: Minimization of $\mathbb{F}(\mathbb{P}) = \frac{1}{2}\mathrm{MMD}^2(\mathbb{P}, \mathbb{Q})$ with $\mathbb{Q}$ a mixture of 3 rings, and with kernels either the Gaussian SW kernel with bandwidth $h = 0.05$ or Riesz SW kernel, for a learning rate of $\tau = 0.1$. We observe that they first form a ring for each distribution, and then each ring converges to a target ring.

$\mu^{c,n} = \frac{1}{n}\sum_{i=1}^{n}\delta_{x_i^c}$ is the distribution of samples belonging to class $c$. We emphasize that we are the first to represent labeled datasets this way. We first verify on synthetic data and datasets of images that minimizing such distance allows to transport classes between the source and target. Then, we leverage this property on a dataset distillation and a transfer learning task. We focus here on learning target distributions of the form $\mathbb{Q} = \frac{1}{C}\sum_{k=1}^{C}\delta_{\nu^{c,n}}$ where $\nu^{c,n} = \frac{1}{n}\sum_{i=1}^{n}\delta_{y_i^c}$ are empirical distributions, each $\nu^{c,n}$ has the same number of particles $n$ and the number of class $C$ is supposed to be known. Similarly as (Hertrich et al., 2024b), we add a momentum to accelerate the scheme for image-based datasets. We refer to Appendix D for more details about the experiments, as well as additional experiments using other kernels and an ablation study for the number of projections to approximate SW. Related works (Alvarez-Melis & Fusi, 2021; Hua et al., 2023) are described in detail in Appendix E.

**Synthetic Data.** We illustrate on Figure 1 the evolution of particles when minimizing the MMD with kernels $K(\mu, \nu) = e^{-\mathrm{SW}_2^2(\mu,\nu)/(2h)}$ and $K(\mu, \nu) = -\mathrm{SW}_2(\mu, \nu)$, for a target being the three-ring dataset. Each ring represents a distribution $\nu^{c,n}$ with $n = 80$, and the target is thus a mixture of three Dirac, *i.e.*, $\mathbb{Q} = \frac{1}{3}\sum_{c=1}^{3}\delta_{\nu^{c,n}}$. We learn a distribution $\mathbb{P}$ of the same form, with the same number of particles for each distribution. We observe for both kernels that the particles of each distribution $\mu^{c,n}$ (*i.e.*, the different point clouds) form a ring early in the gradient flow dynamics, and then move in a structured manner towards the target. This illustrates the two level of interactions at the intra and inter distributions levels. In Appendix D.2, we add comparisons with other hyperparameters and other kernels. Overall, the kernel $K(\mu, \nu) = -\mathrm{SW}_2(\mu, \nu)$ is the simplest to use, as it does not require tuning a bandwidth, and converges well in general. Thus, in the following experiments, we restrict ourselves to this kernel, and name the resulting loss MMDSW.

**Domain Adaptation.** We now focus on the case where both the source $\mathbb{P}_0$ and the target $\mathbb{Q}$ are distributions of im-

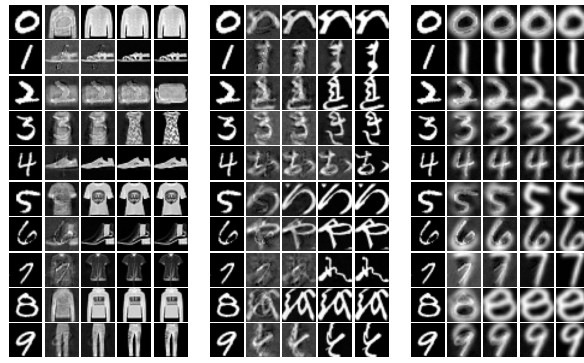

Figure 2: Samples along the flow from MNIST to FMNIST (**Left**), KMNIST (**Middle**) and USPS (**Right**).

ages with $C$ classes. Thus, we have $\mathbb{P}_0 = \frac{1}{C}\sum_{c=1}^{C}\delta_{\mu^{c,n}}$ and $\mathbb{Q} = \frac{1}{C}\sum_{c=1}^{C}\delta_{\nu^{c,n}}$, and $\nu^{c,n}$, $\mu^{c,n}$ represent the empirical distribution of images belonging to the class $c$. We consider the *NIST* datasets, *i.e.*, MNIST (LeCun & Cortes, 2010), Fashion-MNIST (FMNIST) (Wang et al., 2018), KM-NIST (Clanuwat et al., 2018) and USPS (Hull, 1994). These datasets all have $C = 10$ classes and are of size $28 \times 28$ (except for USPS which is upscaled to $28 \times 28$). We also consider CIFAR10 (Krizhevsky et al., 2009) and SVHN (Netzer et al., 2011) which are of size $32 \times 32 \times 3$. We first show in Figure 2 examples of trajectories starting from MNIST to the other *NIST* datasets (with step size $\tau = 0.05$, momentum $m = 0.9$ and $n = 200$). We see that samples from MNIST are sent to samples of the target dataset, *i.e.* that the flow converges well. We also observe that images from each class are mapped one-to-one to images within the same class (see Figure 11 in the Appendix), without overlap or collapse across classes.

To verify this quantitatively, we perform a domain adaptation task as in (Alvarez-Melis & Fusi, 2021, Section 7.3). Here, we first train a classifier on 5000 samples of MNIST with $n = 500$ images by class. Then, we flow the other

Table 1: Accuracy of the classifier on synthetic datasets. We compare Distribution Matching (DM) with the MMD with Riesz SW kernel on MNIST and Fashion MNIST, using $p \in \{1, 10, 50\}$ synthetic images by class.

| Dataset | $p$ | $\psi^\theta = \mathcal{A}^\omega = $ Id | | $\psi^\theta = $ Id | | $\mathcal{A}^w = $ Id | | $\mathcal{A}^w + \psi^\theta$ | | Baselines | |
|---|---|---|---|---|---|---|---|---|---|---|---|
| | | DM | MMDSW | DM | MMDSW | DM | MMDSW | DM | MMDSW | Random | Full data |
| MNIST | 1 | $61.1_{\pm6.5}$ | $\mathbf{66.5}_{\pm5.5}$ | - | $66.8_{\pm5.3}$ | $\mathbf{87.8}_{\pm0.6}$ | $60.3_{\pm3.4}$ | $\mathbf{87.7}_{\pm0.5}$ | $60.9_{\pm3.3}$ | $55.8_{\pm2.0}$ | |
| | 10 | $88.2_{\pm2.8}$ | $\mathbf{93.2}_{\pm0.7}$ | $88.7_{\pm3.3}$ | $\mathbf{93.8}_{\pm0.7}$ | $\mathbf{97.0}_{\pm0.1}$ | $96.4_{\pm0.2}$ | $\mathbf{97.0}_{\pm0.1}$ | $96.4_{\pm0.3}$ | $92.2_{\pm1.1}$ | 99.4 |
| | 50 | $95.9_{\pm0.9}$ | $97.0_{\pm0.2}$ | $95.3_{\pm1.4}$ | $97.5_{\pm0.1}$ | $\mathbf{98.4}_{\pm0.1}$ | $\mathbf{98.4}_{\pm0.1}$ | $\mathbf{98.4}_{\pm0.1}$ | $\mathbf{98.4}_{\pm0.1}$ | $97.6_{\pm0.2}$ | |
| FMNIST | 1 | $54.4_{\pm3.2}$ | $\mathbf{60.0}_{\pm4.1}$ | - | $60.6_{\pm3.6}$ | $58.7_{\pm0.4}$ | $\mathbf{60.9}_{\pm2.6}$ | $58.7_{\pm0.5}$ | $\mathbf{60.8}_{\pm2.2}$ | $49.0_{\pm7.5}$ | |
| | 10 | $74.6_{\pm1.0}$ | $\mathbf{76.7}_{\pm1.0}$ | $74.7_{\pm0.8}$ | $76.6_{\pm1.1}$ | $81.2_{\pm2.3}$ | $78.0_{\pm0.9}$ | $\mathbf{82.5}_{\pm0.3}$ | $78.9_{\pm1.2}$ | $75.3_{\pm0.7}$ | 92.4 |
| | 50 | $81.3_{\pm0.5}$ | $\mathbf{84.2}_{\pm0.1}$ | $81.4_{\pm1.0}$ | $\mathbf{85.0}_{\pm0.2}$ | $\mathbf{87.6}_{\pm0.2}$ | $\mathbf{87.6}_{\pm0.2}$ | $87.5_{\pm0.1}$ | $\mathbf{87.6}_{\pm0.2}$ | $83.2_{\pm0.2}$ | |

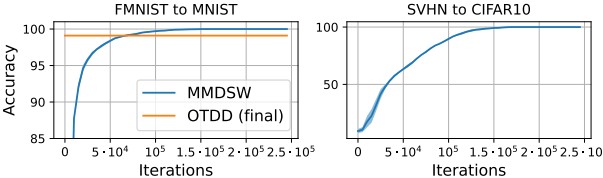

Figure 3: Accuracy of the pretrained classifiers along the flow from FMNIST (**Left**) and SVHN (**Right**) towards MNIST and CIFAR10.

datasets to MNIST (with $\tau = 0.1$ and momentum $m = 0.9$), and measure the accuracy of the pretrained classifier on the flowed dataset. Note that while we use the class labels of the flowed dataset to perform the gradient flow dynamics, we do not know a priori which class in the flowed dataset corresponds to which class in MNIST, yet it is needed for the evaluation of domain adaptation. To perform this alignment, we solve an OT problem with $W_2^2$ as groundcost between $\mathbb{P}$ and $\mathbb{Q}$ (*i.e.*, the WoW OT problem) with $\mathbb{P}$ the distributions obtained at the end of the flow dynamic and $\mathbb{Q}$ the ones of the target dataset. Since these distributions have a finite support of the same size ($C$), solving this OT problem provides such an alignment: we can associate a prediction of the pretrained model to an image and a "true class" of the flowed dataset. We also perform this experiment with a pretrained neural network on CIFAR10, flowing SVHN toward CIFAR10, with $n = 100$ samples by class, step size $\tau = 0.1$ and momentum $m = 0.9$.

On Figure 3, we report the accuracy of the pretrained classifier on the data flowed starting from FMNIST towards MNIST and from SVHN towards CIFAR10, over the iterations (averaged over 3 flows started at different splits of the source data). We also report the value from (Alvarez-Melis & Fusi, 2021) using OTDD on the MNIST dataset. We observe that the classifier converges to 100% accuracy for a sufficient number of iterations. This demonstrates that the flow is able to perfectly match one class from the source dataset with a class of the target dataset, on which the classifier has been trained.

We note that in a realistic setting of unsupervised domain adaptation, we would not have access to the labels of the

source dataset (Courty et al., 2016). Thus, to flow the data as we did just earlier, we would need first to find pseudo-labels on the source datasets, *e.g.* with clustering (Alvarez-Melis & Fusi, 2021; El Hamri et al., 2022). However, this is not the goal of the paper.

**Dataset Distillation.** Dataset distillation or condensation (Wang et al., 2018) seeks to produce a compact synthetic dataset derived from a large training set, such that training a neural network on the synthetic data yields performance close to that obtained with the full dataset. Zhao & Bilen (2023) proposed to learn the synthetic dataset by performing Distribution Matching, *i.e.*, denoting $\nu^c$ the distribution of each class $c$ of the target dataset, they minimize

$$\mathcal{F}\big((\mu^c)_c\big) = \mathbb{E}_{\theta,\omega}\left[\sum_{c=1}^{C} \text{MMD}_k^2\big(\psi^\theta_\# \mathcal{A}^\omega_\# \mu^c, \psi^\theta_\# \mathcal{A}^\omega_\# \nu^c\big)\right],$$
(22)

with $k$ the linear kernel $k(x, y) = \langle x, y \rangle$, $\mathcal{A}^\omega : \mathbb{R}^d \to \mathbb{R}^d$ a random data augmentation (*e.g.* rotation, cropping, see (Zhao & Bilen, 2021)) and $\psi^\theta : \mathbb{R}^d \to \mathbb{R}^{d'}$ with $d' \ll d$ a randomly initialized neural network used to embed the data.

Let $\mathbb{Q} = \frac{1}{C}\sum_{c=1}^{C}\delta_{\nu^c}$ be the target dataset, $\phi^{\theta,\omega}(\mu) = \psi^\theta_\# \mathcal{A}^\omega_\# \mu$ and $\mathbb{P} = \frac{1}{C}\sum_{c=1}^{C}\delta_{\mu^c}$. Note that $\phi^{\theta,\omega}_\#\mathbb{P} = \frac{1}{C}\sum_{c=1}^{C}\delta_{\psi^\theta_\# \mathcal{A}^\omega_\# \mu^c}$. We propose to minimize

$$\tilde{\mathbb{F}}(\mathbb{P}) = \mathbb{E}_{\theta,\omega}\left[\text{MMD}_K^2(\phi^{\theta,\omega}_\#\mathbb{P}, \phi^{\theta,\omega}_\#\mathbb{Q})\right],$$
(23)

with Riesz SW kernel $K$ between distributions. We compare on Table 1 the accuracy of a classifier on a test set of MNIST and FMNIST, trained on the synthetic dataset with $p \in \{1, 10, 50\}$ samples by class, either generated with MMD with Riesz SW kernel (MMDSW) or with Distribution Matching (DM), in 4 scenarios: in the ambient space with ($\psi^\theta = $ Id) and without augmentation ($\psi^\theta = \mathcal{A}^\omega = $ Id), and with an embedding with ($\mathcal{A}^\omega + \psi^\theta$) and without an augmentation ($\mathcal{A}^\omega = $ Id). We solve it with a stochastic gradient descent, sampling one augmentation and embedding at each step, for 20K iterations and initializing the samples on true data. The results are averaged over 3 synthetic datasets obtained initializing the flow at different samples, and 5

Table 2: Accuracy of classifier on augmented datasets for $k \in \{1, 10, 10, 100\}$. M refers to MNIST, F to Fashion MNIST, K to KMNIST and U to USPS.

| Dataset | $k$ | Train on $\mathbb{Q}$ | MMDSW | OTDD | (Hua et al., 2023) |
|---|---|---|---|---|---|
| M to F | 1 | $26.0_{\pm 5.3}$ | $\mathbf{40.5}_{\pm 4.7}$ | $30.5_{\pm 4.2}$ | $36.4_{\pm 3.3}$ |
| | 5 | $38.5_{\pm 6.7}$ | $61.5_{\pm 4.6}$ | $59.7_{\pm 1.8}$ | $\mathbf{62.7}_{\pm 1.1}$ |
| | 10 | $53.9_{\pm 7.9}$ | $65.4_{\pm 1.5}$ | $64.0_{\pm 1.4}$ | $\mathbf{66.2}_{\pm 1.0}$ |
| | 100 | $71.1_{\pm 1.5}$ | $\mathbf{74.7}_{\pm 0.8}$ | - | $73.5_{\pm 0.7}$ |
| M to K | 1 | $18.4_{\pm 3.1}$ | $\mathbf{20.9}_{\pm 2.0}$ | $18.8_{\pm 2.1}$ | $19.4_{\pm 1.9}$ |
| | 5 | $25.9_{\pm 4.0}$ | $37.4_{\pm 2.2}$ | $31.3_{\pm 1.4}$ | $\mathbf{39.0}_{\pm 1.0}$ |
| | 10 | $30.9_{\pm 4.6}$ | $\mathbf{44.7}_{\pm 1.8}$ | $34.1_{\pm 0.9}$ | $44.1_{\pm 1.2}$ |
| | 100 | $60.1_{\pm 1.1}$ | $\mathbf{66.8}_{\pm 0.8}$ | $66.3_{\pm 0.9}$ | $62.4_{\pm 1.2}$ |
| M to U | 1 | $32.4_{\pm 7.9}$ | $37.4_{\pm 6.1}$ | $\mathbf{39.5}_{\pm 7.9}$ | $35.0_{\pm 5.6}$ |
| | 5 | $51.4_{\pm 9.8}$ | $73.0_{\pm 1.0}$ | $\mathbf{73.3}_{\pm 1.4}$ | $69.6_{\pm 1.3}$ |
| | 10 | $60.3_{\pm 10.1}$ | $\mathbf{77.2}_{\pm 1.2}$ | $72.7_{\pm 2.7}$ | $75.6_{\pm 1.2}$ |
| | 100 | $87.5_{\pm 0.7}$ | $\mathbf{89.7}_{\pm 0.4}$ | - | $88.1_{\pm 0.6}$ |

training of the classifier. On a Nvidia v100 GPU, the flow implemented in Jax (Bradbury et al., 2018) runs in around 10 minutes with the embedding and in 30 seconds without it. The baseline "random" refers to the classifier trained on data sampled randomly from the original training set, and "full data" to the classifier trained on the full training set. We observe on Table 1 that MMDSW consistently outperforms DM when flowing in the ambient space, and is competitive when adding an embedding. This indicates that adding interactions between classes appears to improve the results, possibly by distributing the samples more effectively and mitigating the presence of ambiguous samples near class borders.

**Transfer Learning.** We now focus on the task of $k$-shot learning. In this setting, we are interested in training a classifier for datasets which have $k$ samples by class, where $k$ is typically small. Following (Alvarez-Melis & Fusi, 2021; Hua et al., 2023), we propose to augment the dataset by generating new synthetic samples for each class. To do this, we will flow a larger source dataset, with possibly different classes, towards the small target dataset, and then concatenate the synthetic and true samples to train the classifier on it. More precisely, let $\mathbb{Q} = \frac{1}{C} \sum_{c=1}^{C} \delta_{\nu^{c,k}}$ the target dataset, with $\nu^{c,k} = \frac{1}{k} \sum_{i=1}^{k} \delta_{y_i^c}$ an empirical distribution with $k$ samples, representing the distribution of the class $c$. Let $\mathbb{P}_0 = \frac{1}{C} \sum_{c=1}^{C} \delta_{\mu^{c,n}}$ be a source dataset, with $\mu^{c,n} = \frac{1}{n} \sum_{i=1}^{n} \delta_{x_i^c}$ and $n = 200$. Then, the goal is to flow $\mathbb{P}_0$ towards $\mathbb{Q}$ by minimizing $\mathbb{F}(\mathbb{P}) = \frac{1}{2} \mathrm{MMD}^2(\mathbb{P}, \mathbb{Q})$ with kernel $K(\mu, \nu) = -\mathrm{SW}_2(\mu, \nu)$. We expect to augment each class $c$ of $\mathbb{Q}$ with $n$ samples. Then, we train a classifier with a LeNet5 architecture on the dataset obtained as $\hat{\mathbb{Q}} = \frac{1}{C} \sum_{c=1}^{C} \delta_{\eta^{c,n+k}}$ with $\eta^{c,n+k} = \frac{1}{n+k} \sum_{i=1}^{n+k} \delta_{z_i^c}$ where $z_i^c = x_i^c$ for $i \leq n$ and $z_i^c = y_{i-n}^c$ for $i > n$. We report the results for MNIST as $\mathbb{P}_0$ and FMNIST, KMNIST and USPS as $\mathbb{Q}$ on Table 2 for $k \in \{1, 5, 10, 100\}$, compared with the baseline where we train directly on $\mathbb{Q}$, and the baselines where we trained on the synthetic

Table 3: Runtime in seconds for the transfer learning experiment from MNIST to Fashion MNIST.

| Dataset | $k$-shot | MMDSW | OTDD | (Hua et al., 2023) |
|---|---|---|---|---|
| M to F | 1 | $13.95 \pm 1.37$ | $294.53 \pm 5.21$ | $131.77 \pm 2$ |
| | 5 | $14.12 \pm 0.30$ | $1130.89 \pm 108$ | $132.98 \pm 1.1$ |
| | 10 | $14.30 \pm 0.29$ | $2294.13 \pm 48$ | $134.35 \pm 0.75$ |
| | 100 | $47.75 \pm 0.27$ | - | $164.19 \pm 0.6$ |

data obtained by minimizing OTDD (Alvarez-Melis & Fusi, 2021) or the MMD with product kernel as in (Hua et al., 2023). The results are averaged over 5 training of the networks, and 3 outputs of the flows. Both methods have been reimplemented and we add more details in Appendix D.6. We observe that all three methods improve upon the baseline, with a slight advantage for MMDSW.

**Complexity.** Given $\mathbb{P} = \frac{1}{C} \sum_{c=1}^{C} \delta_{\mu^{c,n}}$ and $\mathbb{Q} = \frac{1}{C} \sum_{c=1}^{C} \delta_{\nu^{c,n}}$ with $\mu^{c,n}$ and $\nu^{c,n}$ discrete distributions with $n$ samples each, the MMD with a Sliced-Wasserstein based kernel requires to compute $C^2$ Sliced-Wasserstein distances, which has a total complexity of $O(C^2 Ln(\log n + d))$ using $L$ projections to approximate SW. We report on Table 3 the runtimes for the transfer learning experiment, averaged over 3 outputs of the flows and trained for 5K epochs for each method. MMDSW is much faster than both OTDD and the MMD with product kernel, at least with our implementations in jax detailed in Appendix D.6. Both OTDD and the method of Hua et al. (2023) are implemented using a dimension reduction technique in 2D and a Gaussian approximation to embed the conditional distributions.

## 6. Conclusion

This work provides the first theoretical framework and practical implementation of gradient flows of a suitable MMD objective over the space of random measures, endowed with the Wasserstein over Wasserstein distance. On the theoretical side, we provided a rigorous differential structure on that space and showed that these flows are well-posed. On the numerical side, our results demonstrate that this novel approach provides meaningful dynamics for interpolating between random measures. There are many possible extensions of our study. For instance, it would be interesting to investigate the minimization of alternative functionals over the space of random measures, *e.g.*, MMD with other kernels (Bachoc et al., 2023; Kachaiev & Recanatesi, 2024), integral probability metrics (Müller, 1997; Catalano & Lavenant, 2024) or f-divergences (Csiszár, 1967). Future work could also address the theoretical treatment of non-compact manifolds or derive a continuity equation for Wasserstein over Wasserstein (WoW) gradient flows. Then, another topic of future research would be to provide quantitative guarantees on the convergence of these schemes.

## Acknowledgements

We thank the anonymous reviewers for their valuable comments. We also thank Quentin Mérigot for fruitful discussions, and David Alvarez-Melis for his help on the transfer learning experiment. This work was granted access to the HPC resources of IDRIS under the allocation 2024-AD011015891 made by GENCI. CB and AK acknowledge the support of the Agence nationale de la recherche, through the PEPR PDE-AI project (ANR-23-PEIA-0004) and also thank Apple for their academic support through a research funding. CV acknowledges the support of Région Île-de-France through the DIM AI4IDF project.

## Impact Statement

This paper presents work whose goal is to advance the field of Machine Learning. There are many potential societal consequences of our work, none which we feel must be specifically highlighted here.

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

## A. Background on Optimal Transport

### A.1. Optimal Transport on $\mathcal{P}_2(\mathcal{M})$

Let $\mathcal{M}$ be a Riemannian manifold, and denote by $d : \mathcal{M} \times \mathcal{M} \to \mathbb{R}_+$ the associated geodesic distance. We recall that for any $\mu, \nu \in \mathcal{P}_2(\mathcal{M})$, the Wasserstein distance is defined as

$$W_2^2(\mu, \nu) = \inf_{\tilde{\gamma} \in \Pi(\mu, \nu)} \int d(x, y)^2 \, \mathrm{d}\tilde{\gamma}(x, y). \tag{24}$$

Let $\varphi : \mathcal{M} \to \mathbb{R}$. For a cost $c : \mathcal{M} \times \mathcal{M} \to \mathbb{R}$, we recall that its $c$-transform is defined as $\varphi^c(y) = \inf_{x \in \mathcal{M}} c(x, y) - \varphi(x)$. $\varphi$ is said to be $c$-concave if there exists $\phi : \mathcal{M} \to \mathbb{R}$ such that $\varphi = \phi^c$. Here, we focus on $c(x, y) = \frac{1}{2}d(x, y)^2$. Then, the Wasserstein distance can be written through its dual (see *e.g.* (Villani et al., 2009, Theorem 5.10)) as

$$W_2^2(\mu, \nu) = \sup_{f \in L^1(\mu)} \int f \mathrm{d}\mu + \int f^c \mathrm{d}\nu, \tag{25}$$

with $L^1(\mu) = \{f : \mathcal{M} \to \mathbb{R}, \ \int |f| \mathrm{d}\mu < \infty\}$. The optimal $f$ is called the Kantorovich potential, is noted $\varphi_{\mu,\nu}$, and is a $c$-concave map.

We now recall McCann's theorem, which provides a sufficient condition for the existence of an OT map provided that $\mu$ is absolutely continuous with respect to the volume measure. We state the result for a connected compact Riemannian manifold. But, note that this result was then extended to other manifolds, see *e.g.* (Figalli, 2007, Proposition 3.1) for a similar result on complete Riemannian manifolds.

**Theorem A.1** (Theorem 9 in (McCann, 2001)). *Let $\mathcal{M}$ be a connected compact Riemannian manifold. Let $\mu \in \mathcal{P}_{2,\mathrm{ac}}(\mathcal{M})$ and $\nu \in \mathcal{P}_2(\mathcal{M})$. Then, the optimal coupling $\tilde{\gamma} \in \Pi(\mu, \nu)$ is unique and of the form $\tilde{\gamma} = (\mathrm{Id}, \mathrm{T})_{\#}\mu$ with $\mathrm{T}(x) = \exp_x\big(-\nabla\varphi_{\mu,\nu}(x)\big)$ for all $x \in \mathcal{M}$, where $\varphi_{\mu,\nu}$ is a Kantorovich potential for the pair $\mu, \nu$.*

For any $x \in \mathcal{M}$, recall that the exponential map $\exp : T\mathcal{M} \to \mathcal{M}$ maps tangent vectors $v \in T_x\mathcal{M}$ back to the manifold at the point reached at time $t = 1$ by the geodesic starting at $x$ with initial velocity $v$. Moreover, when it is well defined, its inverse is the logarithm map $\log_x : \mathcal{M} \to \mathcal{M}$, which satisfies, for any $x \in \mathcal{M}$, $y = \exp_x(v)$ where $v \in T_x\mathcal{M}$, $\log_x(y) = v$. However, the exponential map is not always invertible. For instance, on the sphere $S^{d-1}$, there are an infinite number of geodesics, and thus of directions $v \in T_x\mathcal{M}$, between $x \in \mathcal{M}$ and its antipodal point $-x$ (see Figure 4). Therefore, the logarithm map $\log_x(-x)$ is multivalued.

Let $\mu, \nu \in \mathcal{P}_2(\mathcal{M})$. When the exponential map is invertible at $\mu$-almost every $x \in \mathcal{M}$, then a (constant-speed) geodesic between $\mu$ and $\nu$ can be defined for all $t \in [0, 1]$ as $\mu_t = \exp_{\pi^1} \circ (t \log_{\pi^1} \circ \pi^2)_{\#}\tilde{\gamma}$ where $\tilde{\gamma} \in \Pi_o(\mu, \nu)$, *i.e.* it satisfies $W_2(\mu_s, \mu_t) = |t - s| W_2(\mu, \nu)$ for all $s, t \in [0, 1]$. However, the exponential map might not always be invertible, as described in the last paragraph. One way to circumvent this problem is to consider the space

$$\exp_\mu^{-1}(\nu) = \{\gamma \in \mathcal{P}_2(T\mathcal{M}), \ \pi_{\#}^{\mathcal{M}}\gamma = \mu, \ \exp_{\#}\gamma = \nu, \ \int \|v\|_x^2 \, \mathrm{d}\gamma(x, v) = W_2^2(\mu, \nu)\}. \tag{26}$$

This space carries more information than the set of optimal couplings as it precises which geodesic was chosen to move the mass from $\mu$ to $\nu$ (Gigli, 2011). Indeed, regarding the previous example on the sphere, for $x \in S^{d-1}$, $\mu = \delta_x$ and $\nu = \delta_{-x}$, and any $v \in T_x\mathcal{M}$ such that $-x = \exp_x(v)$, we have $\delta_{(x,v)} \in \exp_\mu^{-1}(\nu)$, while the optimal coupling would simply be given by the map $T(x) = -x$. Moreover, it allows to define geodesics as $t \mapsto \mu_t = \exp_{\pi^{\mathcal{M}}} \circ (t\pi^{\mathrm{v}})_{\#}\gamma$ for any $\gamma \in \exp_\mu^{-1}(\nu)$ (Gigli, 2011, Theorem 1.11). By Proposition 2.1, if $\mu \in \mathcal{P}_{2,\mathrm{ac}}(\mathcal{M})$, then there exists a unique $\gamma \in \exp_\mu^{-1}(\nu)$, which is of the form $\gamma = (\mathrm{Id}, -\nabla\varphi_{\mu,\nu})_{\#}\mu$. In this case, the geodesic between $\mu$ and $\nu$ is of the form $\mu_t = \exp_{\mathrm{Id}} \circ (-t\nabla\varphi_{\mu,\nu})_{\#}\mu$.

### A.2. Wasserstein Gradient Flows on $\mathcal{P}_2(\mathcal{M})$

We provide in this Section some background on Wasserstein gradient flows on $\mathcal{P}_2(\mathcal{M})$, with $\mathcal{M}$ a Riemannian manifold with geodesic distance $d$. This presentations follows the one from (Lanzetti et al., 2025) on $\mathcal{P}_2(\mathbb{R}^d)$, adapted to $\mathcal{P}_2(\mathcal{M})$ using results of (Erbar, 2010).

**Differential Structure.** First, we recall sub- and super differentiability on this space.

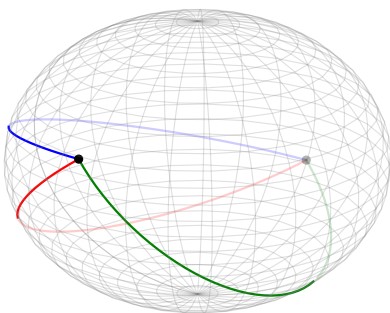

Figure 4: On the sphere, there are an infinite number of geodesics between $x$ and $-x$ (here 3 are represented). Thus, the logarithm map would be multivalued.

**Definition A.2** (Wasserstein sub- and super-differentiability). Let $\mathcal{F} : \mathcal{P}_2(\mathcal{M}) \to \mathbb{R}$ a lower semi-continuous functional. A map $\xi : \mathcal{M} \to T\mathcal{M} \in L^2(\mu, T\mathcal{M})$ belongs to the subdifferential $\partial^- \mathcal{F}(\mu)$ of $\mathcal{F}$ at $\mu$ if for all $\nu \in \mathcal{P}_2(\mathcal{M})$,

$$\mathcal{F}(\nu) \geq \mathcal{F}(\mu) + \sup_{\gamma \in \exp_\mu^{-1}(\nu)} \int \langle \xi(x), v \rangle_x \, \mathrm{d}\gamma(x, v) + o\big(W_2(\mu, \nu)\big). \tag{27}$$

Similarly, $\xi \in L^2(\mu, T\mathcal{M})$ belongs to the superdifferential $\partial^+ \mathcal{F}(\mu)$ of $\mathcal{F}$ at $\mu$ if $-\xi \in \partial^-(-\mathcal{F})(\mu)$.

Similarly as on $\mathcal{P}_2(\mathbb{R}^d)$ (Bonnet, 2019; Lanzetti et al., 2025), we say that a functional is Wasserstein differentiable if it admits sub- and super-differentials which coincide.

**Definition A.3** (Wasserstein differentiability). A functional $\mathcal{F} : \mathcal{P}_2(\mathcal{M}) \to \mathbb{R}$ is Wasserstein differentiable at $\mu \in \mathcal{P}_2(\mathcal{M})$ if $\partial^- \mathcal{F}(\mu) \cap \partial^+ \mathcal{F}(\mu) \neq \emptyset$. In this case, we say that $\nabla_{W_2} \mathcal{F}(\mu) \in \partial^- \mathcal{F}(\mu) \cap \partial^+ \mathcal{F}(\mu)$ is a Wasserstein gradient of $\mathcal{F}$ at $\mu$, and it satisfies for any $\nu \in \mathcal{P}_2(\mathcal{M})$, $\gamma \in \exp_\mu^{-1}(\nu)$,

$$\mathcal{F}(\nu) = \mathcal{F}(\mu) + \int \langle \nabla_{W_2} \mathcal{F}(\mu)(x), v \rangle_x \, \mathrm{d}\gamma(x, v) + o\big(W_2(\mu, \nu)\big). \tag{28}$$

If $\mu \in \mathcal{P}_{2,\mathrm{ac}}(\mathcal{M})$, then by Proposition 2.1, $\gamma \in \exp_\mu^{-1}(\nu)$ is unique and of the form $\gamma = (\mathrm{Id}, -\nabla\varphi_{\mu,\nu})_{\#}\mu$. Thus, in that case, (28) translates as

$$\mathcal{F}(\nu) = \mathcal{F}(\mu) + \int \langle \nabla_{W_2} \mathcal{F}(\mu)(x), -\nabla\varphi_{\mu,\nu}(x) \rangle_x \, \mathrm{d}\mu(x) + o\big(W_2(\mu, \nu)\big), \tag{29}$$

which coincides with (Erbar, 2010, Definition 3.1) (for the subdifferential, and up to a sign as they use $c$-convex maps, and we use $\varphi_{\mu,\nu}$ a $c$-concave map).

If we take $t \mapsto \mu_t = (\exp_{\pi^{\mathcal{M}}} \circ(t\pi^{\mathrm{v}}))_{\#}\gamma$, for $\gamma \in \exp_\mu^{-1}(\nu)$, a geodesic between $\mu, \nu$, then necessarily $(\pi^{\mathcal{M}}, t\pi^{\mathrm{v}})_{\#}\gamma \in \exp_\mu^{-1}(\mu_t)$ (Gigli, 2011, Theorem 1.11), and thus

$$\mathcal{F}(\mu_t) = \mathcal{F}(\mu) + t \int \langle \nabla_{W_2} \mathcal{F}(\mu)(x), v \rangle_x \, \mathrm{d}\gamma(x, v) + o\big(W_2(\mu, \mu_t)\big), \tag{30}$$

which implies $\frac{\mathrm{d}}{\mathrm{d}t} \mathcal{F}(\mu_t)\big|_{t=0} = \int \langle \nabla_{W_2} \mathcal{F}(\mu)(x), v \rangle_x \, \mathrm{d}\gamma(x, v)$.

A priori, the Wasserstein gradient is not unique. Nevertheless, we can always restrict ourselves to a unique gradient belonging to a tangent space whenever it is an Hilbert space. This is the case for $\mu$ absolutely continuous (Gigli, 2011, Corollary 6.6). So, we now focus on $\mathcal{P}_{2,\mathrm{ac}}(\mathcal{M}) \subset \mathcal{P}_2(\mathcal{M})$. In this case, the tangent space can be defined as $T_\mu \mathcal{P}_2(\mathcal{M}) = \overline{\{\nabla\varphi, \ \varphi \in C_c^\infty(\mathcal{M})\}}^{L^2(\mu, T\mathcal{M})}$. This is a closed linear subspace of $L^2(\mu, T\mathcal{M})$ and we can uniquely decompose any $\xi \in L^2(\mu, T\mathcal{M})$ as $\xi = \phi + \psi$ with $\phi \in T_\mu \mathcal{P}_2(\mathcal{M})$ and $\psi \in T_\mu \mathcal{P}_2(\mathcal{M})^\perp$ (Rudin, 1986, Theorem 4.11). Since $\mu \in \mathcal{P}_{2,\mathrm{ac}}(\mathcal{M})$, then by Proposition 2.1, the optimal $\gamma$ is equal to $(\mathrm{Id}, -\nabla\varphi_{\mu,\nu})_{\#}\mu$ with $\varphi_{\mu,\nu}$ a Kantorovich potential between $\mu$ and $\nu$. In this case, it can be shown that

$$\int \langle \psi(x), v \rangle_x \, \mathrm{d}\gamma(x, v) = \int \langle \psi(x), -\nabla\varphi_{\mu,\nu}(x) \rangle_x \, \mathrm{d}\mu(x) = 0, \tag{31}$$

since $\nabla \varphi_{\mu,\nu} \in T_\mu \mathcal{P}_2(\mathcal{M})$ (Erbar, 2010, Lemma 2.6). Thus the only part of the gradient that matters is $\phi$, and we can show that it is unique.

**Proposition A.4.** *Let $\mathcal{F} : \mathcal{P}_2(\mathcal{M}) \to \mathbb{R}$. Its gradient at $\mu \in \mathcal{P}_{2,\mathrm{ac}}(\mathcal{M})$, if it is exists, is the unique element of $T_\mu \mathcal{P}_2(\mathcal{M}) \cap \partial^+ \mathcal{F}(\mu) \cap \partial^- \mathcal{F}(\mu)$.*

*Proof.* See Appendix C.7. □

Another interesting property of gradients belonging to the tangent space is that they are actually strong differentials, meaning that they satisfy the Taylor expansion along any coupling, *i.e.*, for any $\nu \in \mathcal{P}_2(\mathcal{M})$ and $\gamma \in \mathcal{P}_2(T\mathcal{M})$ such that $\pi^{\mathcal{M}}_{\#} \gamma = \mu$ and $\exp_{\#} \gamma = \nu$, $\nabla_{\mathrm{W}_2} \mathcal{F}(\mu) \in T_\mu \mathcal{P}_2(\mathcal{M})$ satisfies

$$\mathcal{F}(\nu) = \mathcal{F}(\mu) + \int \langle \nabla_{\mathrm{W}_2} \mathcal{F}(\mu)(x), v \rangle_x \, \mathrm{d}\gamma(x, v) + o\left( \sqrt{\int \|v\|_x^2 \, \mathrm{d}\gamma(x,v)} \right). \tag{32}$$

Erbar (2010, Lemma 3.2) showed this property for couplings obtained through maps. We extend it for any coupling in the next Proposition. First, for $\mu \in \mathcal{P}_2(\mathcal{M})$ fixed, we define $\mathcal{P}_2(T\mathcal{M})_\mu := \{\gamma \in \mathcal{P}_2(T\mathcal{M}) \mid \pi^{\mathcal{M}}_{\#} \gamma = \mu\}$. For every $\gamma \in \mathcal{P}_2(T\mathcal{M})_\mu$, we define $\|\gamma\|_\mu^2 := \int \|v\|_x^2 \mathrm{d}\gamma(x,v)$, and we further define its barycentric projection to be the unique vector field $\mathcal{B}(\gamma) \in L^2(\mu, T\mathcal{M})$ such that for every $\xi \in L^2(\mu, T\mathcal{M})$,

$$\int \langle \xi(x), v \rangle_x \, \mathrm{d}\gamma(x,v) = \int \langle \xi(x), \mathcal{B}(\gamma)(x) \rangle_x \mathrm{d}\mu(x) = \langle \xi, \mathcal{B}(\gamma) \rangle_{L^2(\mu)}, \tag{33}$$

(see (Gigli, 2011, Chapter 6)). Note that the barycentric projection satisfies $\|\mathcal{B}(\gamma)\|_{L^2(\mu)} \leq \|\gamma\|_\mu$.

**Proposition A.5.** *Let $\xi \in \partial^- \mathcal{F}(\mu) \cap T_\mu \mathcal{P}_2(\mathcal{M})$. Then $\xi$ is an (extended) strong subdifferential of $\mathcal{F}$ at $\mu$, i.e. for every $\gamma \in \mathcal{P}_2(T\mathcal{M})_\mu$,*

$$\mathcal{F}(\exp_{\#} \gamma) \geq \mathcal{F}(\mu) + \int \langle \xi(x), v \rangle_x \, \mathrm{d}\gamma(x,v) + o(\|\gamma\|_\mu). \tag{34}$$

*By symmetry of the arguments, it also holds for superdifferentials and gradients.*

*Proof.* See Appendix C.8. □

We now derive the Wasserstein gradients of well known functionals such as potential energies and interaction energies.

**Proposition A.6.** *Let $V : \mathcal{M} \to \mathbb{R}$ be twice differentiable with Hessian bounded in operator norm by $L$ for all $x \in \mathcal{M}$, i.e. $\|\mathrm{Hess}_{\mathcal{M}} V(x)\| = \max_{v \in T_x \mathcal{M}, \|v\|_x = 1} \|\mathrm{Hess}_{\mathcal{M}} V(x)[v]\|_x \leq L$, and $\mathcal{V} : \mu \mapsto \int V \mathrm{d}\mu$. Then $\mathcal{V}$ is differentiable with gradient $\nabla_{\mathrm{W}_2} \mathcal{V}(\mu) = \nabla_{\mathcal{M}} V$ for any $\mu \in \mathcal{P}_2(\mathcal{M})$.*

*Proof.* See Appendix C.9. □

**Proposition A.7.** *Let $W : \mathcal{M} \times \mathcal{M} \to \mathbb{R}$ be twice differentiable with Hessian for both arguments bounded in operator norm, and $\mathcal{W} : \mu \mapsto \iint W(x,y)\mathrm{d}\mu(x)\mathrm{d}\mu(y)$. Then $\mathcal{W}$ is differentiable with gradient $\nabla_{\mathrm{W}_2} \mathcal{W}(\mu)(x) = \int \left( \nabla_1 W(x,y) + \nabla_2 W(y,x) \right) \mathrm{d}\mu(y)$ for any $\mu \in \mathcal{P}_2(\mathcal{M})$, $x \in \mathcal{M}$.*

*Proof.* See Appendix C.10. □

We also introduce the notion of Hessian on the Wasserstein space, which will be useful to derive smoothness assumptions for the WoW gradients to be well defined.

**Definition A.8.** Let $\mathcal{F} : \mathcal{P}_2(\mathcal{M}) \to \mathbb{R}$. Let $\mu \in \mathcal{P}_2(\mathcal{M})$. The Wasserstein Hessian of $\mathcal{F}$ at $\gamma \in \exp_\mu^{-1}(\nu)$ for some $\nu \in \mathcal{P}_2(\mathcal{M})$, is a map $\mathrm{H}\mathcal{F}_\gamma : T\mathcal{M} \to T\mathcal{M}$ verifying $\frac{\mathrm{d}^2}{\mathrm{d}t^2} \mathcal{F}(\mu_t)\big|_{t=0} = \int \langle \mathrm{H}\mathcal{F}_\gamma(x,v), v \rangle_x \, \mathrm{d}\gamma(x,v)$ for a constant-speed geodesic $\mu_t = \left( \exp_{\pi^{\mathcal{M}}} \circ(t\pi^{\mathrm{v}}) \right)_{\#} \gamma$ with $\gamma \in \exp_\mu^{-1}(\nu)$.

**Wasserstein Gradient Flows.** A Wasserstein gradient flow of $\mathcal{F} : \mathcal{P}_2(\mathcal{M}) \to \mathbb{R}$ is defined as a curve $t \mapsto \mu_t$ on an interval $I$, which is a weak solution of the continuity equation

$$\partial_t \mu_t = \text{div}\big(\mu_t \nabla_{W_2} \mathcal{F}(\mu_t)\big), \tag{35}$$

*i.e.*, which satisfies for any $\varphi \in C_c^\infty(I \times \mathcal{M})$,

$$\int_I \int_{\mathcal{M}} \big(\partial_t \varphi_t(x) - \langle \nabla_{\mathcal{M}} \varphi_t(x), \nabla_{W_2} \mathcal{F}(\mu_t)(x) \rangle_x \big) \, \mathrm{d}\mu_t(x) \mathrm{d}t = 0. \tag{36}$$

Usually, such equation needs to be approximated by a scheme discretized in time. A common way to do it is through the Jordan-Kinderlehrer-Otto (JKO) scheme introduced in (Jordan et al., 1998), which is of the form

$$\forall k \geq 0, \ \mu_{k+1} \in \underset{\mu \in \mathcal{P}_2(\mathcal{M})}{\text{argmin}} \ \frac{W_2^2(\mu, \mu_k)}{2\tau} + \mathcal{F}(\mu). \tag{37}$$

Under suitable conditions, this scheme converges towards the Wasserstein gradient flow of $\mathcal{F}$ (Ambrosio et al., 2008; Erbar, 2010). However, this scheme is generally costly to compute, as it requires to solve an optimization problem at each iteration. In practice, it is more convenient to rely on an explicit discretization, which can be seen as a Riemannian Wasserstein gradient descent (Bonnabel, 2013; Bonet et al., 2025), which is of the form

$$\forall k \geq 0, \ \mu_{k+1} = \exp_{\mu_k}\big(-\tau \nabla_{W_2} \mathcal{F}(\mu_k)\big). \tag{38}$$

We note that this scheme can be obtained by linearizing the objective in (37). Indeed, if $\mu_k \in \mathcal{P}_{2,\text{ac}}(\mathcal{M})$, (37) can be written as

$$\begin{cases} T_{k+1} = \text{argmin}_{T \in L^2(\mu_k, T\mathcal{M})} \ \frac{1}{2\tau} \int \|T(x)\|_x^2 \, \mathrm{d}\mu_k(x) + \mathcal{F}\big((\exp \circ T)_{\#} \mu_k\big) \\ \mu_{k+1} = (\exp \circ T_{k+1})_{\#} \mu_k. \end{cases} \tag{39}$$

Using the coupling $\gamma = (\text{Id}, \exp \circ T)_{\#} \mu_k \in \Pi(\mu_k, (\exp \circ T)_{\#} \mu_k)$ and that $\nabla_{W_2} \mathcal{F}(\mu_k)$ is a strong differential, then we have that

$$\mathcal{F}\big((\exp \circ T)_{\#} \mu_k\big) = \mathcal{F}(\mu_k) + \int \langle \nabla_{W_2} \mathcal{F}(\mu_k)(x), T(x) \rangle_x \, \mathrm{d}\mu_k(x) + o\left(\sqrt{\int \|T(x)\|_x^2 \, \mathrm{d}\mu_k(x)}\right). \tag{40}$$

Plugging this linearization in (39), we obtain

$$T_{k+1} \in \underset{T \in L^2(\mu_k, T\mathcal{M})}{\text{argmin}} \ \frac{1}{2\tau} \|T\|_{L^2(\mu_k, T\mathcal{M})}^2 + \langle \nabla_{W_2} \mathcal{F}(\mu_k), T \rangle_{L^2(\mu_k, T\mathcal{M})}. \tag{41}$$

Taking the first order condition, we recover (38) as $T_{k+1} = -\tau \nabla_{W_2} \mathcal{F}(\mu_k)$.

**Wasserstein Gradient Descent.** In practice, we usually work with particles, *i.e.* we start at $\mu_0 = \frac{1}{n} \sum_{i=1}^n \delta_{x_{i,0}}$, and update each particle at each iteration $k \geq 0$ as

$$\forall i \in \{1, \ldots, n\}, \ x_{i,k+1} = \exp_{x_{i,k}}\big(-\tau \nabla_{W_2} \mathcal{F}(\mu_k)(x_{i,k})\big) \tag{42}$$

for $\mu_k = \frac{1}{n} \sum_{i=1}^n \delta_{x_{i,k}}$. In particular, for $\mathcal{M} = \mathbb{R}^d$, the scheme is obtained as

$$\forall i \in \{1, \ldots, n\}, \ x_{i,k+1} = x_{i,k} - \tau \nabla_{W_2} \mathcal{F}(\mu_k)(x_{i,k}). \tag{43}$$

Moreover, if the functional $\mathcal{F} : \mathcal{P}_2(\mathbb{R}^d) \to \mathbb{R}$ has a closed-form over discrete measures, *i.e.*, there exists $F : (\mathbb{R}^d)^n \to \mathbb{R}$ such that $\mathcal{F}\big(\frac{1}{n} \sum_{i=1}^n \delta_{x_i}\big) = F(x_1, \ldots, x_n)$, then we can use backpropagation on $F$ and use that $\nabla_{W_2} \mathcal{F}\big(\frac{1}{n} \sum_{i=1}^n \delta_{x_{i,k}}\big)(x_{i,k}) = n \nabla_i F(x_1, \ldots, x_n)$.

**Proposition A.9.** *Let $\mathcal{F} : \mathcal{P}_2(\mathbb{R}^d) \to \mathbb{R}$ a Wasserstein differentiable functional, and $F : (\mathbb{R}^d)^n \to \mathbb{R}$ such that for any $\mathbf{x} = (x_1, \ldots, x_n) \notin \Delta_n := \{\mathbf{x} \in (\mathbb{R}^d)^n \,|\, \exists i \neq j, x_i = x_j\}$ and $\mu_n = \frac{1}{n} \sum_{i=1}^n \delta_{x_i}$, $\mathcal{F}(\mu_n) = F(x_1, \ldots, x_n)$. Then, for all $i \in \{1, \ldots, n\}$,*

$$\nabla_{W_2} \mathcal{F}(\mu_n)(x_i) = n \nabla_i F(x_1, \ldots, x_n). \tag{44}$$

*Proof.* See Appendix C.11. □

# B. Wasserstein over Wasserstein Space

## B.1. Function Spaces on $\mathcal{P}_2(M)$

In this section we fix $\mathbb{P} \in \mathcal{P}_2(\mathcal{P}_2(\mathcal{M}))$. Recall that we define the tangent space to $\mathcal{P}_2(\mathcal{M})$ at $\mu$ by $T_\mu \mathcal{P}_2(\mathcal{M}) := \overline{\{\nabla \varphi, \ \varphi \in C_c^\infty(M)\}}^{L^2(\mu, T\mathcal{M})}$ for every $\mu \in \mathcal{P}_2(\mathcal{M})$. We also define the larger tangent space $T^{\mathrm{Der}} \mathcal{P}_2(\mathcal{M})$ by $T_\mu^{\mathrm{Der}} \mathcal{P}_2(\mathcal{M}) := \overline{\Gamma(\mathcal{M}, T\mathcal{M})}^{L^2(\mu, T\mathcal{M})}$ where $\Gamma(\mathcal{M}, T\mathcal{M})$ is the space of smooth vector fields on $\mathcal{M}$, *i.e.* smooth maps from $\mathcal{M}$ to $T\mathcal{M}$. Our goal is to rigorously define the space $L^2(\mathbb{P}, T\mathcal{P}_2(\mathcal{M}))$ and to show that it is indeed a Hilbert space.

Let $B \subseteq \mathcal{M}$ open, then the map $\mu \in \mathcal{P}_2(\mathcal{M}) \mapsto \mu(B)$ is Borel, indeed, it is lower semicontinuous by (Ambrosio et al., 2008, Equation (5.1.16)). Thus, the map $\mu \in X \mapsto \mu \in \mathcal{P}_2(Y)$ with $X = \mathcal{P}_2(\mathcal{M})$ and $Y = \mathcal{M}$ is a Borel map (in the sense of measure-valued maps), and the formula

$$\tilde{\mathbb{P}}(f) = \int_{\mathcal{P}_2(\mathcal{M})} \int_{\mathcal{M}} f(\mu, x) \mathrm{d}\mu(x) \mathrm{d}\mathbb{P}(\mu) \tag{45}$$

defines a probability measure $\tilde{\mathbb{P}}$ on $\mathcal{P}_2(\mathcal{M}) \times \mathcal{M}$ (we follow the same reasoning as in (Ambrosio et al., 2008, Section 5.3)).

We then define $L^2(\mathbb{P}, T\mathcal{M})$ to be the quotient of the space of measurable functions $f : \mathcal{P}_2(\mathcal{M}) \times \mathcal{M} \to T\mathcal{M}$, such that $f(\mu, x) \in T_x \mathcal{M}$ for every $(\mu, x) \in \mathcal{P}_2(\mathcal{M}) \times \mathcal{M}$, by the equivalence relation corresponding to equality $\tilde{\mathbb{P}}$-almost everywhere, and we equip it with the norm $\| \cdot \|_{L^2(\mathbb{P})}$ defined by

$$\|f\|_{L^2(\mathbb{P})}^2 := \int_{\mathcal{P}_2(\mathcal{M})} \|f(\mu)\|_{L^2(\mu)}^2 \mathrm{d}\mathbb{P}(\mu) \tag{46}$$

(we view $f \in L^2(\mathbb{P}, T\mathcal{M})$ interchangeably as a function with signatures $\mathcal{P}_2(\mathcal{M}) \times \mathcal{M} \to T\mathcal{M}$ and $\mathcal{P}_2(\mathcal{M}) \to (\mathcal{M} \to T\mathcal{M})$, hence the notation $f(\mu)$). It is a Hilbert space: indeed, if $\mathcal{M}$ is an open set $U \subseteq \mathbb{R}^n$, then since $TU = U \times \mathbb{R}^n$, $L^2(\mathbb{P}, TU)$ is a Hilbert space as it is the direct sum of $n$ copies of the Hilbert space $L^2(\tilde{\mathbb{P}})$, and in the general case, we can show that $L^2(\mathbb{P}, T\mathcal{M})$ is complete by showing that Cauchy sequences converge, by using local charts and a partition of unity of $\mathcal{M}$ to fall back on the case where $\mathcal{M}$ is an open of $\mathbb{R}^n$.

We can now define $L^2(\mathbb{P}, T\mathcal{P}_2(\mathcal{M}))$ as the space of functions $f \in L^2(\mathbb{P}, T\mathcal{M})$ such that $f(\mu) \in T_\mu \mathcal{P}_2(\mathcal{M})$ for $\mathbb{P}$-ae $\mu$. It is closed in $L^2(\mathbb{P}, T\mathcal{M})$ and is therefore a Hilbert space. Indeed, if $\{f_n\}_{n=1}^\infty \subseteq L^2(\mathbb{P}, T\mathcal{P}_2(\mathcal{M}))$ converges to $f \in L^2(\mathbb{P}, T\mathcal{M})$,

$$\lim_{n \to \infty} \int \|f_n(\mu) - f(\mu)\|_{L^2(\mu)}^2 \mathrm{d}\mathbb{P}(\mu) = 0. \tag{47}$$

This implies that, up to extracting a subsequence, we have $\|f_n(\mu) - f(\mu)\|_{L^2(\mu)} \to 0$ for $\mathbb{P}$-ae $\mu$[2]. But since the $T_\mu \mathcal{P}_2(\mathcal{M})$ are Hilbert spaces, this implies that $f(\mu) \in T_\mu \mathcal{P}_2(\mathcal{M})$ for $\mathbb{P}$-ae $\mu$, and $f$ indeed belongs to $L^2(\mathbb{P}, T\mathcal{P}_2(\mathcal{M}))$. We define similarly $L^2(\mathbb{P}, T^{\mathrm{Der}}\mathcal{P}_2(\mathcal{M}))$ and show that it is a Hilbert space. This latter space $T^{\mathrm{Der}}\mathcal{P}_2(\mathcal{M})$ is useful in that it allows us to define a notion of differential for $W_2$-Lipschitz functions on $\mathcal{P}_2(\mathcal{M})$, as we will see in the next subsection.

## B.2. Lipschitz Functions and Rademacher Property

For every smooth vector field $w \in \Gamma(\mathcal{M}, T\mathcal{M})$, let $(\psi^{w,t})_{t \in \mathbb{R}}$ be its flow on $\mathcal{M}$, that is, the diffeomorphic flow solution of

$$\begin{cases} \forall (t, x) \in \mathbb{R} \times \mathcal{M}, \ \frac{\mathrm{d}}{\mathrm{d}t} \psi^{w,t}(x) = w(\psi^{w,t}(x)) \\ \psi_0 = \mathrm{Id}, \end{cases} \tag{48}$$

---

[2]We recall the argument. For every $\varepsilon > 0$, we have $\mathbb{P}[\|f(\mu) - f_n(\mu)\|_{L^2(\mu)} \geq \varepsilon] \to 0$ as $\mathbb{P}[\|f(\mu) - f_n(\mu)\|_{L^2(\mu)} \geq \varepsilon] \leq \frac{1}{\varepsilon^2} \int \|f(\mu) - f_n(\mu)\|_{L^2(\mu)}^2 \mathrm{d}\mathbb{P}(\mu)$. So, up to extracting a subsequence, we may assume that for every $n$, $\mathbb{P}[\|f(\mu) - f_n(\mu)\|_{L^2(\mu)} \geq n^{-1}] \leq \frac{1}{n^2}$. Then, we can check that the set $A = \bigcap_N \bigcup_{n \geq N} \{\|f(\mu) - f_n(\mu)\|_{L^2(\mu)} \geq n^{-1}\}$ has null $\mathbb{P}$-measure and that for any $\mu \notin A$, $f_n(\mu) \to f(\mu)$ in $L^2(\mu)$.

and denote $\Psi^{w,t}$ the map $\mathcal{P}_2(\mathcal{M}) \mapsto \mathcal{P}_2(\mathcal{M})$ induced by the pushforward by $\psi^{w,t}$.

The following definition is taken from (Emami & Pass, 2025, Definition 9).

**Definition B.1.** (Emami and Pass, 2024) We say that a measure $\mathbb{P}_0 \in \mathcal{P}_2(\mathcal{P}_2(\mathcal{M}))$ satisfies the Rademacher property if for every $W_2$-Lipschitz function $U : \mathcal{P}_2(\mathcal{M}) \mapsto \mathbb{R}$, there exists $D_{\mathbb{P}_0} U \in L^2(\mathbb{P}_0, T^{\mathrm{Der}} \mathcal{P}_2(T\mathcal{M}))$ such that for every $w \in \Gamma(\mathcal{M}, T\mathcal{M})$,

$$\lim_{t \to 0} \frac{U(\Psi^{w,t}(\cdot)) - U(\cdot)}{t} = \langle D_{\mathbb{P}_0} U(\cdot), w \rangle_{L^2(\cdot)} \text{ in } L^2(\mathbb{P}_0). \tag{49}$$

Thus, every time we have a reference measure $\mathbb{P}_0 \in \mathcal{P}_2(\mathcal{P}_2(\mathcal{M}))$ satisfying the Rademacher property, we can define for every $W_2$-Lipschitz function $U$ a measurable section $D_{\mathbb{P}_0} U$ of $T^{\mathrm{Der}} \mathcal{P}_2(\mathcal{M})$ that acts as a "differential" of sorts, for perturbations given by smooth vector fields on $\mathcal{M}$.

**B.3. Wasserstein Geometry of $\mathcal{P}_2(\mathcal{P}_2(M))$**

We recall that the WoW distance between $\mathbb{P}, \mathbb{Q} \in \mathcal{P}_2(\mathcal{P}_2(\mathcal{M}))$ is defined as

$$W_{W_2}(\mathbb{P}, \mathbb{Q})^2 = \inf_{\Gamma \in \Pi(\mathbb{P}, \mathbb{Q})} \int W_2^2(\mu, \nu) \, d\Gamma(\mu, \nu). \tag{50}$$

A natural question is to find the conditions under which this problem admits an OT map. Emami & Pass (2025) showed it is the case for $\mathcal{M}$ a compact connected Riemannian manifold, for absolutely continuous measures *w.r.t* a reference measure $\mathbb{P}_0$ satisfying the following assumption:

**Assumption B.2.**

- $\mathbb{P}_0$ has no atoms

- $\mathbb{P}_0$ satisfies the following integration by parts formula: for any $\mathcal{F}, \mathcal{G} \in \mathrm{Cyl}(\mathcal{P}_2(\mathcal{M}))$, and any smooth vector field $w \in \Gamma(\mathcal{M}, T\mathcal{M})$, there exists a measurable map $\mu \mapsto \nabla_w^* \mathcal{G}(\mu) \in T_\mu^{\mathrm{Der}} \mathcal{P}_2(\mathcal{M})$ such that

$$\int_{\mathcal{P}_2(\mathcal{M})} \langle \nabla_{W_2} \mathcal{F}(\mu), w \rangle_{L^2(\mu)} \cdot \mathcal{G}(\mu) \, d\mathbb{P}_0(\mu) = \int_{\mathcal{P}_2(\mathcal{M})} \mathcal{F}(\mu) \cdot \nabla_w^* \mathcal{G}(\mu) \, d\mathbb{P}_0(\mu). \tag{51}$$

- $\mathbb{P}_0$ is quasi-invariant with respect to the action of the flows generated by smooth vector fields, *i.e.* for any smooth vector field $w \in \Gamma(\mathcal{M}, T\mathcal{M})$, $\mathbb{P}_0$ and $\mathbb{P}_0^{t,w} := \Psi_\#^{w,t} \mathbb{P}_0$ are mutually absolutely continuous for every $t \in \mathbb{R}$, and the Radon-Nikodym derivative

$$R_r^w = \frac{d\mathbb{P}_0^{t,w} \otimes dr}{d\mathbb{P}_0 \otimes dr}, \quad r \in \mathbb{R} \tag{52}$$

  satisfies, for $\mathbb{P}_0$-a.e. $\mu$, $\mathcal{L}^1 - \mathrm{essinf}_{r \in (s,t)} R_r^w(\mu) > 0$ for all $s, t \in \mathbb{R}$ with $s \leq t$.

These assumptions were first proposed by Dello Schiavo (2020), and we refer to (Dello Schiavo, 2020) for examples of measures satisfying them. By (Dello Schiavo, 2020, Theorem 2.10), any $\mathbb{P}_0$ satisfying Assumption B.2 also satisfies the Rademacher property, which Emami & Pass (2025) leveraged to show the existence of an OT map. In all the following, we fix a reference measure $\mathbb{P}_0 \in \mathcal{P}_2(\mathcal{P}_2(\mathcal{M}))$ satisfying Assumption B.2, and when there is no ambiguity, the "differentials" of a $W_2$-Lipschitz function $U$ will be denoted $DU$. Moreover, by (Dello Schiavo, 2020, Theorem 2.10 (2)), if $U \in \mathrm{Cyl}(\mathcal{P}_2(\mathcal{M}))$, then its differential coincides with the usual Wasserstein gradient, *i.e.*, $DU = \nabla_{W_2} U$.

**Theorem B.3** (Theorem 13 in (Emami & Pass, 2025)). *Let $\mathbb{P} \in \mathcal{P}_2(\mathcal{P}_{2,\mathrm{ac}}(\mathcal{M}))$ such that $\mathbb{P} \ll \mathbb{P}_0$ and $\mathbb{Q} \in \mathcal{P}_2(\mathcal{P}_2(\mathcal{M}))$. Then, there is a unique optimal plan $\Gamma$, which is of the form $\Gamma = (\mathrm{Id}, \mathrm{T})_\# \mathbb{P}$ with $\mathrm{T} : \mathcal{P}_2(\mathcal{M}) \to \mathcal{P}_2(\mathcal{M})$ satisfying $\mathrm{T}_\# \mathbb{P} = \mathbb{Q}$. Moreover, $\mathrm{T}$ is of the form $\mathrm{T}(\mu) = \exp(-D_{\mathbb{P}_0} U(\mu))_\# \mu$, where $U$ is a $(\frac{1}{2} W_2^2$-concave) Kantorovich potential for $\mathbb{P}, \mathbb{Q}$. In fact, for $\mathbb{P}$-a.e. $\mu$, $D_{\mathbb{P}_0} U(\mu) = \nabla \varphi_{\mu, \mathrm{T}(\mu)}$, where $\varphi_{\mu, \mathrm{T}(\mu)}$ is a $(\frac{1}{2} d^2$-concave) Kantorovich potential for $\mu, \mathrm{T}(\mu)$.*

Note that the last two statements on the form of $\mathrm{T}$ are not part of the statement of the theorem in (Emami & Pass, 2025), but can be found in its proof.

**Geodesics on** $\mathcal{P}_2\big(\mathcal{P}_2(\mathcal{M})\big)$**.**   We recall that a constant-speed geodesic between $\mathbb{P}, \mathbb{Q} \in \mathcal{P}_2\big(\mathcal{P}_2(\mathcal{M})\big)$ is a curve $t \mapsto \mathbb{P}_t$ defined on $[0, 1]$, which satisfies $\mathbb{P}_0 = \mathbb{P}$, $\mathbb{P}_1 = \mathbb{Q}$ and for all $s, t \in [0, 1]$, $\mathrm{W}_{\mathrm{W}_2}(\mathbb{P}_s, \mathbb{P}_t) = |t - s| \mathrm{W}_{\mathrm{W}_2}(\mathbb{P}, \mathbb{Q})$ (see *e.g.* (Santambrogio, 2015, Box 5.2)).

As we work on manifolds, we introduce similarly as on $\mathcal{P}_2(\mathcal{M})$ a generalized inverse of the exponential map, which allows to characterize geodesics even when the optimal coupling is not unique. The multivalued inverse of the exponential map between $\mathbb{P}, \mathbb{Q} \in \mathcal{P}_2\big(\mathcal{P}_2(\mathcal{M})\big)$ is then

$$\exp_{\mathbb{P}}^{-1}(\mathbb{Q}) = \left\{ \Gamma \in \mathcal{P}_2\big(\mathcal{P}_2(T\mathcal{M})\big), \ \phi_{\#}^{\mathcal{M}} \Gamma = \mathbb{P}, \ \phi_{\#}^{\exp} \Gamma = \mathbb{Q}, \ \iint \|v\|_x^2 \, \mathrm{d}\gamma(x, v) \mathrm{d}\Gamma(\gamma) = \mathrm{W}_{\mathrm{W}_2}(\mathbb{P}, \mathbb{Q})^2 \right\}, \quad (53)$$

where for any $\gamma \in \mathcal{P}_2(T\mathcal{M})$, $\phi^{\mathcal{M}}(\gamma) = \pi_{\#}^{\mathcal{M}} \gamma$ and $\phi^{\exp}(\gamma) = \exp_{\#} \gamma$.

**Proposition B.4.** *Let* $\Gamma \in \exp_{\mathbb{P}}^{-1}(\mathbb{Q})$. *Then the curve* $t \mapsto \mathbb{P}_t = \exp_{\phi^{\mathcal{M}}} \circ (t\phi^{\mathrm{v}})_{\#} \Gamma$ *defines a geodesic between* $\mathbb{P}$ *and* $\mathbb{Q}$.

*Proof.* See Appendix C.12.  □

### B.4. Differentiability on $\mathcal{P}_2\big(\mathcal{P}_2(\mathcal{M})\big)$

We recall that by Section 3.2, if $\mathbb{F}$ is Wasserstein differentiable at $\mathbb{P}$, then the WoW gradient $\nabla_{\mathrm{W}_{\mathrm{W}_2}} \mathbb{F}(\mathbb{P})$ satisfies for any $\Gamma \in \exp_{\mathbb{P}}^{-1}(\mathbb{Q})$,

$$\mathbb{F}(\mathbb{Q}) = \mathbb{F}(\mathbb{P}) + \iint \langle \nabla_{\mathrm{W}_{\mathrm{W}_2}} \mathbb{F}(\mathbb{P})(\pi_{\#}^{\mathcal{M}} \gamma)(x), v \rangle_x \, \mathrm{d}\gamma(x, v) \mathrm{d}\Gamma(\gamma) + o\big(\mathrm{W}_{\mathrm{W}_2}(\mathbb{P}, \mathbb{Q})\big). \quad (54)$$

Using this formula, we now derive the gradient of potential and interaction energies.

**Proposition B.5.** *Let* $\mathcal{M}$ *be a compact and connected Riemannian manifold,* $\mathcal{F} : \mathcal{P}_2(\mathcal{M}) \to \mathbb{R}$ *a twice Wasserstein differentiable functional with Hessian bounded in operator norm for all* $\gamma \in \mathcal{P}_2(T\mathcal{M})$, *i.e.* $\sup_{(x,v) \in \mathrm{supp}(\gamma), \, \|v\|_x = 1} \|\mathrm{H}\mathcal{F}_\gamma(x, v)\|_x \leq L$, *and* $\mathbb{F}(\mathbb{P}) = \int \mathcal{F}(\mu) \, \mathrm{d}\mathbb{P}(\mu)$ *for* $\mathbb{P} \in \mathcal{P}_2\big(\mathcal{P}_2(\mathcal{M})\big)$. *Then,* $\mathbb{F}$ *is Wasserstein differentiable, and its gradient is* $\nabla_{\mathrm{W}_{\mathrm{W}_2}} \mathbb{F}(\mathbb{P}) = \nabla_{\mathrm{W}_2} \mathcal{F}$.

*Proof.* See Appendix C.13.  □

**Proposition B.6.** *Let* $\mathcal{M}$ *be a compact and connected Riemannian manifold,* $\mathcal{W} : \mathcal{P}_2(\mathcal{M}) \times \mathcal{P}_2(\mathcal{M}) \to \mathbb{R}$ *be Wasserstein differentiable with respect to each of its argument and with bounded Hessian in operator norm for all* $\gamma \in \mathcal{P}_2(T\mathcal{M})$ *as in Proposition B.5. Let* $\mathbb{P} \in \mathcal{P}_2\big(\mathcal{P}_2(\mathcal{M})\big)$ *and* $\mathbb{F}(\mathbb{P}) = \iint \mathcal{W}(\mu, \nu) \, \mathrm{d}\mathbb{P}(\mu)\mathrm{d}\mathbb{P}(\nu)$. *Then,* $\nabla_{\mathrm{W}_{\mathrm{W}_2}} \mathbb{F}(\mathbb{P})(\mu) = \int \big(\nabla_{\mathrm{W}_2,1} \mathcal{W}(\mu, \nu) + \nabla_{\mathrm{W}_2,2} \mathcal{W}(\nu, \mu)\big) \, \mathrm{d}\mathbb{P}(\nu)$.

*Proof.* See Appendix C.14.  □

**Relation with first variation.**   The gradients of the potential and interaction energies derived in the last two propositions are computed by showing that they satisfy the definition of the WoW gradients thanks to coupling arguments. Similarly to the case on $\mathcal{P}_2(\mathcal{M})$, we expect that they can also be computed as the gradient of the first variation, which is a much simpler way to compute Wasserstein gradient of generic functionals. Let $\mathbb{F} : \mathcal{P}_{\mathrm{ac}}\big(\mathcal{P}_2(\mathcal{M})\big) \to \mathbb{R}$. Then the first variation $\frac{\delta \mathbb{F}}{\delta \mathbb{P}}(\mathbb{P}) : \mathcal{P}_2(\mathcal{M}) \to \mathbb{R}$ at $\mathbb{P}$ is defined as the unique function (up to a constant) satisfying

$$\lim_{\varepsilon \to 0} \frac{\mathbb{F}(\mathbb{P} + \varepsilon \chi) - \mathbb{F}(\mathbb{P})}{\varepsilon} = \int \frac{\delta \mathbb{F}}{\delta \mathbb{P}}(\mathbb{P}) \, \mathrm{d}\chi, \quad (55)$$

where $\int \mathrm{d}\chi = 0$ and $\mathbb{P} + \varepsilon \chi \in \mathcal{P}_{\mathrm{ac}}\big(\mathcal{P}_2(\mathcal{M})\big)$ for $\varepsilon$ small. Then, we expect that the WoW gradient of $\mathbb{F}$ can be computed as

$$\nabla_{\mathrm{W}_{\mathrm{W}_2}} \mathbb{F}(\mathbb{P}) = \nabla_{\mathrm{W}_2} \frac{\delta \mathbb{F}}{\delta \mathbb{P}}(\mathbb{P}). \quad (56)$$

We leave for future work to show formally this formula. Nonetheless, we verify that it holds for potential and interaction energies. Indeed, for $\mathbb{V}(\mathbb{P}) = \int \mathcal{F}(\mu) \, \mathrm{d}\mathbb{P}(\mu)$, we have $\frac{\delta \mathbb{V}}{\delta \mathbb{P}}(\mathbb{P}) = \mathcal{F}$ since

$$\mathbb{V}(\mathbb{P} + \varepsilon \chi) = \int \mathcal{F}(\mu)\mathrm{d}\mathbb{P}(\mu) + \varepsilon \int \mathcal{F}(\mu)\mathrm{d}\chi(\mu), \quad (57)$$

and thus the WoW gradient derived in Proposition B.5 coincides well with $\nabla_{W_2} \frac{\delta \mathbb{V}}{\delta \mathbb{P}}(\mathbb{P})$. Similarly, for $\mathbb{W}(\mathbb{P}) = \iint \mathcal{W}(\mu, \nu) \, d\mathbb{P}(\mu) d\mathbb{P}(\nu)$,

$$
\begin{aligned}
\frac{\mathbb{W}(\mathbb{P} + t\chi) - \mathbb{W}(\mathbb{P})}{t} &= \frac{1}{t} \left( \mathbb{W}(\mathbb{P}) + t \iint \mathcal{W}(\mu, \nu) \, d\mathbb{P}(\mu) d\chi(\nu) + t \iint \mathcal{W}(\mu, \nu) \, d\chi(\mu) d\mathbb{P}(\nu) \right. \\
&\qquad \left. + t^2 \iint \mathcal{W}(\mu, \nu) \, d\chi(\mu) d\chi(\nu) - \mathbb{W}(\mathbb{P}) \right) \\
&\xrightarrow[t \to 0]{} \int \left( \int \mathcal{W}(\mu, \nu) \, d\mathbb{P}(\mu) + \int \mathcal{W}(\nu, \mu) \, d\mathbb{P}(\mu) \right) \, d\chi(\nu).
\end{aligned}
\tag{58}
$$

Thus, the first variation is $\frac{\delta \mathbb{W}}{\delta \mathbb{P}}(\mathbb{P})(\nu) = \int \mathcal{W}(\mu, \nu) \, d\mathbb{P}(\mu) + \int \mathcal{W}(\nu, \mu) \, d\mathbb{P}(\mu)$, and its Wasserstein gradient coincides well with the WoW gradient derived in Proposition B.6.

**Relation with Euclidean gradient.** We provide now the analog of Proposition A.9 for WoW gradients, which we use in practice to compute them.

We fix here a number of classes $C > 0$ and a number of samples $n > 0$, and we consider the class of (fully) discrete measures of $\mathcal{P}_2(\mathcal{P}_2(\mathbb{R}^d))$ defined by

$$
\mathbb{P}_{\mathbf{x}} := \frac{1}{C} \sum_{c=1}^{C} \delta_{\mu_{\mathbf{x}^c}}, \quad \mathbf{x} \in (\mathbb{R}^d)^{C \times n}, \quad \mu_{\mathbf{x}^c} := \frac{1}{n} \sum_{i=1}^{n} \delta_{x_i^c}, \quad \mathbf{x}^c \in (\mathbb{R}^d)^n.
\tag{59}
$$

We also define the space

$$
X := \{ \mathbf{x} \in (\mathbb{R}^d)^{C \times n} \mid \forall c, \ \mathbf{x}^c \notin \Delta_n \text{ and } \forall c \neq c', \ \mu_{\mathbf{x}^c} \neq \mu_{\mathbf{x}^{c'}} \}.
\tag{60}
$$

where $\Delta_n := \{ \mathbf{x} \in (\mathbb{R}^d)^n \mid \exists i \neq j, x_i = x_j \}$ is the generalized diagonal of $(\mathbb{R}^d)^n$. Informally, $X$ is the space of vectors $\mathbf{x}$ such that the empirical measures $\mu_{\mathbf{x}^c}$ in the support of $\mathbb{P}_{\mathbf{x}}$ are all distinct, and are each supported on $n$ distinct points of $\mathbb{R}^d$.

**Proposition B.7.** *Let $\mathbb{F} : \mathcal{P}_2(\mathcal{P}_2(\mathbb{R}^d)) \mapsto \mathbb{R}$ a functional, and $F : (\mathbb{R}^d)^{C \times n} \mapsto \mathbb{R}$ such that for every $\mathbf{x} \in X$, $\mathbb{F}(\mathbb{P}_{\mathbf{x}}) = F(\mathbf{x})$. If $\mathbb{F}$ is Wasserstein differentiable at $\mathbb{P}_{\mathbf{x}}$ and $F$ is differentiable at $\mathbf{x}$ for some $\mathbf{x} \in X$, then for every $c \in \{1, \ldots, C\}$ and $i \in \{1, \ldots, n\}$,*

$$
\nabla_{W_{W_2}} \mathbb{F}(\mathbb{P}_{\mathbf{x}})(\mu_{\mathbf{x}^c})(x_i^c) = Cn \nabla_{c,i} F(\mathbf{x}).
\tag{61}
$$

*Proof.* See Appendix C.15. $\qquad \square$

## B.5. Convexity on $\mathcal{P}_2(\mathcal{P}_2(\mathcal{M}))$

In this section, we focus on $\mathcal{P}_2(\mathcal{P}_{2,\mathrm{ac}}(\mathbb{R}^d))$, and we show the convexity along generalized geodesics of potential energies and interaction energies. Hence, Proposition 4.1 can be applied to these functionals.

We recall that on $\mathcal{P}_2(\mathbb{R}^d)$, a generalized geodesic between $\mu, \nu \in \mathcal{P}_2(\mathbb{R}^d)$ is of the form $t \mapsto \mu_t = ((1-t)\pi^{1,2} + t\pi^{1,3})_{\#} \gamma$ with $\gamma \in \Pi(\eta, \mu, \nu)$ such that $\pi_{\#}^{1,2} \gamma \in \Pi_o(\eta, \mu)$ and $\pi_{\#}^{1,3} \gamma \in \Pi_o(\eta, \nu)$. Then a functional $\mathcal{F} : \mathcal{P}_2(\mathbb{R}^d) \to \mathbb{R}$ $\lambda$-convex along this curve satisfies for all $t \in [0, 1]$,

$$
\mathcal{F}(\mu_t) \leq (1-t)\mathcal{F}(\mu) + t\mathcal{F}(\nu) - \frac{\lambda t(1-t)}{2} W_2^2(\mu, \nu).
\tag{62}
$$

When $\eta \in \mathcal{P}_{2,\mathrm{ac}}(\mathbb{R}^d)$, by Brenier's theorem, there are OT maps $T_\eta^\mu$ between $\eta$ and $\mu$ and $T_\eta^\nu$ between $\eta$ and $\nu$, and the generalized geodesic translates as $\mu_t = ((1-t)T_\eta^\mu + tT_\eta^\nu)_{\#} \eta$.

We define similarly a generalized geodesic between $\mathbb{Q}, \mathbb{O} \in \mathcal{P}_2(\mathcal{P}_{2,\mathrm{ac}}(\mathbb{R}^d))$ as $t \mapsto \mathbb{P}_t = (((1-t)T_{\pi^1}^{\pi^2} + tT_{\pi^1}^{\pi^3})_{\#} \pi^1)_{\#} \Gamma$ where $\Gamma \in \Pi(\mathbb{P}, \mathbb{Q}, \mathbb{O})$, $\pi_{\#}^{1,2} \Gamma \in \Pi_o(\mathbb{P}, \mathbb{Q})$ and $\pi_{\#}^{1,3} \Gamma \in \Pi_o(\mathbb{P}, \mathbb{O})$. We provide sufficient conditions for potential and interaction energies to be $\lambda$-convex along generalized geodesics in $\mathcal{P}_2(\mathcal{P}_{2,\mathrm{ac}}(\mathbb{R}^d))$.

**Proposition B.8.** *Let $\lambda \geq 0$ and $\mathcal{F} : \mathcal{P}_2(\mathbb{R}^d) \to \mathbb{R}$ be $\lambda$-convex along generalized geodesics of $\mathcal{P}_2(\mathbb{R}^d)$. Then, the potential energy $\mathbb{F}(\mathbb{P}) = \int \mathcal{F}(\mu) d\mathbb{P}(\mu)$ is $\lambda$-convex along generalized geodesics on $\mathcal{P}_2(\mathcal{P}_{2,\mathrm{ac}}(\mathbb{R}^d))$.*

*Proof.* See Appendix C.16. □

**Proposition B.9.** *Let* $\mathcal{W} : \mathcal{P}_2(\mathbb{R}^d) \times \mathcal{P}_2(\mathbb{R}^d) \to \mathbb{R}$ *be joint convex along generalized geodesics of* $\mathcal{P}_2(\mathbb{R}^d)$. *Then, the interaction energy* $\mathbb{W}(\mathbb{P}) = \frac{1}{2} \iint \mathcal{W}(\mu, \nu) \, \mathrm{d}\mathbb{P}(\mu) \mathrm{d}\mathbb{P}(\nu)$ *is convex along generalized geodesics on* $\mathcal{P}_2\big(\mathcal{P}_{2,\mathrm{ac}}(\mathbb{R}^d)\big)$.

*Proof.* See Appendix C.17. □

We can also show that $\mathbb{F} : \mathbb{Q} \mapsto \frac{1}{2} \mathrm{W}_{\mathrm{W}_2}(\mathbb{Q}, \mathbb{P})^2$ is 1-convex along particular generalized geodesics, which have as anchor point $\mathbb{P}$.

**Proposition B.10.** *Let* $\mathbb{P}, \mathbb{Q}, \mathbb{O} \in \mathcal{P}_2\big(\mathcal{P}_{2,\mathrm{ac}}(\mathbb{R}^d)\big)$. *Define the generalized geodesic* $t \mapsto \mathbb{P}_t = \big(\big((1-t)\mathrm{T}_{\pi^1}^{\pi^2} + t\mathrm{T}_{\pi^1}^{\pi^3}\big)_{\#} \pi^1\big)_{\#} \Gamma$ *where* $\Gamma \in \Pi(\mathbb{P}, \mathbb{Q}, \mathbb{O})$, $\pi_{\#}^{1;2}\Gamma \in \Pi_o(\mathbb{P}, \mathbb{Q})$ *and* $\pi_{\#}^{1;3}\Gamma \in \Pi_o(\mathbb{P}, \mathbb{O})$. *Then* $\mathbb{F} : \mathbb{Q} \mapsto \frac{1}{2}\mathrm{W}_{\mathrm{W}_2}(\mathbb{Q}, \mathbb{P})^2$ *is 1-convex along this generalized geodesic,* i.e., *it satisfies for all* $t \in [0, 1]$,

$$\mathbb{F}(\mathbb{P}_t) \leq (1-t)\mathbb{F}(\mathbb{Q}) + t\mathbb{F}(\mathbb{O}) - \frac{t(1-t)}{2}\mathrm{W}_{\mathrm{W}_2}(\mathbb{Q}, \mathbb{O})^2. \tag{63}$$

*Proof.* See Appendix C.18. □

In particular, Proposition B.10 is the main result allowing to show Proposition 4.1, which can be applied for $\lambda$-convex potential energies with $\lambda \geq 0$ (or more generally, for any $\lambda \in \mathbb{R}$ such that $\frac{1}{\tau} + \lambda \geq 0$, see (Ambrosio et al., 2008, Assumption 4.0.1 and Theorem 4.0.4)), and for convex interaction energies.

## C. Proofs

### C.1. Proof of Proposition 2.1

In the forthcoming proofs, we will make use of the following selection theorem, whose statement can be found in (Villani et al., 2009, Chapter 5, Bibliographical notes) :

**Theorem C.1.** *If* $f : A \mapsto B$ *is a Borel surjective map between Polish spaces (i.e. separable complete metric spaces), such that all the fibers* $f^{-1}(y)$, $y \in B$ *are compact, then* $f$ *admits a Borel right-inverse.*

This allows us to prove

**Lemma C.2.** *There exists a measurable selection* $s : \mathcal{M}^2 \mapsto T\mathcal{M}$ *of the map*

$$f : \begin{cases} A = \{(x, v) \in T\mathcal{M}, \, d\big(x, \exp_x(v)\big) = \|v\|_x\} & \to & \mathcal{M} \times \mathcal{M} \\ (x, v) & \mapsto & \big(x, \exp_x(v)\big). \end{cases} \tag{64}$$

*Proof.* First, $A \subseteq T\mathcal{M}$ is a Polish space, as a closed subset of $T\mathcal{M}$ which is Polish. Second, for every $(x, y) \in \mathcal{M}^2$, the fiber $f^{-1}(x, y)$ is compact. Indeed, if we let $\{(x_n, v_n)\}_{n=1}^{\infty} \subseteq f^{-1}(x, y)$, then we have for every $n$, $x_n = x$, $y = \exp_x(v_n)$ and $\|v_n\|_x = d(x, y)$, so by compactness of the spheres in the tangent space $T_x\mathcal{M}$, up to extracting a subsequence there exists $v \in T_x\mathcal{M}$ such that $v_n \to v$, and by continuity $y = \exp_x(v)$ so that $(x, v) \in f^{-1}(x, y)$. We can thus apply Theorem C.1 to $f$ to deduce the existence of $s$. □

Now, we can prove Proposition 2.1. If $\gamma \in \exp_{\mu}^{-1}(\nu)$, we have $(\pi^{\mathcal{M}}, \exp)_{\#}\gamma \in \Pi(\mu, \nu)$ by definition of $\exp_{\mu}^{-1}(\nu)$. Furthermore

$$\int_{\mathcal{M}^2} d(x, y)^2 \mathrm{d}(\pi^{\mathcal{M}}, \exp)_{\#}\gamma(x, y) = \int_{T\mathcal{M}} d(x, \exp_x(v))^2 \mathrm{d}\gamma(x, v) \tag{65}$$

$$\leq \int_{T\mathcal{M}} \|v\|_x^2 \mathrm{d}\gamma(x, v) = \mathrm{W}_2^2(\mu, \nu), \tag{66}$$

so this transport plan is optimal. In particular the inequality is an equality, and we find that $d(x, \exp_x(v)) = \|v\|_x$ for $\gamma$-a.e. $(x, v)$. The map is thus well defined. To show that it is surjective, take $\gamma \in \Pi_o(\mu, \nu)$, and set $\tilde{\gamma} := s(\pi_1, \pi_2)_{\#}\gamma$,

where $s : \mathcal{M}^2 \mapsto T\mathcal{M}$ is the selection map defined in Lemma C.2. Then $\tilde{\gamma} \in \exp_\mu^{-1}(\nu)$. Indeed, by construction, $(\pi^{\mathcal{M}}, \exp)_{\#}\tilde{\gamma} = \gamma \in \Pi(\mu, \nu)$, and also

$$\mathrm{W}_2^2(\mu, \nu) = \int_{\mathcal{M}^2} d(x, y)^2 \mathrm{d}\gamma(x, y) = \int_{T\mathcal{M}} d(x, \exp_x(v))^2 \mathrm{d}\tilde{\gamma}(x, v) = \int_{T\mathcal{M}} \|v\|_x^2 \mathrm{d}\tilde{\gamma}(x, v). \tag{67}$$

This proves the surjectivity of the map.

Now, assume that $\mu$ is absolutely continuous, and that $\mathcal{M}$ is compact and connected. Let $\gamma \in \exp_\mu^{-1}(\nu)$ and $\tilde{\gamma} := (\pi^{\mathcal{M}}, \exp)_{\#}\gamma \in \Pi_o(\mu, \nu)$. By Theorem A.1, $\tilde{\gamma}$ is of the form $(\mathrm{Id}, T)_{\#}\mu$ where $T$ is the unique optimal transport map from $\mu$ to $\nu$. Furthermore $T$ itself is of the form $T(x) = \exp_x(-\nabla\varphi(x))$ where $\varphi$ is a $c$-concave function $\mathcal{M} \to \mathbb{R}$ (in fact a Kantorovich potential for the pair $\mu, \nu$). Furthermore, we have, for $\mu$-a.e. $x \in \mathcal{M}$, $T(x)$ belongs to

$$\partial^c \varphi(x) := \{y \in \mathcal{M}, \ \varphi(x) + \varphi^c(y) = \frac{1}{2}d(x, y)^2\}. \tag{68}$$

This implies, by (Gigli, 2011, Theorem 1.8), that for $\mu$-a.e. $x \in \mathcal{M}$, $\exp_x^{-1}(T(x)) \subseteq -\partial^+\varphi(x)$, where $\partial^+\varphi(x)$ is the superdifferential of $\varphi$. However, since $\mathcal{M}$ is compact, $\varphi$ is Lipschitz and is thus differentiable almost everywhere, so that $\partial^+\varphi(x) = \{\nabla\varphi(x)\}$ almost everywhere (in particular this holds $\mu$-a.e. as $\mu$ is absolutely continuous). From this, we conclude that for $\gamma$-a.e. $(x, v)$, we have $T(x) = \exp_x(v)$ with $\exp_x^{-1}(T(x)) = \{-\nabla\varphi(x)\}$, so that $v = -\nabla\varphi(x)$. Thus, we have proved that

$$\gamma = (\mathrm{Id}, -\nabla\varphi)_{\#}\mu. \tag{69}$$

This finishes the proof. $\qquad\square$

### C.2. Proof of Proposition 3.1

We first state and prove another selection result:

**Lemma C.3.** *There exists a measurable selection $s : \mathcal{P}_2(\mathcal{M})^2 \mapsto \mathcal{P}_2(T\mathcal{M})$ of the map*

$$f : A = \left\{ \begin{array}{ccc} \{\gamma \in \mathcal{P}_2(T\mathcal{M}) | \gamma \in \exp_{\pi_{\#}^{\mathcal{M}}\gamma}^{-1}(\exp_{\#}\gamma)\} & \to & \mathcal{P}_2(\mathcal{M}) \times \mathcal{P}_2(\mathcal{M}) \\ \gamma & \mapsto & (\pi_{\#}^{\mathcal{M}}\gamma, \exp_{\#}\gamma). \end{array} \right. \tag{70}$$

*Proof.* We first prove that $A \subseteq \mathcal{P}_2(T\mathcal{M})$ is a Polish space. All we need to do is to prove that it is closed : indeed, since $T\mathcal{M}$ is a connected Riemannian manifold (with the Sasaki metric), $(\mathcal{P}_2(T\mathcal{M}), \mathrm{W}_2)$ is a Polish space. (See (Ambrosio et al., 2008, Proposition 7.1.5). Similarly $\mathcal{P}_2(\mathcal{M})$ is a Polish space.) Note first that if $\gamma \in A$, it is supported on the compact set $K = \{(x, v) \in T\mathcal{M}, \ \|v\|_x \leq \mathrm{diam}(\mathcal{M})\} \subseteq T\mathcal{M}$, as $\|v\|_x = d(x, \exp_x(v))$ $\gamma$-almost everywhere. Let $\{\gamma_n\}_{n=1}^\infty \subseteq A$ converging to $\gamma \in \mathcal{P}_2(T\mathcal{M})$ in the $\mathrm{W}_2$ metric. Then $\gamma$ is also supported on $K$, and for every $n$, since $\gamma_n \in \exp_{\pi_{\#}^{\mathcal{M}}\gamma_n}^{-1}(\exp_{\#}\gamma_n)$, we have

$$\int \|v\|_x^2 \mathrm{d}\gamma_n(x, v) = \mathrm{W}_2^2(\pi_{\#}^{\mathcal{M}}\gamma_n, \exp_{\#}\gamma_n). \tag{71}$$

Letting $n \to \infty$, we find

$$\int \|v\|_x^2 \mathrm{d}\gamma(x, v) = \mathrm{W}_2^2(\pi_{\#}^{\mathcal{M}}\gamma, \exp_{\#}\gamma). \tag{72}$$

Indeed, $\mathrm{W}_2(\gamma_n, \gamma) \to 0$ implies weak convergence of $\gamma_n$ to $\gamma$, and thus of $\pi_{\#}^{\mathcal{M}}\gamma_n$ and $\exp_{\#}\gamma_n$ to respectively $\pi_{\#}^{\mathcal{M}}\gamma$ and $\exp_{\#}\gamma$, and since $\mathcal{M}$ is compact, weak convergence on $\mathcal{P}_2(\mathcal{M})$ is the same as $\mathrm{W}_2$ convergence, which in turn implies the convergence of the right-hand side. The left-hand side converges similarly in virtue of the weak convergence of $\gamma_n$ to $\gamma$ (recall that they are supported on the compact set $K$ on which the function $(x, v) \to \|v\|_x^2$ is bounded). Therefore, (72) implies that $\gamma \in \exp_{\pi_{\#}^{\mathcal{M}}\gamma}^{-1}(\exp_{\#}\gamma)$, so that $\gamma \in A$, and $A$ is thus closed in $\mathcal{P}_2(T\mathcal{M})$, and a Polish space.

Now, we prove that the fibers of $f$ are compact. Let $\mu, \nu \in \mathcal{P}_2(\mathcal{M})$ and $\{\gamma_n\}_{n=1}^\infty \subseteq f^{-1}(\mu, \nu)$. Since they are supported on $K$, and $\mathcal{P}(K)$ is compact by compactness of $K$, there exists $\gamma \in \mathcal{P}(K) \subseteq \mathcal{P}_2(T\mathcal{M})$ such that, up to extracting a subsequence, $\gamma_n$ converges to $\gamma$, both weakly and in the $\mathrm{W}_2$ metric. We then check as above that $\gamma \in A$, with $\pi_{\#}^{\mathcal{M}}\gamma = \mu$ and $\exp_{\#}\gamma = \nu$. This proves that the fibers of $f$ are compact. Now, the existence of $s$ follows again from Theorem C.1. $\quad\square$

The proof of Proposition 3.1 is pretty much similar to the previous one. If $\Gamma \in \exp_{\mathbb{P}}^{-1}(\mathbb{Q})$, we have $(\phi^{\mathcal{M}}, \phi^{\exp})_{\#}\Gamma \in \Pi(\mathbb{P}, \mathbb{Q})$ by definition of $\exp_{\mathbb{P}}^{-1}(\mathbb{Q})$. Furthermore,

$$\int_{\mathcal{P}_2(\mathcal{M})^2} W_2^2(\mu, \nu) \mathrm{d}(\phi^{\mathcal{M}}, \phi^{\exp})_{\#}\Gamma(\mu, \nu) = \int_{\mathcal{P}_2(T\mathcal{M})} W_2^2(\pi_{\#}^{\mathcal{M}}\gamma, \exp_{\#}\gamma) \mathrm{d}\Gamma(\gamma) \tag{73}$$

$$\leq \int_{\mathcal{P}_2(T\mathcal{M})} \int_{\mathcal{M}^2} d(x,y)^2 \mathrm{d}(\pi_{\mathcal{M}}, \exp)_{\#}\gamma(x,y) \mathrm{d}\Gamma(\gamma) \tag{74}$$

$$\leq \int_{\mathcal{P}_2(T\mathcal{M})} \int_{T\mathcal{M}} d(x, \exp_x(v))^2 \mathrm{d}\gamma(x,v) \mathrm{d}\Gamma(\gamma) \tag{75}$$

$$\leq \int_{\mathcal{P}_2(T\mathcal{M})} \int_{T\mathcal{M}} \|v\|_x^2 \mathrm{d}\gamma(x,v) \mathrm{d}\Gamma(\gamma) = W_{W_2}(\mathbb{P}, \mathbb{Q})^2, \tag{76}$$

so this transport plan is optimal. In particular all the inequalities are equalities, and we find that for $\Gamma$-a.e. $\gamma$, $W_2^2(\pi_{\#}^{\mathcal{M}}\gamma, \exp_{\#}\gamma) = \int \|v\|_x^2 \mathrm{d}\gamma(x,v)$ hence $\gamma \in \exp_{\pi_{\#}^{\mathcal{M}}\gamma}^{-1}(\exp_{\#}\gamma)$. The map is thus well defined. To show that it is surjective, take $\Gamma \in \Pi_o(\mathbb{P}, \mathbb{Q})$, and set $\tilde{\Gamma} := s(\pi_1, \pi_2)_{\#}\Gamma$, where $s : \mathcal{P}_2(\mathcal{M})^2 \mapsto \mathcal{P}_2(T\mathcal{M})$ is the selection map defined in Lemma C.3. Then $\tilde{\Gamma} \in \exp_{\mathbb{P}}^{-1}(\mathbb{Q})$ as, by construction, $(\phi^{\mathcal{M}}, \phi^{\exp})_{\#}\tilde{\Gamma} = \Gamma \in \Pi(\mathbb{P}, \mathbb{Q})$, and also

$$W_{W_2}(\mathbb{P}, \mathbb{Q})^2 = \int_{\mathcal{P}_2(\mathcal{M})^2} W_2^2(\mu, \nu) \mathrm{d}\Gamma(\mu, \nu) \tag{77}$$

$$= \int_{\mathcal{P}_2(T\mathcal{M})} W_2^2(\pi_{\#}^{\mathcal{M}}\gamma, \exp_{\#}\gamma) \mathrm{d}\tilde{\Gamma}(\gamma) \tag{78}$$

$$= \int_{\mathcal{P}_2(T\mathcal{M})} \int_{T\mathcal{M}} \|v\|_x^2 \mathrm{d}\gamma(x,v) \mathrm{d}\tilde{\Gamma}(\gamma). \tag{79}$$

This proves the surjectivity of the map.

Now, assume that $\mathbb{P}$ is absolutely continuous with respect to $\mathbb{P}_0$, and that $\mathcal{M}$ is compact and connected. Let $\Gamma \in \exp_{\mathbb{P}}^{-1}(\mathbb{Q})$ and $\tilde{\Gamma} := (\phi^{\mathcal{M}}, \phi^{\exp})_{\#}\Gamma \in \Pi_o(\mathbb{P}, \mathbb{Q})$. By (Emami & Pass, 2025, Theorem 13), $\tilde{\Gamma}$ is of the form $(\mathrm{Id}, T)_{\#}\mathbb{P}$ where $T$ is the unique optimal transport map from $\mathbb{P}$ to $\mathbb{Q}$. Furthermore, for $\Gamma$-a.e. $\gamma$, we have $\gamma \in \exp_{\mu_\gamma}^{-1}(\nu_\gamma)$, with $\mu_\gamma := \pi_{\#}^{\mathcal{M}}\gamma$, and $\nu_\gamma := \exp_{\#}\gamma$. Since $\tilde{\Gamma} = (\mathrm{Id}, T)_{\#}\mathbb{P}$ and $\mathbb{P}$ is concentrated on absolutely continuous measures (as $\mathbb{P} \ll \mathbb{P}_0$), we also have that for $\Gamma$-a.e. $\gamma$, $\mu_\gamma$ is absolutely continuous and $\nu_\gamma = T(\mu_\gamma)$. Thus, by Proposition 2.1, we have $\gamma = (\mathrm{Id}, -\nabla\varphi_{\mu_\gamma, T(\mu_\gamma)})_{\#}\mu_\gamma = (\mu \to (\mathrm{Id}, -\nabla\varphi_{\mu, T(\mu)})) \circ \phi^{\mathcal{M}}(\gamma)$ for $\Gamma$-a.e. $\gamma$. Therefore, we have proved that

$$\Gamma = (\mu \to (\mathrm{Id}, -\nabla\varphi_{\mu, T(\mu)}))_{\#}\phi_{\#}^{\mathcal{M}}\Gamma = (\mu \to (\mathrm{Id}, -\nabla\varphi_{\mu, T(\mu)})_{\#}\mu)_{\#}\mathbb{P}. \tag{80}$$

This finishes the proof. $\qquad\square$

*Remark* C.4. As a side note, there exists a more explicit expression for the unique $\Gamma \in \exp_{\mathbb{P}}^{-1}(\mathbb{Q})$. Let indeed $U : \mathcal{P}_2(\mathcal{M}) \mapsto \mathbb{R}$ be a Kantorovich potential for $\mathbb{P}, \mathbb{Q}$ (*i.e.* a $\frac{1}{2}W_2^2$-concave function solving the dual problem). Then, by the same reasoning as in the proof of (Emami & Pass, 2025, Theorem 13), for $\mathbb{P}$-almost every $\mu \in \mathcal{P}_2(\mathcal{M})$, we have $\nabla\varphi_{\mu, T(\mu)} = DU(\mu)$. Thus, we have

$$\Gamma = (\mu \to (\mathrm{Id}, -DU(\mu))_{\#}\mu)_{\#}\mathbb{P}. \tag{81}$$

### C.3. Proof of Proposition 3.7

The proof is inspired from (Erbar, 2010, Proposition 2.5) and (Ambrosio et al., 2008, Theorem 8.3.1).

First, fix $\varphi \in \mathrm{Cyl}(\mathcal{P}_2(\mathcal{M}))$, such that $\varphi(\mu) := F\left(\int V_1 \mathrm{d}\mu, \ldots, \int V_m \mathrm{d}\mu\right)$ with $F \in C_c^\infty(\mathbb{R}^m)$ and $V_1, \ldots, V_m \in C_c^\infty(\mathcal{M})$. Let us define $H : \mathcal{P}_2(\mathcal{M}) \times \mathcal{P}_2(\mathcal{M}) \to \mathbb{R}$ as

$$H(\mu, \nu) := \begin{cases} \frac{|\varphi(\mu) - \varphi(\nu)|}{W_2(\mu, \nu)} & \text{if } \mu \neq \nu \\ \|\nabla_{W_2}\varphi(\mu)\|_{L^2(\mu)} & \text{if } \mu = \nu. \end{cases} \tag{82}$$

We show in Lemma C.5 that $H$ is upper semicontinuous. We want to prove that $t \to \mathbb{P}_t(\varphi) = \int \varphi \mathrm{d}\mathbb{P}_t$ is absolutely

continuous and bound its metric derivative. For every $s, t \in I$, let $\Gamma_{s,t} \in \Pi_o(\mathbb{P}_s, \mathbb{P}_t)$. Then

$$
\begin{aligned}
\frac{1}{|h|} |\mathbb{P}_{s+h}(\varphi) - \mathbb{P}_s(\varphi)| &\leq \frac{1}{|h|} \int_{\mathcal{P}(\mathcal{M})^2} |\varphi(\mu) - \varphi(\nu)| \, d\Gamma_{s+h,s}(\mu, \nu) \\
&\leq \frac{1}{|h|} \int_{\mathcal{P}(\mathcal{M})^2} W_2(\mu, \nu) H(\mu, \nu) \, d\Gamma_{s+h,s}(\mu, \nu) \\
&\leq \frac{W_{W_2}(\mathbb{P}_{s+h}, \mathbb{P}_s)}{|h|} \sqrt{\int_{\mathcal{P}(\mathcal{M})^2} H^2(\mu, \nu) \, d\Gamma_{s+h,s}(\mu, \nu)}.
\end{aligned}
\tag{83}
$$

Now, we have $\Gamma_{s+h,s} \rightharpoonup (\mathrm{Id}, \mathrm{Id})_{\#} \mathbb{P}_s$ when $h \to 0$, so, since $H$ is upper semicontinuous, by (Ambrosio et al., 2008, Lemma 5.1.7), we have

$$
\limsup_{h \to 0} \int_{\mathcal{P}(\mathcal{M})^2} H^2(\mu, \nu) \, d\Gamma_{s+h,s}(\mu, \nu) \leq \int_{\mathcal{P}(\mathcal{M})} H^2(\mu, \mu) \, d\mathbb{P}_s(\mu) = \int_{\mathcal{P}(\mathcal{M})} \|\nabla_{W_2} \varphi(\mu)\|^2_{L^2(\mu)} \, d\mathbb{P}_s(\mu),
\tag{84}
$$

and thus $s \mapsto \mathbb{P}_s(\varphi)$ is absolutely continuous, with metric derivative bounded from above by

$$
|\mathbb{P}'|(s) \|\nabla_{W_2} \varphi\|_{L^2(\mathbb{P}_s, T\mathcal{P}_2(\mathcal{P}_2(\mathcal{M})))}.
\tag{85}
$$

Let now $\varphi \in \mathrm{Cyl}(I \times \mathcal{P}_2(\mathcal{M}))$, $Q = I \times \mathcal{P}_2(\mathcal{M})$, and $\lambda = \int_I d\mathbb{P}_t dt$. Then, for any interval $J \subseteq I$ such that $\mathrm{spt}(\varphi) \subseteq J \times \mathcal{P}_2(\mathcal{M})$,

$$
\begin{aligned}
\left| \int_Q \partial_t \varphi \, d\lambda \right| &= \left| \lim_{h \to 0^+} \int_Q \frac{\varphi(t, \mu) - \varphi(t - h, \mu)}{h} \, d\lambda(t, \mu) \right| \\
&\leq \limsup_{h \to 0^+} \left| \int_J \frac{\mathbb{P}_t(\varphi_t) - \mathbb{P}_{t+h}(\varphi_t)}{h} \, dt \right| \\
&\leq \int_J \limsup_{h \to 0^+} \frac{|\mathbb{P}_t(\varphi_t) - \mathbb{P}_{t+h}(\varphi_t)|}{|h|} \, dt \\
&\leq \int_J |\mathbb{P}'|(t) \|\nabla_{W_2} \varphi_t\|_{L^2(\mathbb{P}_t)} \, dt \\
&\leq \sqrt{\int_J |\mathbb{P}'|^2(t) \, dt} \sqrt{\int_J \|\nabla_{W_2} \varphi_t\|^2_{L^2(\mathbb{P}_t)} \, dt}.
\end{aligned}
\tag{86}
$$

From this, we infer that the linear form

$$
L(\nabla_{W_2} \varphi) := -\int_Q \partial_t \varphi \, d\lambda
\tag{87}
$$

is well-defined and Lipschitz continuous. In particular, there exists $v \in \overline{\{\nabla_{W_2} \varphi, \; \varphi \in \mathrm{Cyl}(I \times \mathcal{P}_2(\mathcal{M}))\}}^{L^2(\lambda, T\mathcal{P}_2(\mathcal{M}))}$ such that for every $\varphi \in \mathrm{Cyl}(I \times \mathcal{P}_2(\mathcal{M}))$,

$$
L(\nabla_{W_2} \varphi) = \langle v, \nabla_{W_2} \varphi \rangle_{L^2(\lambda)} = \int_I \int_{\mathcal{P}_2(\mathcal{M})} \langle v_t(\mu), \nabla_{W_2} \varphi(\mu) \rangle_{L^2(\mu)} \, d\mathbb{P}_t(\mu) dt,
\tag{88}
$$

and we have the continuity equation.

Moreover, for a.e. $t \in I$, $v_t \in \overline{\{\nabla_{W_2} \varphi, \; \varphi \in \mathrm{Cyl}(I \times \mathcal{P}_2(\mathcal{M}))\}}^{L^2(\mathbb{P}_t, T\mathcal{P}_2(\mathcal{M}))}$, and for every interval $J \subseteq I$, there exists a sequence $(\nabla_{W_2} \varphi_n)_n$ supported in $J \times \mathcal{P}_2(\mathcal{M})$ such that $\nabla_{W_2} \varphi_n \to v_t \mathbb{1}_J$. For all $n$,

$$
L(\nabla_{W_2} \varphi_n) \leq \left( \int_J |\mathbb{P}'|(t)^2 \, dt \right)^{\frac{1}{2}} \left( \int_J \|\nabla_{W_2} \varphi_n\|^2_{L^2(\mathbb{P}_t, T\mathcal{P}_2(\mathcal{M}))} \, dt \right)^{\frac{1}{2}}
\tag{89}
$$

and letting $n \to \infty$, we find

$$
\begin{aligned}
\int_J \|v_t\|^2_{L^2(\mathbb{P}_t, T\mathcal{P}_2(\mathcal{M}))} \, dt &\leq \left( \int_J |\mathbb{P}'|(t)^2 \, dt \right)^{\frac{1}{2}} \left( \int_J \|v_t\|^2_{L^2(\mathbb{P}_t, T\mathcal{P}_2(\mathcal{M}))} \, dt \right)^{\frac{1}{2}} \\
\int_J \|v_t\|^2_{L^2(\mathbb{P}_t, T\mathcal{P}_2(\mathcal{M}))} \, dt &\leq \int_J |\mathbb{P}'|(t)^2 \, dt.
\end{aligned}
\tag{90}
$$

We thus conclude that for a.e. $t \in I$,

$$\|v_t\|_{L^2(\mathbb{P}_t, T\mathcal{P}_2(\mathcal{M}))} \leq |\mathbb{P}'|(t). \tag{91}$$

$\square$

We now show that $H$ is upper semicontinuous.

**Lemma C.5.** *Let $\varphi \in \mathrm{Cyl}\big(\mathcal{P}_2(\mathcal{M})\big)$. The function $H : \mathcal{P}_2(\mathcal{M}) \times \mathcal{P}_2(\mathcal{M}) \to \mathbb{R}$ defined as*

$$H(\mu, \nu) := \begin{cases} \frac{|\varphi(\mu) - \varphi(\nu)|}{W_2(\mu, \nu)} & \text{if } \mu \neq \nu \\ \|\nabla_{W_2} \varphi(\mu)\|_{L^2(\mu)} & \text{if } \mu = \nu, \end{cases} \tag{92}$$

*is upper semicontinuous.*

*Proof.* We want to show that the function $H : \mathcal{P}_2(\mathcal{M}) \times \mathcal{P}_2(\mathcal{M}) \to \mathbb{R}$ defined by

$$H(\mu, \nu) := \begin{cases} \frac{|\varphi(\mu) - \varphi(\nu)|}{W_2(\mu, \nu)} & \text{if } \mu \neq \nu \\ \|\nabla_{W_2} \varphi(\mu)\|_{L^2(\mu)} & \text{if } \mu = \nu \end{cases} \tag{93}$$

is upper semicontinuous. Let $\mu, \nu \in \mathcal{P}_2(\mathcal{M})$, and let $(\mu_t)_{t \in [0,1]}$ be the constant speed geodesic from $\mu$ to $\nu$ with velocity field $v_t \in L^2(\mu_t)$. Then, we have

$$\varphi(\nu) - \varphi(\mu) = \int_0^1 \frac{d}{dt} \varphi(\mu_t) \, dt \tag{94}$$

$$= \int_0^1 \frac{d}{dt} F\left(\int V_1 d\mu_t, \ldots, \int V_m d\mu_t\right) dt \tag{95}$$

$$= \int_0^1 \sum_{i=1}^m \frac{\partial F}{\partial x_i}\left(\int V_1 d\mu_t, \ldots, \int V_m d\mu_t\right) \frac{d}{dt} \int V_i d\mu_t \, dt \tag{96}$$

$$= \int_0^1 \sum_{i=1}^m \frac{\partial F}{\partial x_i}\left(\int V_1 d\mu_t, \ldots, \int V_m d\mu_t\right) \int \langle \nabla V_i, v_t \rangle \, d\mu_t dt \tag{97}$$

$$= \int_0^1 \langle \nabla_{W_2} \varphi(\mu_t), v_t \rangle_{L^2(\mu_t)} dt \tag{98}$$

$$\leq \sqrt{\int_0^1 \|v_t\|_{L^2(\mu_t)}^2 \, dt} \sqrt{\int_0^1 \|\nabla_{W_2} \varphi(\mu_t)\|_{L^2(\mu_t)}^2 dt} \tag{99}$$

$$\leq W_2(\mu, \nu) \sqrt{\int_0^1 \|\nabla_{W_2} \varphi(\mu_t)\|_{L^2(\mu_t)}^2 dt}. \tag{100}$$

Hence,

$$H(\mu, \nu) \leq \sqrt{\int_0^1 \|\nabla_{W_2} \varphi(\mu_t)\|_{L^2(\mu_t)}^2 dt} < \infty \tag{101}$$

as the right-hand side is finite because the $F, V_i$ have compact support. Note that this inequality is also true when $\mu = \nu$ as $\mu_t = \mu$ for every $t$.

Furthermore, notice that the map $f : \mu \mapsto \|\nabla_{W_2} \varphi(\mu)\|_{L^2(\mu)}^2$ is continuous. Indeed, we can check that we have

$$f(\mu) = \int G\left(\int V_1 d\mu, \ldots, \int V_m d\mu, x\right) d\mu(x) \tag{102}$$

for some Lipschitz function $G$ with Lipschitz constant $L$, so that if $\mu^n \rightharpoonup \mu$, we have

$$|f(\mu^n) - f(\mu)| = \left| \int G\left( \int V_1 d\mu^n, \ldots, \int V_m d\mu^n, x \right) d\mu^n(x) - \int G\left( \int V_1 d\mu, \ldots, \int V_m d\mu, x \right) d\mu(x) \right| \quad (103)$$

$$\leq \left| \int G\left( \int V_1 d\mu^n, \ldots, \int V_m d\mu^n, x \right) d\mu^n(x) - \int G\left( \int V_1 d\mu, \ldots, \int V_m d\mu, x \right) d\mu^n(x) \right| \quad (104)$$

$$+ \left| \int G\left( \int V_1 d\mu, \ldots, \int V_m d\mu, x \right) d\mu^n(x) - \int G\left( \int V_1 d\mu, \ldots, \int V_m d\mu, x \right) d\mu(x) \right| \quad (105)$$

$$\leq L \sum_{i=1}^m \left| \int V_i \, d(\mu^n - \mu) \right| + \left| \int G\left( \int V_1 d\mu, \ldots, \int V_m d\mu, x \right) d(\mu^n - \mu)(x) \right| \to 0. \quad (106)$$

Now, if $\mu^n \rightharpoonup \mu$ and $\nu^n \rightharpoonup \mu$, then $\mu_t^n \rightharpoonup \mu$ for every $t$ (indeed $W_2(\mu_t^n, \mu) \leq W_2(\mu_t^n, \mu^n) + W_2(\mu^n, \mu) = t W_2(\nu^n, \mu^n) + W_2(\mu^n, \mu) \to 0$), and thus by what precedes $\|\nabla_{W_2}\varphi(\mu_t^n)\|_{L^2(\mu_t)}^2 \mapsto \|\nabla_{W_2}\varphi(\mu)\|_{L^2(\mu)}^2$ for every $t$. Therefore, by (101), we deduce

$$\limsup_n H(\mu^n, \nu^n) \leq \|\nabla_{W_2}\varphi(\mu)\|_{L^2(\mu)} = H(\mu, \mu). \quad (107)$$

This proves the upper semicontinuity of $H$. $\qquad\square$

## C.4. Proof of Proposition 3.8

Let $\mathbb{P} \in \mathcal{P}_2(\mathcal{P}_2(\mathcal{M}))$, and define $\mathcal{P}_2(\mathcal{P}_2(T\mathcal{M}))_{\mathbb{P}} := \{\Gamma \in \mathcal{P}_2(\mathcal{P}_2(T\mathcal{M})), \phi_\#^{\mathcal{M}}\Gamma = \mathbb{P}\}$. Fix $\Gamma \in \mathcal{P}_2(\mathcal{P}_2(T\mathcal{M}))_{\mathbb{P}}$, we define

$$\|\Gamma\|_{\mathbb{P}}^2 := \iint \|v\|_x^2 d\gamma(x, v) d\Gamma(\gamma). \quad (108)$$

By the disintegration theorem (see for example (Ambrosio et al., 2008, Theorem 5.3.1)), there exists a $\mathbb{P}$-a.e. unique family of probability measures $(\Gamma_\mu)_{\mu \in \mathcal{P}_2(\mathcal{M})}$ such that $\Gamma_\mu$ is supported on $\mathcal{P}_2(T\mathcal{M})_\mu$ and, for every measurable test function $f : \mathcal{P}_2(T\mathcal{M}) \mapsto \mathbb{R}^+$,

$$\int f(\gamma) d\Gamma(\gamma) = \iint f(\gamma) d\Gamma_\mu(\gamma) d\mathbb{P}(\mu). \quad (109)$$

We can use this family of measures to define the barycentric projection of $\Gamma$:

**Definition C.6.** The barycentric projection of $\Gamma$ is the vector field $\mathcal{B}(\Gamma) \in L^2(\mathbb{P}, T\mathcal{M})$ defined by

$$\mathcal{B}(\Gamma)(\mu) := \int \mathcal{B}(\gamma) d\Gamma_\mu(\gamma) \in L^2(\mu, T\mathcal{M}). \quad (110)$$

Note that we work here in the space $L^2(\mathbb{P}, T\mathcal{M})$, which is defined in Appendix B.1. It is a larger space than $L^2(\mathbb{P}, T\mathcal{P}_2(\mathcal{M}))$, with which it should not be confused. The barycentric projection satisfies the following properties:

**Proposition C.7.** For every $\xi \in L^2(\mathbb{P}, T\mathcal{M})$, it holds

$$\iint \langle \xi(\pi_\#^{\mathcal{M}}\gamma)(x), v \rangle_x d\gamma(x, v) \Gamma(\gamma) = \int \langle \xi(\mu), \mathcal{B}(\Gamma)(\mu) \rangle_{L^2(\mu)} d\mathbb{P}(\mu) = \langle \xi, \mathcal{B}(\Gamma) \rangle_{L^2(\mathbb{P})}. \quad (111)$$

Furthermore $\|\mathcal{B}(\Gamma)\|_{L^2(\mathbb{P})} \leq \|\Gamma\|_{\mathbb{P}}$.

*Proof.* For every $\xi \in L^2(\mathbb{P}, T\mathcal{M})$, we have

$$\iint \langle \xi(\pi_\#^{\mathcal{M}}\gamma)(x), v \rangle_x d\gamma(x, v) \Gamma(\gamma) = \iiint \langle \xi(\pi_\#^{\mathcal{M}}\gamma)(x), v \rangle_x d\gamma(x, v) d\Gamma_\mu(\gamma) d\mathbb{P}(\mu) \quad (112)$$

$$= \iiint \langle \xi(\mu)(x), v \rangle_x d\gamma(x, v) d\Gamma_\mu(\gamma) d\mathbb{P}(\mu) \quad (113)$$

$$= \iint \langle \xi(\mu), \mathcal{B}(\gamma) \rangle_{L^2(\mu)} d\Gamma_\mu(\gamma) d\mathbb{P}(\mu) \quad (114)$$

$$= \int \langle \xi(\mu), \mathcal{B}(\Gamma)(\mu) \rangle_{L^2(\mu)} d\mathbb{P}(\mu) = \langle \xi, \mathcal{B}(\Gamma) \rangle_{L^2(\mathbb{P})}. \quad (115)$$

Furthermore,

$$\|\mathcal{B}(\Gamma)\|_{L^2(\mathbb{P})}^2 = \iint \left\| \int \mathcal{B}(\gamma)(x) \mathrm{d}\Gamma_\mu(\gamma) \right\|_x^2 \mathrm{d}\mu(x) \mathrm{d}\mathbb{P}(\mu) \tag{116}$$

$$\leq \iiint \|\mathcal{B}(\gamma)(x)\|_x^2 \mathrm{d}\Gamma_\mu(\gamma) \mathrm{d}\mu(x) \mathrm{d}\mathbb{P}(\mu) \tag{117}$$

$$\leq \iint \|\mathcal{B}(\gamma)\|_{L^2(\mu)}^2 \mathrm{d}\Gamma_\mu(\gamma) \mathrm{d}\mathbb{P}(\mu) \tag{118}$$

$$\leq \iiint \|v\|_x^2 \mathrm{d}\gamma(x,v) \mathrm{d}\Gamma_\mu(\gamma) \mathrm{d}\mathbb{P}(\mu) \tag{119}$$

$$\leq \iint \|v\|_x^2 \mathrm{d}\gamma(x,v) \mathrm{d}\Gamma(\gamma) = \|\Gamma\|_{\mathbb{P}}^2, \tag{120}$$

where we used $\|\mathcal{B}(\gamma)\|_{L^2(\mu)}^2 \leq \|\gamma\|_\mu^2 = \int \|v\|_x^2 \mathrm{d}\gamma(x,v)$ to obtain the fourth line. $\qquad\square$

We now show Proposition 3.8.

Assume by contradiction that $\xi$ is not a strong subdifferential of $\mathbb{F}$ at $\mathbb{P}$. Then, there exists $\delta > 0$ and a sequence $\{\Gamma_n\}_{n=1}^\infty \subseteq \mathcal{P}_2\big(\mathcal{P}_2(T\mathcal{M})\big)_{\mathbb{P}}$ such that $\varepsilon_n := \|\Gamma_n\|_{\mathbb{P}} \xrightarrow[n\to\infty]{} 0$, and, for every $n$,

$$\mathbb{F}(\mathbb{P}_n) - \mathbb{F}(\mathbb{P}) - \iint \langle \xi(\pi_\#^{\mathcal{M}}\gamma), v\rangle_x \mathrm{d}\gamma(x,v) \mathrm{d}\Gamma_n(\gamma) \leq -\delta\varepsilon_n \tag{121}$$

with $\mathbb{P}_n := \phi_\#^{\exp}\Gamma_n$. Now, for every $n$, fix $\Upsilon_n \in \exp_{\mathbb{P}}^{-1}(\mathbb{P}_n)$. Since $\xi \in \partial^-\mathbb{F}(\mathbb{P})$, there exists $N > 0$ such that for every $n > N$,

$$\mathbb{F}(\mathbb{P}_n) - \mathbb{F}(\mathbb{P}) \geq \iint \langle \xi(\pi_\#^{\mathcal{M}}\gamma)(x), v\rangle_x \mathrm{d}\gamma(x,v) \mathrm{d}\Upsilon_n(\gamma) - \frac{\delta}{2} \mathrm{W}_{\mathrm{W}_2}(\mathbb{P}_n, \mathbb{P}). \tag{122}$$

Denoting $\Psi_n := \mathcal{B}(\Gamma_n)$ and $\Phi_n := \mathcal{B}(\Upsilon_n)$, we have, combining (121) and (122), that

$$\langle \xi, \Psi_n\rangle_{L^2(\mathbb{P})} - \delta\varepsilon_n \geq \langle \xi, \Phi_n\rangle_{L^2(\mathbb{P})} - \frac{\delta}{2} \mathrm{W}_{\mathrm{W}_2}(\mathbb{P}_n, \mathbb{P}). \tag{123}$$

Furthermore, we have $\mathrm{W}_{\mathrm{W}_2}(\mathbb{P}_n, \mathbb{P}) \leq \varepsilon_n$, since

$$\mathrm{W}_{\mathrm{W}_2}(\mathbb{P}_n, \mathbb{P})^2 \leq \int \mathrm{W}_2^2(\pi_\#^{\mathcal{M}}\gamma, \exp_\# \gamma) \mathrm{d}\Gamma_n(\gamma) \tag{124}$$

$$\leq \iint d^2(x, \exp_x(v)) \mathrm{d}\gamma(x,v) \mathrm{d}\Gamma_n(\gamma) \tag{125}$$

$$\leq \iint \|v\|_x^2 \mathrm{d}\gamma(x,v) \mathrm{d}\Gamma_n(\gamma) = \|\Gamma_n\|_{\mathbb{P}}^2 = \varepsilon_n^2. \tag{126}$$

Thus, we find for every $n > N$

$$\langle \xi, \Phi_n - \Psi_n\rangle_{L^2(\mathbb{P})} \leq -\frac{\delta}{2}\varepsilon_n. \tag{127}$$

Now, since $\|\Psi_n\|_{L^2(\mathbb{P})} \leq \|\Gamma_n\|_{\mathbb{P}} = \varepsilon_n$ and (by optimality of $\Upsilon_n$) $\|\Phi_n\|_{L^2(\mathbb{P})} \leq \|\Upsilon_n\|_{\mathbb{P}} = \mathrm{W}_{\mathrm{W}_2}(\mathbb{P}_n, \mathbb{P}) \leq \varepsilon_n$ for every $n$, it ensues that, up to extracting a subsequence, there exists $\Psi, \Phi \in L^2(\mathbb{P}, T\mathcal{M}))$ towards which $\varepsilon_n^{-1}\Psi_n$ and $\varepsilon_n^{-1}\Phi_n$ respectively converge weakly in $L^2(\mathbb{P}, T\mathcal{M})$. Therefore, dividing (127) by $\varepsilon_n$ and passing to the limit, we find

$$\langle \xi, \Phi - \Psi\rangle_{L^2(\mathbb{P})} \leq -\frac{\delta}{2}. \tag{128}$$

Now, fix $\mathcal{F} \in \mathrm{Cyl}\big(\mathcal{P}_2(\mathcal{M})\big)$. By applying Lemma C.8 to $\Gamma_n$ and $\Upsilon_n$, we find

$$\int \mathcal{F} \mathrm{d}\mathbb{P}_n = \int \mathcal{F} \mathrm{d}\mathbb{P} + \langle \nabla_{\mathrm{W}_2}\mathcal{F}, \Phi_n\rangle_{L^2(\mathbb{P})} + O(\varepsilon_n^2), \tag{129}$$

$$\int \mathcal{F} \mathrm{d}\mathbb{P}_n = \int \mathcal{F} \mathrm{d}\mathbb{P} + \langle \nabla_{\mathrm{W}_2}\mathcal{F}, \Psi_n\rangle_{L^2(\mathbb{P})} + O(\varepsilon_n^2). \tag{130}$$

Subtracting these two equations, dividing by $\varepsilon_n$ and passing to the limit, we thus find

$$\langle \nabla_{W_2} \mathcal{F}, \Phi - \Psi \rangle_{L^2(\mathbb{P})} = 0, \tag{131}$$

and this holds for any $\mathcal{F} \in \mathrm{Cyl}(\mathcal{P}_2(\mathcal{M}))$. However, by assumption, $\xi \in T_{\mathbb{P}} \mathcal{P}_2(\mathcal{P}_2(\mathcal{M}))$, and we recall that

$$T_{\mathbb{P}} \mathcal{P}_2(\mathcal{P}_2(\mathcal{M})) = \overline{\{\nabla_{W_2} \mathcal{F}, \mathcal{F} \in \mathrm{Cyl}(\mathcal{P}_2(\mathcal{M}))\}}^{L^2(\mathbb{P}, T\mathcal{P}_2(\mathcal{P}_2(\mathcal{M})))}. \tag{132}$$

This implies immediately that $\langle \xi, \Phi - \Psi \rangle_{L^2(\mathbb{P})} = 0$, which contradicts (128). $\qquad\square$

**Lemma C.8.** *Let $\mathcal{F} \in \mathrm{Cyl}(\mathcal{P}_2(\mathcal{M}))$, then, for every $\mathbb{P} \in \mathcal{P}_2(\mathcal{P}_2(\mathcal{M}))$ and $\Gamma \in \mathcal{P}_2(\mathcal{P}_2(T\mathcal{M}))_{\mathbb{P}}$,*

$$\left| \int \mathcal{F} \mathrm{d}\left( \phi_\#^{\exp} \Gamma \right) - \int \mathcal{F} \mathrm{d}\mathbb{P} - \iint \langle \nabla_{W_2} \mathcal{F}(\pi_\#^{\mathcal{M}} \gamma)(x), v \rangle_x \mathrm{d}\gamma(x, v) \mathrm{d}\Gamma(\gamma) \right| \leq C \|\Gamma\|_{\mathbb{P}}^2 \tag{133}$$

*for some constant $C$ depending only on $\mathcal{F}$.*

*Proof.* Let $F \in C_c^\infty(\mathbb{R}^m)$ and $V_1, \ldots, V_m \in C_c^\infty(\mathcal{M})$ be such that

$$\mathcal{F}(\mu) = F\left( \int V_1 \mathrm{d}\mu, \ldots, \int V_m \mathrm{d}\mu \right), \quad \mu \in \mathcal{P}_2(\mathcal{M}). \tag{134}$$

Since $F$ is compactly supported, there exists $C > 0$ which only depends on $F$ such that for every $x, h \in \mathbb{R}^m$,

$$|F(x + h) - F(x) - \langle \nabla F(x), h \rangle| \leq C \|h\|^2. \tag{135}$$

Fix $\gamma \in \mathcal{P}_2(T\mathcal{M})$, and let $\mu := \pi_\#^{\mathcal{M}} \gamma$ and $\nu := \exp_\# \gamma$. Since the $V_i$ are compactly supported, we know by Lemma C.12 that there exists some constant $L > 0$, which depends only on the $V_i$, such that for every $i$,

$$\left| \int V_i \mathrm{d}\nu - \int V_i \mathrm{d}\mu - \int \langle \nabla V_i(x), v \rangle \mathrm{d}\gamma(x, v) \right| \leq L \|\gamma\|_\mu^2. \tag{136}$$

Now, we have

$$\left| \mathcal{F}(\nu) - \mathcal{F}(\mu) - \int \langle \nabla_{W_2} \mathcal{F}(\mu)(x), v \rangle_x \mathrm{d}\gamma(x, v) \right| \tag{137}$$

$$= \left| \mathcal{F}(\nu) - \mathcal{F}(\mu) - \sum_{i=1}^m \frac{\partial F}{\partial x_i} \int \langle \nabla V_i(x), v \rangle_x \mathrm{d}\gamma(x, v) \right| \tag{138}$$

$$\leq \left| \mathcal{F}(\nu) - \mathcal{F}(\mu) - \sum_{i=1}^m \frac{\partial F}{\partial x_i} \left( \int V_i \mathrm{d}\nu - \int V_i \mathrm{d}\mu \right) \right| \tag{139}$$

$$+ \left| \sum_{i=1}^m \frac{\partial F}{\partial x_i} \left( \int V_i \mathrm{d}\nu - \int V_i \mathrm{d}\mu - \int \langle \nabla V_i(x), v \rangle_x \mathrm{d}\gamma(x, v) \right) \right| \tag{140}$$

$$\leq C \sum_{i=1}^m \left| \int V_i \mathrm{d}\nu - \int V_i \mathrm{d}\mu \right|^2 + C \sum_{i=1}^m \left| \int V_i \mathrm{d}\nu - \int V_i \mathrm{d}\mu - \int \langle \nabla V_i(x), v \rangle \mathrm{d}\gamma(x, v) \right| \tag{141}$$

$$\leq C \sum_{i=1}^m \left| \int V_i \mathrm{d}\nu - \int V_i \mathrm{d}\mu \right|^2 + C \|\gamma\|_\mu^2 \tag{142}$$

$$\leq C \sum_{i=1}^m \left| \int V_i \mathrm{d}\nu - \int V_i \mathrm{d}\mu - \int \langle \nabla V_i(x), v \rangle \mathrm{d}\gamma(x, v) \right|^2 + \left| \int \langle \nabla V_i(x), v \rangle \mathrm{d}\gamma(x, v) \right|^2 + C \|\gamma\|_\mu^2 \tag{143}$$

$$\leq C \|\gamma\|_\mu^4 + C \|\gamma\|_\mu^2, \tag{144}$$

where we used (135) in the fifth line, with $x_i = \int V_i \mathrm{d}\mu$ and $h_i = \int V_i \mathrm{d}\nu - \int V_i \mathrm{d}\mu$, we used (136) in the sixth and eighth lines, and we used the Cauchy-Schwarz inequality in the eight line. Throughout the derivation, $C$ denotes a constant that

may change between lines but which only depends on $F$ and the $V_i$. In particular, there exists a constant $C$ which only depends on $F$ and the $V_i$ such that for every $\gamma \in \mathcal{P}_2(T\mathcal{M})$ with $\mu = \pi_\#^\mathcal{M}\gamma$ and $\|\gamma\|_\mu \leq \operatorname{diam}(\mathcal{M})$,

$$\left| \mathcal{F}(\exp_\# \gamma) - \mathcal{F}(\mu) - \int \langle \nabla_{W_2}\mathcal{F}(\mu)(x), v \rangle_x \mathrm{d}\gamma(x,v) \right| \leq C\|\gamma\|_\mu^2. \tag{145}$$

Now, if $\gamma \in \mathcal{P}_2(T\mathcal{M})$ is such that $\|\gamma\|_\mu > \operatorname{diam}(\mathcal{M})$ with $\mu := \pi_\#^\mathcal{M}\gamma$, let $\nu := \exp_\# \gamma$ and $\eta \in \exp_\mu^{-1}(\nu)$. Since $\|\eta\|_\mu = W_2(\mu,\nu) \leq \operatorname{diam}(\mathcal{M})$, this implies

$$\left| \mathcal{F}(\nu) - \mathcal{F}(\mu) - \int \langle \nabla_{W_2}\mathcal{F}(\mu)(x), v \rangle_x \mathrm{d}\gamma(x,v) \right| \leq \left| \mathcal{F}(\nu) - \mathcal{F}(\mu) - \int \langle \nabla_{W_2}\mathcal{F}(\mu)(x), v \rangle_x \mathrm{d}\eta(x,v) \right| \tag{146}$$

$$+ \left| \int \langle \nabla_{W_2}\mathcal{F}(\mu)(x), v \rangle_x \, \mathrm{d}(\eta - \gamma)(x,v) \right| \tag{147}$$

$$\leq C\|\eta\|_\mu^2 + |\langle \nabla_{W_2}\mathcal{F}(\mu), \mathcal{B}(\eta) - \mathcal{B}(\gamma) \rangle_{L^2(\mu)}| \tag{148}$$

$$\leq C\|\eta\|_\mu^2 + C(\|\mathcal{B}(\eta)\|_{L^2(\mu)} + \|\mathcal{B}(\gamma)\|_{L^2(\mu)}) \tag{149}$$

$$\leq C(\|\eta\|_\mu^2 + \|\eta\|_\mu + \|\gamma\|_\mu) \tag{150}$$

$$\leq C(\|\gamma\|_\mu^2 + \|\gamma\|_\mu) \tag{151}$$

$$\leq C\|\gamma\|_\mu^2, \tag{152}$$

where we used (145) in the third line, and we obtain the fourth line using the Cauchy-Schwarz inequality and the fact that $\sup_{\mu \in \mathcal{P}_2(\mathcal{M})} \|\nabla_{W_2}\mathcal{F}(\mu)\|_{L^2(\mu)} < +\infty$ (since the $F$ and $V_i$ are compactly supported). Again the $C$'s denote a constant depending only on $F$ and the $V_i$ (and $\operatorname{diam}(\mathcal{M})$). Thus, we have shown that there exists a constant $C$ depending only on $\mathcal{F}$ such that for every $\gamma \in \mathcal{P}_2(T\mathcal{M})$,

$$\left| \mathcal{F}(\exp_\# \gamma) - \mathcal{F}(\pi_\#^\mathcal{M}\gamma) - \int \langle \nabla_{W_2}\mathcal{F}(\pi_\#^\mathcal{M}\gamma)(x), v \rangle_x \mathrm{d}\gamma(x,v) \right| \leq C\|\gamma\|_\mu^2. \tag{153}$$

Hence for every $\Gamma \in \mathcal{P}_2\big(\mathcal{P}_2(T\mathcal{M})\big)$, noting $\mathbb{P} := \phi_\#^\mathcal{M}\Gamma$ and $\mathbb{Q} := \phi_\#^{\exp}\Gamma$,

$$\left| \int \mathcal{F}\mathrm{d}\mathbb{Q} - \int \mathcal{F}\mathrm{d}\mathbb{P} - \iint \langle \nabla_{W_2}\mathcal{F}(\pi_\#^\mathcal{M}\gamma)(x), v \rangle_x \mathrm{d}\gamma(x,v)\mathrm{d}\Gamma(\gamma) \right| \tag{154}$$

$$= \left| \int \mathcal{F}(\exp_\# \gamma) - \mathcal{F}(\pi_\#^\mathcal{M}\gamma) - \langle \nabla_{W_2}\mathcal{F}(\pi_\#^\mathcal{M}\gamma)(x), v \rangle_x \mathrm{d}\gamma(x,v)\mathrm{d}\Gamma(\gamma) \right| \tag{155}$$

$$\leq C \int \|\gamma\|_{\pi_\#^\mathcal{M}\gamma}^2 \mathrm{d}\Gamma(\gamma) = C\|\Gamma\|_\mathbb{P}^2. \tag{156}$$

This finishes the proof. $\qquad\square$

## C.5. Proof of Proposition 3.9, and existence of gradients in the tangent space

First, we prove Proposition 3.9. Let $\xi_1, \xi_2 \in \partial^-\mathbb{F}(\mathbb{P}) \cap \partial^+\mathbb{F}(\mathbb{P}) \cap T_\mathbb{P}\mathcal{P}_2\big(\mathcal{P}_2(\mathcal{M})\big)$. Using Proposition 3.8, we know that they are also strong gradients of $\mathbb{F}$ at $\mathbb{P}$. Therefore, letting $\xi = \xi_1 - \xi_2$, for every $\Psi \in L^2(\mathbb{P}, T\mathcal{P}_2(\mathcal{M}))$, we have

$$\int \langle \xi(\mu), \Psi(\mu) \rangle_{L^2(\mu)} \mathrm{d}\mathbb{P}(\mu) = o(\|\Psi\|_{L^2(\mathbb{P})}). \tag{157}$$

that is, $\langle \xi, \Psi \rangle_{L^2(\mathbb{P})} = o(\|\Psi\|_{L^2(\mathbb{P})})$. Considering $\Psi = \varepsilon\xi$, we obtain $\varepsilon\|\xi\|_{L^2(\mathbb{P})}^2 = o(\varepsilon)$ that is $\|\xi\|_{L^2(\mathbb{P})}^2 = o(1)$, and this implies $\xi = \xi_1 - \xi_2 = 0$, and this finishes the proof. $\qquad\square$

Now, a natural question is to ask whether there is a gradient in $T_\mathbb{P}\mathcal{P}_2\big(\mathcal{P}_2(\mathcal{M})\big)$ whenever there exists a gradient $\xi \in \partial^-\mathbb{F}(\mathbb{P}) \cap \partial^+\mathbb{F}(\mathbb{P})$. While a complete answer to this question is out of the scope of this article, a partial answer can be provided using results laid out in (Dello Schiavo, 2020). First, we consider the following assumption:

**Assumption C.9.** (Smooth transport property, (Dello Schiavo, 2020, Assumption 2.9)) We say that $\mathcal{M}$ satisfies the *smooth transport property* if, whenever $\mu, \nu \in \mathcal{P}_2(\mathcal{M})$ are absolutely continuous with smooth nowhere vanishing densities, then there exists a smooth optimal transport map $T : \mathcal{M} \mapsto \mathcal{M}$ from $\mu$ to $\nu$ (in the sense of Theorem A.1).

This is a relatively restrictive assumption on $\mathcal{M}$. By (Dello Schiavo, 2020, Theorem 5.9), it holds whenever $\mathcal{M}$ satisfies the strong Ma-Trudinger-Wang condition $\mathrm{MTW}(K)$ for some $K > 0$ (we refer to (Dello Schiavo, 2020, Section 5.2) for further details). Under this assumption, we can prove the following result on the existence of a gradient in the tangent space:

**Proposition C.10.** *Assume that $\mathcal{M}$ satisfies Assumption C.9, and that $\mathbb{P}$ satisfies Assumption B.2. Then, if $\mathbb{F}$ admits a WoW gradient at $\mathbb{P}$ (i.e. $\partial^- \mathbb{F}(\mathbb{P}) \cap \partial^+ \mathbb{F}(\mathbb{P})$ is not empty), then it admits a WoW gradient in $T_{\mathbb{P}} \mathcal{P}_2\big(\mathcal{P}_2(\mathcal{M})\big)$ (i.e., $\partial^- \mathbb{F}(\mathbb{P}) \cap \partial^+ \mathbb{F}(\mathbb{P}) \cap T_{\mathbb{P}} \mathcal{P}_2\big(\mathcal{P}_2(\mathcal{M})\big)$ is not empty).*

*Proof.* All we need to do is to prove that for every $\xi \in T_{\mathbb{P}} \mathcal{P}_2\big(\mathcal{P}_2(\mathcal{M})\big)^\perp$, $\mathbb{Q} \in \mathcal{P}_2\big(\mathcal{P}_2(\mathcal{M})\big)$ and $\Gamma \in \exp_{\mathbb{P}}^{-1}(\mathbb{Q})$,

$$\iint \langle \xi(\pi_\#^{\mathcal{M}} \gamma)(x), v \rangle_x \mathrm{d}\gamma(x,v) \mathrm{d}\Gamma(\gamma) = 0. \tag{158}$$

Indeed, this ensures that if $\xi \in \partial^- \mathbb{F}(\mathbb{P}) \cap \partial^+ \mathbb{F}(\mathbb{P})$ is a WoW gradient of $\mathbb{F}$ at $\mathbb{P}$, then its orthogonal projection on $T_{\mathbb{P}} \mathcal{P}_2\big(\mathcal{P}_2(\mathcal{M})\big)$ is also a WoW gradient.

We thus fix $\xi \in T_{\mathbb{P}} \mathcal{P}_2\big(\mathcal{P}_2(\mathcal{M})\big)^\perp$, $\mathbb{Q} \in \mathcal{P}_2\big(\mathcal{P}_2(\mathcal{M})\big)$ and $\Gamma \in \exp_{\mathbb{P}}^{-1}(\mathbb{Q})$. Since $\mathbb{P}$ satisfies Assumption B.2, by Remark C.4, $\Gamma$ is of the form $(\mu \mapsto (\mathrm{Id}, -D_{\mathbb{P}} U)_\# \mu)_\# \mathbb{P}$ where $U$ is a Kantorovich potential for the pair $\mathbb{P}, \mathbb{Q}$. However, since $\mathcal{M}$ satisfies Assumption C.9, and $U$ is $W_2$-Lipschitz (by (Emami & Pass, 2025, Lemma 12)), (Dello Schiavo, 2020, Theorem 2.10(3)) implies that $D_{\mathbb{P}} U \in T_{\mathbb{P}} \mathcal{P}_2\big(\mathcal{P}_2(\mathcal{M})\big)$ (as a limit in $L^2(\mathbb{P}, T\mathcal{P}_2(\mathcal{M}))$ of functions of the form $\nabla_{W_2} \mathcal{F}$, $\mathcal{F} \in \mathrm{Cyl}\big(\mathcal{P}_2(\mathcal{M})\big)$), so that

$$\iint \langle \xi(\pi_\#^{\mathcal{M}} \gamma)(x), v \rangle_x \mathrm{d}\gamma(x,v) \mathrm{d}\Gamma(\gamma) = -\int \langle \xi(\mu), D_{\mathbb{P}} U(\mu) \rangle_{L^2(\mu)} \mathrm{d}\mathbb{P}(\mu) = 0. \tag{159}$$

This finishes the proof. $\square$

Note that for this proposition to hold, $\mathbb{P}$ must not simply be absolutely continuous w.r.t $\mathbb{P}_0$, but must itself satisfy Assumption B.2. According to (Dello Schiavo, 2020, Proposition 5.2), this is the case whenever, for instance, $\mathbb{P} = \varphi^2 \mathbb{P}_0$ where $\varphi$ is a strictly positive $W_2$-Lipschitz function on $\mathcal{P}_2(\mathcal{M})$.

## C.6. Proof of Proposition 4.1

We aim at applying (Ambrosio et al., 2008, Theorem 4.0.4). Since by hypothesis, $\mathbb{F}$ is $\lambda$-convex along the curve $\mathbb{P}_t = \big(((1-t)\mathrm{T}_{\pi^1}^{\pi^2} + t\mathrm{T}_{\pi^1}^{\pi^3})_\# \pi^1\big)_\# \Gamma$ for $\Gamma \in \Pi(\mathbb{P}, \mathbb{Q}, \mathbb{O})$ and $\mathbb{P} \in \mathcal{P}_2\big(\mathcal{P}_{2,\mathrm{ac}}(\mathbb{R}^d)\big)$, we need to show that $\mathbb{G} : \mathbb{Q} \mapsto \frac{1}{2} W_{W_2}(\mathbb{Q}, \mathbb{P})^2$ is 1-convex along $\mathbb{P}_t$ (see *e.g.* (Ambrosio et al., 2008, Lemma 9.2.7)). This is well the case by Proposition B.10.

Now, let $\mathbb{P}_k \in \mathcal{P}_2\big(\mathcal{P}_{2,\mathrm{ac}}(\mathbb{R}^d)\big)$ and $\mathbb{J}(\mathbb{P}) = \frac{1}{2\tau} W_{W_2}(\mathbb{P}, \mathbb{P}_k)^2 + \mathbb{F}(\mathbb{P})$ the functional solved at each step of the JKO scheme. Then, we have

$$\begin{aligned}
\mathbb{J}(\mathbb{P}_t) &= \frac{1}{2\tau} W_{W_2}(\mathbb{P}_t, \mathbb{P}_k)^2 + \mathbb{F}(\mathbb{P}_t) \\
&= \frac{1}{\tau} \mathbb{G}(\mathbb{P}) + \mathbb{F}(\mathbb{P}) \\
&\leq \frac{1}{\tau}\Big((1-t)\mathbb{G}(\mathbb{Q}) + t\mathbb{G}(\mathbb{O}) - \frac{t(1-t)}{2} W_{W_2}(\mathbb{Q}, \mathbb{O})^2\Big) \\
&\quad + (1-t)\mathbb{F}(\mathbb{Q}) + t\mathbb{F}(\mathbb{O}) - \frac{\lambda t(1-t)}{2} W_{W_2}(\mathbb{Q}, \mathbb{O})^2 \\
&= (1-t)\mathbb{J}(\mathbb{Q}) + t\mathbb{J}(\mathbb{O}) - \frac{\lambda\tau + 1}{2\tau} t(1-t) W_{W_2}(\mathbb{Q}, \mathbb{O})^2.
\end{aligned} \tag{160}$$

Thus, we conclude that $\mathbb{J}$ satisfies well (Ambrosio et al., 2008, Assumption (4.0.1)), and then apply (Ambrosio et al., 2008, Theorem 4.0.4). $\square$

## C.7. Proof of Proposition A.4

Let $\xi, \xi' \in T_\mu \mathcal{P}_2(\mathcal{M}) \cap \partial^+ \mathcal{F}(\mu) \cap \partial^- \mathcal{F}(\mu)$. By density, for any $\varepsilon > 0$, there exist $\varphi_\varepsilon, \varphi_\varepsilon' \in C_c^\infty(\mathcal{M})$ such that $\|\xi - \nabla\varphi_\varepsilon\|_{L^2(\mu, T\mathcal{M})} \leq \frac{\varepsilon}{2}$ and $\|\xi' - \nabla\varphi'\|_{L^2(\mu, T\mathcal{M})} \leq \frac{\varepsilon}{2}$.

We rely on the following Lemma, which provides an OT map for any $\psi \in C_c^\infty(\mathcal{M})$ for $s$ small enough.

**Lemma C.11.** *Let $\mu \in \mathcal{P}_2(\mathcal{M})$ and $\psi \in C_c^\infty(\mathcal{M})$. Then, there exists $\bar{s}$ such that $x \mapsto \exp_x(s\nabla\psi(x))$ is an OT map between $\mu$ and $\left(\exp \circ (s\nabla\psi)\right)_{\#}\mu$ for all $s \in ]-\bar{s}, \bar{s}[$.*

*Proof.* Suppose $\psi \neq 0$. Let $\varepsilon > 0$ and $\bar{s} = \frac{\varepsilon}{\max_x \|\nabla^2\psi(x)\|}$. It exists as $\psi$ is supported on a compact. Then, for any $s \in (-\bar{s}, \bar{s})$, $\|s\nabla^2\psi\| \leq \bar{s}\|\nabla^2\psi\| \leq \varepsilon$. Then, by (Villani et al., 2009, Theorem 13.5), $s\psi$ is $d^2/2$ convex, and by McCann's theorem, $\exp(s\nabla\psi)$ is an OT map. $\qquad\square$

By Lemma C.11, there exists $s > 0$ such that $\gamma = (\mathrm{Id}, s\nabla\varphi_\varepsilon)_{\#}\mu \in \exp_\mu^{-1}(\nu)$ and $\gamma' = (\mathrm{Id}, s\nabla\varphi'_\varepsilon)_{\#}\mu \in \exp_\mu^{-1}(\nu')$ with $\nu = \left(\exp \circ (s\nabla\varphi_\varepsilon)\right)_{\#}\mu$ and $\nu' = \left(\exp \circ (s\nabla\varphi'_\varepsilon)\right)_{\#}\mu$. Thus, we have using the definitions of sub-differentials that

$$\begin{cases} \mathcal{F}(\nu) \geq \mathcal{F}(\mu) + s \int \langle \xi(x), \nabla\varphi_\varepsilon(x)\rangle_x \, \mathrm{d}\mu(x) + o(s) \\ \mathcal{F}(\nu') \geq \mathcal{F}(\mu) + s \int \langle \xi'(x), \nabla\varphi'_\varepsilon(x)\rangle_x \, \mathrm{d}\mu(x) + o(s). \end{cases} \tag{161}$$

Likewise, by the definition of the super-differentials, we have

$$\begin{cases} \mathcal{F}(\nu') \leq \mathcal{F}(\mu) + s \int \langle \xi(x), \nabla\varphi'_\varepsilon(x)\rangle_x \, \mathrm{d}\mu(x) + o(s) \\ \mathcal{F}(\nu) \leq \mathcal{F}(\mu) + s \int \langle \xi'(x), \nabla\varphi_\varepsilon(x)\rangle_x \, \mathrm{d}\mu(x) + o(s). \end{cases} \tag{162}$$

Dividing by $s > 0$ and rearranging the terms, we have

$$\begin{cases} \frac{\mathcal{F}(\nu)-\mathcal{F}(\mu)}{s} \geq \langle \xi, \nabla\varphi_\varepsilon\rangle_{L^2(\mu,T\mathcal{M})} + o(1) \\ \frac{\mathcal{F}(\nu')-\mathcal{F}(\mu)}{s} \geq \langle \xi', \nabla\varphi'_\varepsilon\rangle_{L^2(\mu,T\mathcal{M})} + o(1) \\ \frac{\mathcal{F}(\mu)-\mathcal{F}(\nu')}{s} \geq \langle -\xi, \nabla\varphi'_\varepsilon\rangle_{L^2(\mu,T\mathcal{M})} + o(1) \\ \frac{\mathcal{F}(\mu)-\mathcal{F}(\nu)}{s} \geq \langle -\xi', \nabla\varphi_\varepsilon\rangle_{L^2(\mu,T\mathcal{M})} + o(1). \end{cases} \tag{163}$$

Summing them, we get,

$$0 \geq \langle \xi - \xi', \nabla\varphi_\varepsilon\rangle_{L^2(\mu,T\mathcal{M})} + \langle \xi' - \xi, \nabla\varphi'_\varepsilon\rangle_{L^2(\mu,T\mathcal{M})} + o(1) = \langle \xi - \xi', \nabla\varphi_\varepsilon - \nabla\varphi'_\varepsilon\rangle_{L^2(\mu,T\mathcal{M})} + o(1). \tag{164}$$

Then, we have

$$\begin{aligned} \|\xi - \xi'\|_{L^2(\mu,T\mathcal{M})} &\leq \sqrt{\|\xi - \xi'\|_{L^2(\mu,T\mathcal{M})}^2 - 2\langle \xi - \xi', \nabla\varphi_\varepsilon - \nabla\varphi'_\varepsilon\rangle_{L^2(\mu,T\mathcal{M})} + \|\nabla\varphi_\varepsilon - \nabla\varphi'_\varepsilon\|_{L^2(\mu,T\mathcal{M})}^2} \\ &= \|\xi - \xi' - (\nabla\varphi_\varepsilon - \nabla\varphi'_\varepsilon)\|_{L^2(\mu,T\mathcal{M})} \\ &\leq \|\xi - \nabla\varphi_\varepsilon\|_{L^2(\mu,T\mathcal{M})} + \|\xi' - \nabla\varphi'_\varepsilon\|_{L^2(\mu,T\mathcal{M})} \\ &\leq \varepsilon. \end{aligned} \tag{165}$$

Taking the limit $\varepsilon \to 0$, we conclude that $\xi = \xi'$. $\qquad\square$

### C.8. Proof of Proposition A.5

We assume by contradiction that $\xi$ is not an extended strong subdifferential. Then there exists a sequence $\{\gamma_n\}_{n=1}^\infty \subseteq \mathcal{P}_2(T\mathcal{M})_\mu$ and $\delta > 0$ such that $\varepsilon_n := \|\gamma_n\|_\mu \to 0$ and for every $n$, denoting $\mu_n := \exp_{\#}\gamma_n$,

$$\mathcal{F}(\mu_n) - \mathcal{F}(\mu) - \int \langle \xi(x), v\rangle_x \mathrm{d}\gamma_n(x, v) \leq -\delta\varepsilon_n. \tag{166}$$

Now, let $\eta_n \in \exp_\mu^{-1}(\mu_n)$. Since $\xi$ is a subdifferential, there exists $N$ such that for every $n > N$,

$$\mathcal{F}(\mu_n) - \mathcal{F}(\mu) - \int \langle \xi(x), v\rangle_x \mathrm{d}\eta_n(x, v) \geq -\frac{\delta}{2}W_2(\mu, \mu_n). \tag{167}$$

Since, by optimality of $\eta_n$, we have $\|\eta_n\|_\mu = W_2(\mu, \mu_n) \leq \|\gamma_n\|_\mu = \varepsilon_n$, combining these inequalities, we find

$$\int \langle \xi(x), v\rangle_x \mathrm{d}\eta_n(x, v) - \int \langle \xi(x), v\rangle_x \mathrm{d}\gamma_n(x, v) \leq -\frac{\delta}{2}\varepsilon_n \tag{168}$$

that is

$$\langle \xi, \Phi_n - \Psi_n \rangle_{L^2(\mu)} \leq -\frac{\delta}{2}\varepsilon_n \tag{169}$$

for every $n > N$, where we have defined $\Psi_n := \mathcal{B}(\gamma_n)$ and $\Phi_n := \mathcal{B}(\eta_n)$. Since we have $\|\Psi_n\|_{L^2(\mu)} \leq \|\gamma_n\|_\mu = \varepsilon_n$ and likewise $\|\Phi_n\|_{L^2(\mu)} \leq \|\eta_n\|_\mu \leq \varepsilon_n$, up to extracting a subsequence we can assume that there exists $\Psi, \Phi \in L^2(\mu, T\mathcal{M})$ towards which $\varepsilon_n^{-1}\Psi_n$ and $\varepsilon_n^{-1}\Phi_n$ respectively converge weakly in $L^2(\mu, T\mathcal{M})$. Thus, dividing (169) by $\varepsilon_n$ and passing to the limit, we find

$$\langle \xi, \Phi - \Psi \rangle_{L^2(\mu)} \leq -\frac{\delta}{2}. \tag{170}$$

Now, fix some $\varphi \in C_c^\infty(\mathcal{M})$. By Lemma C.12, we have

$$\int \varphi \mathrm{d}\mu_n = \int \varphi \mathrm{d}\mu + \int \langle \nabla\varphi(x), v \rangle_x \mathrm{d}\eta_n(x, v) + O(\|\eta_n\|_\mu^2) \tag{171}$$

$$= \int \varphi \mathrm{d}\mu + \langle \nabla\varphi, \Phi_n \rangle_{L^2(\mu)} + O(\varepsilon_n^2), \tag{172}$$

and similarly

$$\int \varphi \mathrm{d}\mu_n = \int \varphi \mathrm{d}\mu + \langle \nabla\varphi, \Psi_n \rangle_{L^2(\mu)} + O(\varepsilon_n^2). \tag{173}$$

Subtracting these two equations, dividing by $\varepsilon_n$ and passing to the limit, we find

$$\langle \nabla\varphi, \Phi - \Psi \rangle_{L^2(\mu)} = 0 \tag{174}$$

and this holds for any $\varphi \in C_c^\infty(\mathcal{M})$. However, by assumption, $\xi \in T_\mu \mathcal{P}_2(\mathcal{M}) = \overline{\{\nabla\varphi, \varphi \in C_c^\infty(\mathcal{M})\}}^{L^2(\mu, T\mathcal{M})}$. This implies immediately that $\langle \xi, \Phi - \Psi \rangle_{L^2(\mu)} = 0$, which contradicts (170). $\qquad\square$

**Lemma C.12.** *Let $\varphi \in C_c^\infty(\mathcal{M})$, then, for every $\mu \in \mathcal{P}_2(\mathcal{M})$ and $\gamma \in \mathcal{P}_2(T\mathcal{M})_\mu$,*

$$\left| \int \varphi \, \mathrm{d}(\exp_\# \gamma) - \int \varphi \mathrm{d}\mu - \int \langle \nabla\varphi(x), v \rangle_x \mathrm{d}\gamma(x, v) \right| \leq \frac{1}{2}L\|\gamma\|_\mu^2 \tag{175}$$

*where $L := \max_{(x,v) \in T\mathcal{M}, \|v\|_x = 1} \|\mathrm{Hess}_{\mathcal{M}} \varphi(x)[v]\| < \infty$.*

*Proof.* Let $(x, v) \in T\mathcal{M}$. Applying (Boumal, 2023, Exercise 5.40) to the geodesic given by $c(t) = \exp_x(tv)$, it ensues that there exists $t \in (0, 1)$ such that

$$\varphi(\exp_x(v)) = \varphi(x) + \langle \nabla\varphi(x), v \rangle_x + \frac{1}{2}\langle \mathrm{Hess}\,\varphi(c(t))[c'(t)], c'(t) \rangle_{c(t)} \tag{176}$$

so that, since $\|c'(t)\|_{c(t)} = \|v\|_x$,

$$|\varphi(\exp_x(v)) - \varphi(x) - \langle \nabla\varphi(x), v \rangle_x| \leq \frac{1}{2}L\|v\|_x^2. \tag{177}$$

This immediately implies

$$\left| \int \varphi \, \mathrm{d}(\exp_\# \gamma) - \int \varphi \mathrm{d}\mu - \int \langle \nabla\varphi(x), v \rangle \mathrm{d}\gamma(x, v) \right| = \left| \int \varphi(\exp_x(v)) - \varphi(x) - \langle \nabla\varphi(x), v \rangle \mathrm{d}\gamma(x, v) \right| \tag{178}$$

$$\leq \frac{1}{2}L \int \|v\|_x^2 \mathrm{d}\gamma(x, v) = \frac{1}{2}L\|\gamma\|_\mu^2. \tag{179}$$

$\qquad\square$

## C.9. Proof of Proposition A.6

Let $\nu, \mu \in \mathcal{P}_2(\mathcal{M})$, and $\gamma \in \exp_\mu^{-1}(\nu)$. For any $x \in \mathcal{M}$, $v \in T_x\mathcal{M}$, let us note $c_{x,v}(t) = \exp_x(tv)$ the geodesic starting from $x$ with direction $v$. By (Boumal, 2023, Exercise 5.40), we have that there exists $t \in [0,1]$ such that

$$V\big(\exp_x(v)\big) = V(x) + \langle \nabla_\mathcal{M} V(x), v \rangle_x + \frac{1}{2} \langle \mathrm{Hess}V\big(c_{x,v}(t)\big)[c'_{x,v}(t)], c'_{x,v}(t) \rangle_{c_{x,v}(t)}. \tag{180}$$

Then,

$$\begin{aligned}
\mathcal{V}(\nu) - \mathcal{V}(\mu) &= \int \big(V(\exp_x(v)) - V(x)\big) \, \mathrm{d}\gamma(x,v) \\
&= \int \langle \nabla_\mathcal{M} V(x), v \rangle_x + \frac{1}{2} \langle \mathrm{Hess}V\big(c_{x,v}(t)\big)[c'_{x,v}(t)], c'_{x,v}(t) \rangle_{c_{x,v}(t)} \, \mathrm{d}\gamma(x,v) \\
&= \int \langle \nabla_\mathcal{M} V(x), v \rangle_x \, \mathrm{d}\gamma(x,v) + \frac{1}{2} \int \langle \mathrm{Hess}V\big(c_{x,v}(t)\big)[c'_{x,v}(t)], c'_{x,v}(t) \rangle_{c_{x,v}(t)} \, \mathrm{d}\gamma(x,v).
\end{aligned} \tag{181}$$

Moreover, using that $V$ has bounded Hessian and that geodesics are constant speed and thus satisfy $\|c'_{x,v}(t)\|_{c_{x,v}(t)} = \|c'_{x,v}(0)\|_{c_{x,v}(0)} = \|v\|_x$ (Lee, 2006, Lemma 5.5), we have that the last term is bounded by $\mathrm{W}_2^2(\mu, \nu)$ as

$$\begin{aligned}
\left| \int \langle \mathrm{Hess}V\big(c_{x,v}(t)\big)[c'_{x,v}(t)], c'_{x,v}(t) \rangle_{c_{x,v}(t)} \, \mathrm{d}\gamma(x,v) \right| &\leq L \int \|c'_{x,v}(t)\|_{c_{x,v}(t)}^2 \, \mathrm{d}\gamma(x,v) \\
&= L \int \|v\|_x^2 \, \mathrm{d}\gamma(x,v) = L\mathrm{W}_2^2(\mu,\nu).
\end{aligned} \tag{182}$$

Thus, we conclude

$$\mathcal{V}(\nu) = \mathcal{V}(\mu) + \int \langle \nabla_\mathcal{M} V(x), v \rangle_x \, \mathrm{d}\gamma(x,v) + o\big(\mathrm{W}_2(\mu,\nu)\big). \tag{183}$$

Now, let us verify that $\nabla_\mathcal{M} V \in L^2(\mu)$. We denote by $\mathrm{PT}_{x \to y}$ the parallel transport between $T_x\mathcal{M}$ and $T_y\mathcal{M}$ along the geodesic between $x$ and $y$ (see (Boumal, 2023, Definition 10.35) for the definition). By (Boumal, 2023, Corollary 10.48, 3.), $V$ having its Hessian bounded in operator norm by $L$ is equivalent with having for all $x, y \in \mathcal{M}$, for all $(x,v) \in T\mathcal{M}$,

$$\|\nabla_\mathcal{M} V(x) - \mathrm{PT}_{\exp_x(v) \to x} \nabla_\mathcal{M} V\big(\exp_x(v)\big)\|_x \leq L\|v\|_x. \tag{184}$$

Thus, let $\mu \in \mathcal{P}_2(\mathcal{M})$, $o$ some origin, and $\gamma \in \exp_\mu^{-1}(\delta_o)$. Then, we have by using sequentially the definition of $\gamma$, $\|x+y\|^2 \leq 2\|x\|^2 + 2\|y\|^2$, (184), that for any $(x,v) \in \mathrm{supp}(\gamma)$, $\exp_x(v) = o$ and $\mathrm{PT}_{o \to x}$ is an isometry (Boumal, 2023, Proposition 10.36), and $\gamma \in \mathcal{P}_2(T\mathcal{M})$,

$$\begin{aligned}
\|\nabla_\mathcal{M} V\|_{L^2(\mu)}^2 &= \int \|\nabla_\mathcal{M} V(x)\|_x^2 \, \mathrm{d}\mu(x) \\
&= \int \|\nabla_\mathcal{M} V(x)\|_x^2 \, \mathrm{d}\gamma(x,v) \\
&\leq 2 \int \|\nabla_\mathcal{M} V(x) - \mathrm{PT}_{\exp_x(v) \to x} \nabla_\mathcal{M} V\big(\exp_x(v)\big)\|_x^2 \, \mathrm{d}\gamma(x,v) \\
&\quad + 2 \int \|\mathrm{PT}_{\exp_x(v) \to x} \nabla_\mathcal{M} V\big(\exp_x(v)\big)\|_x^2 \, \mathrm{d}\gamma(x,v) \\
&\leq 2 \int L\|v\|_x^2 \, \mathrm{d}\gamma(x,v) + 2\|\nabla_\mathcal{M} V(o)\|_o^2 \\
&< +\infty.
\end{aligned} \tag{185}$$

Therefore, we can conclude that $\nabla_{\mathrm{W}_2} \mathcal{V}(\mu) = \nabla_\mathcal{M} V$ by Definition A.3. $\qquad \square$

## C.10. Proof of Proposition A.7

Let $\nu, \mu \in \mathcal{P}_2(\mathcal{M})$, and $\gamma \in \exp_\mu^{-1}(\nu)$. First, we recall that the product space $\mathcal{M} \times \mathcal{M}$ is a Riemannian manifold with tangent space $T\mathcal{M} \times T\mathcal{M}$. For any $(x, v), (x', v') \in T\mathcal{M}$ and note $c_x(t) = \exp_x(tv)$, $c_{x',v'}(t) = \exp_{x'}(tv')$ and $c_{x,x',v,v'}(t) = (c_{x,v}(t), c_{x',v'}(t))$ the geodesics starting at $(x, x')$ with direction $(v, v')$. Then, by (Boumal, 2023, Exercise 5.40), there exists $t \in ]0, 1[$ such that

$$
\begin{aligned}
W\left(\exp_x(v), \exp_{x'}(v')\right) = {} & W(x, x') + \langle \nabla_1 W(x, x'), v \rangle_x + \langle \nabla_2 W(x, x'), v' \rangle_{x'} \\
& + \frac{1}{2} \langle \mathrm{Hess}W(c_{x,v}(t), c_{x',v'}(t))[c'_{x,v}(t), c'_{x',v'}(t)], [c'_{x,v}(t), c'_{x',v'}(t)] \rangle_{c_{x,x',v,v'}(t)}.
\end{aligned}
\tag{186}
$$

Moreover, by the same argument as (182) in Proposition A.6, we have

$$
\left| \iint \langle \mathrm{Hess}W(c_{x,v,x',v'}(t))[c'_{x,v}(t), c'_{x',v'}(t)], [c'_{x,v}(t), c'_{x',v'}(t)] \rangle_{c_{x,x',v,v'}(t)} \, \mathrm{d}\gamma(x, v)\mathrm{d}\gamma(x', v') \right| \leq 2LW_2^2(\mu, \nu).
\tag{187}
$$

Then, we have

$$
\begin{aligned}
\mathcal{W}(\nu) - \mathcal{W}(\mu) = {} & \iint W(y, y') \, \mathrm{d}\nu(y)\mathrm{d}\nu(y') - \iint W(x, x') \, \mathrm{d}\mu(x)\mathrm{d}\mu(x') \\
= {} & \iint \left( W(\exp_x(v), \exp_{x'}(v')) - W(x, x') \right) \mathrm{d}\gamma(x, v)\mathrm{d}\gamma(x', v') \\
= {} & \int \left( \langle \nabla_1 W(x, x'), v \rangle_x + \langle \nabla_2 W(x, x'), v' \rangle_{x'} \right. \\
& \left. + \frac{1}{2} \langle \mathrm{Hess}W(c_{x,v}(t), c_{x',v'}(t))[c'_{x,v}(t), c_{x',v'}(t)], [c'_{x,v}(t), c_{x',v'}(t')] \rangle_{c_{x,x',v,v'}(t)} \right) \mathrm{d}\gamma(x, v)\mathrm{d}\gamma(x', v') \\
= {} & \int \langle \int \nabla_1 W(x, x')\mathrm{d}\mu(x'), v \rangle_x \, \mathrm{d}\gamma(x, v) + \int \langle \int \nabla_2 W(x, x')\mathrm{d}\mu(x), v' \rangle_{x'} \, \mathrm{d}\gamma(x', v') + o\left(\mathrm{W}_2(\mu, \nu)\right) \\
= {} & \int \left\langle \int \left( \nabla_1 W(x, x') + \nabla_2 W(x', x) \right) \mathrm{d}\mu(x'), v \right\rangle_x \, \mathrm{d}\gamma(x, v) + o\left(\mathrm{W}_2(\mu, \nu)\right).
\end{aligned}
\tag{188}
$$

Now, let $\nabla_{\mathrm{W}_2} \mathcal{W}(\mu) = \int \left( \nabla_1 W(\cdot, x) + \nabla_2(x, \cdot) \right) \mathrm{d}\mu(x)$. Using that by Jensen's inequality,

$$
\begin{aligned}
\|\nabla_{\mathrm{W}_2} \mathcal{W}\|_{L^2(\mu)}^2 = {} & \int \left\| \int \left( \nabla_1 W(x, x') + \nabla_2 W(x', x) \right) \mathrm{d}\mu(x') \right\|_x^2 \, \mathrm{d}\mu(x) \\
\leq {} & 2 \iint \left( \|\nabla_1 W(x, x')\|_x^2 + \|\nabla_2 W(x, x')\|_{x'}^2 \right) \mathrm{d}\mu(x)\mathrm{d}\mu(x'),
\end{aligned}
\tag{189}
$$

and a similar reasoning of (185), we find that $\nabla_{\mathrm{W}_2} \mathcal{W} \in L^2(\mu)$, and we can conclude that $\nabla_{\mathrm{W}_2} \mathcal{W}$ is a Wasserstein gradient by Definition A.3. $\qquad \square$

## C.11. Proof of Proposition A.9

Let $\nu \in \mathcal{P}_2(\mathbb{R}^d)$ and $\mu_n = \frac{1}{n} \sum_{i=1}^n \delta_{x_i}$. Since $\mathcal{F}$ is Wasserstein differentiable, the Wasserstein gradient $\nabla_{\mathrm{W}_2} \mathcal{F}(\mu_n)$ satisfies for any coupling $\gamma \in \Pi(\mu_n, \nu)$ (Lanzetti et al., 2025, Proposition 2.12),

$$
\mathcal{F}(\nu) = \mathcal{F}(\mu_n) + \int \langle \nabla_{\mathrm{W}_2} \mathcal{F}(\mu_n)(x), y - x \rangle \, \mathrm{d}\gamma(x, y) + o\left( \sqrt{\int \|x - y\|_2^2 \, \mathrm{d}\gamma(x, y)} \right).
\tag{190}
$$

Let $h_1, \ldots, h_n \in \mathbb{R}^d$, $\nu_n = \frac{1}{n} \sum_{i=1}^n \delta_{x_i + h_i}$ and $\gamma_n = \frac{1}{n} \sum_{i=1}^n \delta_{(x_i, x_i + h_i)} \in \Pi(\mu_n, \nu_n)$. Then, since $F(x_1, \ldots, x_n) = \mathcal{F}(\mu_n)$ and $F(x_1 + h_1, \ldots, x_n + h_n) = \mathcal{F}(\nu_n)$, we get

$$
\begin{aligned}
F(x_1 + h_1, \ldots, x_n + h_n) &= \mathcal{F}(\nu_n) \\
&= \mathcal{F}(\mu_n) + \int \langle \nabla_{W_2} \mathcal{F}(\mu_n)(x), y - x \rangle \, \mathrm{d}\gamma_n(x, y) + o\left( \sqrt{\int \|x - y\|_2^2 \, \mathrm{d}\gamma_n(x, y)} \right) \\
&= \mathcal{F}(\mu_n) + \frac{1}{n} \sum_{i=1}^n \langle \nabla_{W_2} \mathcal{F}(\mu_n)(x_i), h_i \rangle + o\left( \sqrt{\sum_{i=1}^n \|h_i\|_2^2} \right) \\
&= F(x_1, \ldots, x_n) + \sum_{i=1}^n \langle \frac{1}{n} \nabla_{W_2} \mathcal{F}(\mu_n)(x_i), h_i \rangle + o\left( \sqrt{\sum_{i=1}^n \|h_i\|_2^2} \right).
\end{aligned}
\tag{191}
$$

Thus, by definition of the gradient of $F$, we deduce that $\nabla_i F(x_1, \ldots, x_n) = \frac{1}{n} \nabla_{W_2} \mathcal{F}(\mu_n)(x_i)$. $\qquad \square$

### C.12. Proof of Proposition B.4

Let $\Gamma \in \exp_{\mathbb{P}}^{-1}(\mathbb{Q})$. Let $s, t \in [0, 1]$, and $\phi^s(\gamma) = (\exp_{\pi_{\mathcal{M}}} \circ (s\pi^v))_\# \gamma$ for $\gamma \in \mathcal{P}_2(T\mathcal{M})$. Then, $(\phi^s, \phi^t)_\# \Gamma \in \Pi(\mathbb{P}_s, \mathbb{P}_t)$. Therefore,

$$
W_{W_2}(\mathbb{P}_s, \mathbb{P}_t)^2 \leq \int W_2^2 \big( \phi^s(\gamma), \phi^t(\gamma) \big) \, \mathrm{d}\Gamma(\gamma). \tag{192}
$$

Moreover, since for $\Gamma$-a.e. $\gamma$, $\big( \exp_{\pi_{\mathcal{M}}} \circ (s\pi^v), \exp_{\pi_{\mathcal{M}}} \circ (t\pi^v) \big)_\# \gamma \in \Pi(\phi^s(\gamma), \phi^t(\gamma))$, we have the following inequality:

$$
\begin{aligned}
W_{W_2}(\mathbb{P}_s, \mathbb{P}_t)^2 &\leq \int W_2^2 \big( \phi^s(\gamma), \phi^t(\gamma) \big) \, \mathrm{d}\Gamma(\gamma) \\
&\leq \iint d\big( \exp_x(sv), \exp_x(tv) \big)^2 \, \mathrm{d}\gamma(x, v) \mathrm{d}\Gamma(\gamma) \\
&= |t - s|^2 \iint \|v\|_x^2 \, \mathrm{d}\gamma(x, v) \mathrm{d}\Gamma(\gamma) \\
&= |t - s|^2 W_{W_2}(\mathbb{P}, \mathbb{Q})^2,
\end{aligned}
\tag{193}
$$

where we used that $\Gamma \in \exp_{\mathbb{P}}^{-1}(\mathbb{Q})$ and that $d\big( \exp_x(tv), \exp_x(sv) \big) = |t - s| \|v\|_x$.

For the other inequality, we have for any $0 \leq s < t \leq 1$, using the triangle inequality and the previous inequality,

$$
\begin{aligned}
W_{W_2}(\mathbb{P}, \mathbb{Q}) &\leq W_{W_2}(\mathbb{P}, \mathbb{P}_s) + W_{W_2}(\mathbb{P}_s, \mathbb{P}_t) + W_{W_2}(\mathbb{P}_t, \mathbb{Q}) \\
&\leq s W_{W_2}(\mathbb{P}, \mathbb{Q}) + W_{W_2}(\mathbb{P}_s, \mathbb{P}_t) + (1 - t) W_{W_2}(\mathbb{P}, \mathbb{Q}).
\end{aligned}
\tag{194}
$$

This is equivalent with

$$
(t - s) W_{W_2}(\mathbb{P}, \mathbb{Q}) \leq W_{W_2}(\mathbb{P}_s, \mathbb{P}_t). \tag{195}
$$

Thus, we can conclude that $W_{W_2}(\mathbb{P}_s, \mathbb{P}_t) = |t - s| W_{W_2}(\mathbb{P}, \mathbb{Q})$ and thus $t \mapsto \mathbb{P}_t$ is a constant-speed geodesic between $\mathbb{P}$ and $\mathbb{Q}$. $\qquad \square$

### C.13. Proof of Proposition B.5

We first state a lemma showing a relation between $\gamma \in \exp_\mu^{-1}(\nu)$ and a specifically constructed $\gamma_t \in \exp_{\mu_\gamma(t)}^{-1}(\nu)$, with $t \mapsto \mu_\gamma(t)$ a geodesic between $\mu$ and $\nu$.

**Lemma C.13.** *Let $\mu, \nu \in \mathcal{P}_2(\mathcal{M})$, $\gamma \in \exp_\mu^{-1}(\nu)$ and the geodesic between $\mu$ and $\nu$ defined for all $t \in [0, 1]$ as $\mu_\gamma(t) = \big( \exp_{\pi_{\mathcal{M}}} \circ (t\pi^v) \big)_\# \gamma$. Let $\gamma_t = \big( \exp_{\pi_{\mathcal{M}}} \circ (t\pi^v), (1 - t) \mathrm{PT}_{\pi_{\mathcal{M}} \to \exp_{\pi_{\mathcal{M}}} \circ (t\pi^v)} \circ \pi^v \big)_\# \gamma$. Then, $\gamma_t \in \exp_{\mu_\gamma(t)}^{-1}(\nu)$, and, for every $s \in [0, 1]$, $\mu_{\gamma_t}(s) = \big( \exp_{\pi_{\mathcal{M}}} \circ (s\pi^v) \big)_\# \gamma_t = \mu_\gamma(t + (1 - t)s)$.*

*Proof.* First, we verify the equality $\mu_{\gamma_t}(s) = \mu_\gamma(t + s(1 - t))$. Fix $s \in [0, 1]$ and let $h : \mathcal{M} \to \mathbb{R}$ be a bounded measurable map. Then,

$$
\begin{aligned}
\int h(y) \, \mathrm{d}(\mu_{\gamma_t}(s))(y) &= \int h\big(\exp_{x_t}(sv_t)\big) \, \mathrm{d}\gamma_t(x_t, v_t) \\
&= \int h\big(\exp_{\exp_x(tv)}\big(s(1 - t)\mathrm{PT}_{x \to \exp_x(tv)}(v)\big)\big) \, \mathrm{d}\gamma(x, v).
\end{aligned}
\tag{196}
$$

Fixing $(x, v) \in T\mathcal{M}$, let $c(t) = \exp_x(tv)$, $t \in [0, 1]$ be the unique geodesic starting from $x$ with $\dot{c}(0) = v$. Then, we have $\mathrm{PT}_{x \to c(t)}(v) = \dot{c}(t)$ by the properties of the parallel transport[3]. Furthermore, by definition of the exponential map, for every $u \in [0, 1]$,

$$
\exp_{\exp_x(tv)}\big(u\mathrm{PT}_{x \to \exp_x(tv)}(v)\big) = \exp_{c(t)}\big(u\dot{c}(t)\big) = c_2(u)
\tag{197}
$$

where $c_2$ is the unique geodesic such that $c_2(0) = c(t)$ and $\dot{c}_2(0) = \dot{c}(t)$. By uniqueness of the geodesics, we thus have $c_2(u) = c(t + u) = \exp_x((t + u)v)$ for every $0 \le u \le 1 - t$. From this, we obtain

$$
\begin{aligned}
\int h(y) \, \mathrm{d}(\mu_{\gamma_t}(s))(y) &= \int h\big(\exp_{\exp_x(tv)}\big(s(1 - t)\mathrm{PT}_{x \to \exp_x(tv)}(v)\big)\big) \, \mathrm{d}\gamma(x, v) \\
&= \int h\big(\exp_x((t + s(1 - t))v)\big) \, \mathrm{d}\gamma(x, v) \\
&= \int h(y) \, \mathrm{d}(\mu_\gamma(t + s(1 - t)))(y),
\end{aligned}
\tag{198}
$$

and thus we have proved $\mu_{\gamma_t}(s) = \mu_\gamma(t + (1 - s)t)$. In particular, we have $\pi^{\mathcal{M}}_\# \gamma_t = \mu_{\gamma_t}(0) = \mu_\gamma(t)$, and $\exp_\# \gamma_t = \mu_{\gamma_t}(1) = \mu_\gamma(1) = \exp_\# \gamma = \nu$, so $\gamma_t$ has the correct marginals. Moreover, it is optimal as

$$
\begin{aligned}
\int \|v_t\|^2_{x_t} \, \mathrm{d}\gamma_t(x_t, v_t) &= \int \|(1 - t)\mathrm{PT}_{x \to \exp_x(tv)}(v)\|^2_{\exp_x(tv)} \, \mathrm{d}\gamma(x, v) \\
&= (1 - t)^2 \int \|v\|^2_x \, \mathrm{d}\gamma(x, v) = (1 - t)^2 \mathrm{W}_2^2(\mu, \nu) = \mathrm{W}_2^2(\mu_\gamma(t), \nu),
\end{aligned}
\tag{199}
$$

where we used in the last line that $\gamma \in \exp_\mu^{-1}(\nu)$ and $\mu_\gamma$ is a geodesic such that $\mu_\gamma(0) = \mu$ and $\mu_\gamma(1) = \nu$, and in particular, $\mathrm{W}_2^2(\mu_\gamma(t), \nu) = \mathrm{W}_2^2\big(\mu_\gamma(t), \mu_\gamma(1)\big) = (1 - t)^2 \mathrm{W}_2^2\big(\mu_\gamma(0), \mu_\gamma(1)\big)$. $\qquad\square$

Now, we state a second lemma providing a Taylor remainder theorem on $\mathcal{P}_2(\mathcal{M})$.

**Lemma C.14.** *Let $\mathcal{F} : \mathcal{P}_2(\mathcal{M}) \to \mathbb{R}$ a twice Wasserstein differentiable functional, $\mu, \nu \in \mathcal{P}_2(\mathcal{M})$ and $\gamma \in \exp_\mu^{-1}(\nu)$, and note $\mu_\gamma : [0, 1] \to \mathcal{M}$ the geodesic between $\mu$ and $\nu$ defined as $\mu_\gamma(t) = \big(\exp_{\pi^{\mathcal{M}}} \circ (t\pi^{\mathrm{v}})\big)_\# \gamma$, and $\gamma_t \in \exp_{\mu_\gamma(t)}^{-1}(\nu)$ given by Lemma C.13. Then, there exists $t \in ]0, 1[$ such that*

$$
\mathcal{F}(\nu) = \mathcal{F}(\mu) + \int \langle \nabla_{\mathrm{W}_2} \mathcal{F}(\mu)(x), v\rangle_x \, \mathrm{d}\gamma(x, v) + \frac{1}{2(1 - t)^2} \int \langle \mathrm{H}\mathcal{F}_{\gamma_t}(x_t, v_t), v_t\rangle_{x_t} \, \mathrm{d}\gamma_t(x_t, v_t).
\tag{200}
$$

*Proof.* First, let us note that

$$
\begin{aligned}
\mathcal{F}\big(\mu_\gamma(1)\big) - \mathcal{F}\big(\mu_\gamma(0)\big) &= \int_0^1 \frac{\mathrm{d}}{\mathrm{d}t} \mathcal{F}\big(\mu_\gamma(t)\big) \, \mathrm{d}t \\
&= \frac{\mathrm{d}}{\mathrm{d}t} \mathcal{F}\big(\mu_\gamma(t)\big)\big|_{t=0} + \int_0^1 \left(\frac{\mathrm{d}}{\mathrm{d}t} \mathcal{F}\big(\mu_\gamma(t)\big) - \frac{\mathrm{d}}{\mathrm{d}t} \mathcal{F}\big(\mu_\gamma(t)\big)\big|_{t=0}\right) \mathrm{d}t \\
&= \frac{\mathrm{d}}{\mathrm{d}t} \mathcal{F}\big(\mu_\gamma(t)\big)\big|_{t=0} + \int_0^1 \int_0^t \frac{\mathrm{d}^2}{\mathrm{d}s^2} \mathcal{F}\big(\mu_\gamma(s)\big) \, \mathrm{d}s\mathrm{d}t \\
&= \frac{\mathrm{d}}{\mathrm{d}t} \mathcal{F}\big(\mu_\gamma(t)\big)\big|_{t=0} + \int_0^1 (1 - s)\frac{\mathrm{d}^2}{\mathrm{d}s^2} \mathcal{F}\big(\mu_\gamma(s)\big) \, \mathrm{d}s.
\end{aligned}
\tag{201}
$$

---

[3]Recall that a vector field $X$ along a smooth curve $c$ is said to be parallel if $D_t X = 0$, where $D_t$ is the covariant derivative along $c$, and that for every $s, t$, the parallel transport operator $\mathrm{PT}_{c(t) \to c(s)}$ sends every $v \in T_{c(t)}\mathcal{M}$ to $X(s)$ where $X$ is the unique parallel vector field along $c$ such that $X(t) = v$. Then, since the condition for $c$ to be a geodesic is that $D_t \dot{c} = 0$, if $c$ is a geodesic, we have $\mathrm{PT}_{c(t) \to c(s)}\dot{c}(t) = \dot{c}(s)$ for every $s, t$.

For the first term, we get by the chain rule (see (30)) $\frac{d}{dt}\mathcal{F}(\mu_\gamma(t))\big|_{t=0} = \int \langle \nabla_{W_2}\mathcal{F}(\mu)(x), v \rangle_x \, d\gamma(x,v)$.

For the second term, using the mean value theorem (since $s \mapsto \frac{d^2}{ds^2}\mathcal{F}(\mu_\gamma(s))$ is continuous, and $1 - s \geq 0$ for all $s \in [0,1]$), there exists $t \in ]0,1[$ such that

$$\int_0^1 (1-s)\frac{d^2}{ds^2}\mathcal{F}(\mu_\gamma(s)) \, ds = \frac{d^2}{dt^2}\mathcal{F}(\mu_\gamma(t)) \int_0^1 (1-s)ds = \frac{1}{2}\frac{d^2}{dt^2}\mathcal{F}(\mu_\gamma(t)). \tag{202}$$

Since, by Lemma C.13, we have $\mu_{\gamma_t}(s) = \mu_\gamma(t + s(1-t))$ for every $s \in [0,1]$, we have by Definition A.8

$$\int \langle H\mathcal{F}_{\gamma_t}(x_t, v_t), v_t \rangle_{x_t} d\gamma_t(x_t, v_t) = \frac{d^2}{ds^2}\mathcal{F}(\mu_{\gamma_t}(s))\big|_{s=0} \tag{203}$$

$$= \frac{d^2}{ds^2}\mathcal{F}(\mu_\gamma(t + (1-t)s))\big|_{s=0} \tag{204}$$

$$= (1-t)^2 \frac{d^2}{ds^2}\mathcal{F}(\mu_\gamma(s))\big|_{s=t}. \tag{205}$$

This finishes the proof. $\qquad\square$

Let $\mathbb{P}, \mathbb{Q} \in \mathcal{P}_2(\mathcal{P}_2(\mathcal{M}))$. Let $\mathcal{F} : \mathcal{P}_2(\mathcal{M}) \to \mathbb{R}$ a Wasserstein differentiable functional, $\mathbb{F}(\mathbb{P}) = \int \mathcal{F}(\mu) \, d\mathbb{P}(\mu)$ and $\Gamma \in \exp_{\mathbb{P}}^{-1}(\mathbb{Q})$. Let $\gamma$ in the support of $\Gamma$, then we know that ($\Gamma$-almost surely), $\gamma \in \exp_\mu^{-1}(\nu)$ where $\mu = \pi_\#^{\mathcal{M}}\gamma$ and $\nu = \exp_\# \gamma$. In particular, by Lemma C.14, there exists some $t \in ]0,1[$ and $\gamma_t \in \exp_{\mu_\gamma(t)}^{-1}(\nu)$ such that

$$\mathcal{F}(\nu) = \mathcal{F}(\mu) + \int \langle \nabla_{W_2}\mathcal{F}(\mu)(x), v \rangle_x \, d\gamma(x,v) + \frac{1}{2(1-t)^2} \int \langle H\mathcal{F}_{\gamma_t}(x_t, v_t), v_t \rangle_{x_t} \, d\gamma_t(x_t, v_t), \tag{206}$$

so that, by the assumption on the Hessian of $\mathcal{F}$,

$$\left| \mathcal{F}(\nu) - \mathcal{F}(\mu) - \int \langle \nabla_{W_2}\mathcal{F}(\mu)(x), v \rangle_x \, d\gamma(x,v) \right| \leq \frac{1}{2(1-t)^2} \int |\langle H\mathcal{F}_{\gamma_t}(x_t, v_t), v_t \rangle_{x_t}| \, d\gamma_t(x_t, v_t) \tag{207}$$

$$\leq \frac{1}{2(1-t)^2}L \int \|v_t\|_{x_t}^2 \, d\gamma_t(x_t, x_t) \tag{208}$$

$$\leq \frac{1}{2(1-t)^2}LW_2^2(\mu_\gamma(t), \nu) = \frac{1}{2}LW_2^2(\mu, \nu) \tag{209}$$

$$\leq \frac{L}{2} \int \|v\|_x^2 \, d\gamma(x,v). \tag{210}$$

From this, we deduce that

$$\left| \mathbb{F}(\mathbb{Q}) - \mathbb{F}(\mathbb{P}) - \iint \langle \nabla_{W_2}\mathcal{F}(\pi_\#^{\mathcal{M}}\gamma)(x), v \rangle_x d\gamma(x,v)d\Gamma(\gamma) \right| \tag{211}$$

$$= \left| \int \left( \mathcal{F}(\exp_\# \gamma) - \mathcal{F}(\pi_\#^{\mathcal{M}}\gamma) - \int \langle \nabla_{W_2}\mathcal{F}(\pi_\#^{\mathcal{M}}\gamma)(x), v \rangle_x d\gamma(x,v) \right) d\Gamma(\gamma) \right| \tag{212}$$

$$\leq \frac{L}{2} \iint \|v\|_x^2 d\gamma(x,v)d\Gamma(\gamma) = \frac{L}{2}W_{W_2}(\mathbb{P}, \mathbb{Q})^2. \tag{213}$$

Thus, we can conclude that

$$\mathbb{F}(\mathbb{Q}) = \mathbb{F}(\mathbb{P}) + \iint \langle \nabla_{W_2}\mathcal{F}(\pi_\#^{\mathcal{M}}\gamma)(x), v \rangle_x \, d\gamma(x,v)d\Gamma(\gamma) + o(W_{W_2}(\mathbb{P}, \mathbb{Q})). \tag{214}$$

Moreover, as we assumed $\mathcal{M}$ compact, $\nabla_{W_2}\mathcal{F}(\mu)$ is bounded for any $\mu$ and thus $\int \|\nabla_{W_2}\mathcal{F}(\mu)\|_{L^2(\mu)}^2 \, d\mathbb{P}(\mu) < +\infty$. Therefore, by Definition 3.3, $\nabla_{W_{W_2}}\mathbb{F}(\mathbb{P}) = \nabla_{W_2}\mathcal{F} \in L^2(\mathbb{P})$. $\qquad\square$

## C.14. Proof of Proposition B.6

Let $\mathbb{P}, \mathbb{Q} \in \mathcal{P}_2\big(\mathcal{P}_2(\mathcal{M})\big)$. Let $\mathcal{W} : \mathcal{P}_2(\mathcal{M}) \times \mathcal{P}_2(\mathcal{M}) \to \mathbb{R}$ a Wasserstein differentiable functional, $\mathbb{W}(\mathbb{P}) = \iint \mathcal{W}(\mu, \nu)\, \mathrm{d}\mathbb{P}(\mu)\mathrm{d}\mathbb{P}(\nu)$ and $\Gamma \in \exp_{\mathbb{P}}^{-1}(\mathbb{Q})$. Let $\gamma$ and $\gamma'$ be in the support of $\Gamma$, then $\gamma \in \exp_\mu^{-1}(\nu)$ and $\gamma' \in \exp_{\mu'}^{-1}(\nu')$ where $\mu = \pi_\#^{\mathcal{M}} \gamma$, $\mu' = \pi_\#^{\mathcal{M}} \gamma'$ and $\nu = \exp_\# \gamma$, $\nu' = \exp_\# \gamma'$. For notation simplicity, we write $\nabla_1$ and $\nabla_2$ instead of $\nabla_{W_2,1}$ and $\nabla_{W_2,2}$. By the remainder Taylor theorem (Lemma C.14) applied on the product space $\mathcal{P}_2(\mathcal{M}) \times \mathcal{P}_2(\mathcal{M})$, we get that there exists $t \in ]0,1[$ such that

$$\mathcal{W}(\nu, \nu') = \mathcal{W}(\mu, \mu') + \int \langle \nabla_1 \mathcal{W}(\mu, \mu')(x), v\rangle_x \, \mathrm{d}\gamma(x,v) + \int \langle \nabla_2 \mathcal{W}(\mu, \mu')(x), v\rangle_x \, \mathrm{d}\gamma'((x,v) + \frac{1}{2}\frac{\mathrm{d}^2}{\mathrm{d}t^2}\mathcal{W}\big(\mu_\gamma(t), \mu_{\gamma'}(t)\big).$$
(215)

The last term is a Hessian term, which can be written as $\frac{\mathrm{d}^2}{\mathrm{d}t^2}\mathcal{W}\big(\mu_\gamma(t), \mu_{\gamma'}(t)\big) = \frac{1}{(1-t)^2}\int \langle H\mathcal{W}_{\gamma_t, \gamma_t'}[(x_t, v_t), (x_t', v_t')], (v_t, v_t')\rangle_{(x_t, x_t')} \, \mathrm{d}\gamma_t(x_t, v_t)\mathrm{d}\gamma_t'(x_t', v_t')$, where we define $H\mathcal{W}_{\gamma, \gamma'} : T\mathcal{M} \times T\mathcal{M} \to T\mathcal{M} \times T\mathcal{M}$ the Hessian operator at $(\gamma, \gamma')$ similarly as in Definition A.8. By the assumption on the Hessian, we thus have

$$\left| \mathcal{W}(\nu, \nu') - \mathcal{W}(\mu, \mu') - \int \langle \nabla_1 \mathcal{W}(\mu, \mu')(x), v\rangle_x \, \mathrm{d}\gamma(x,v) - \int \langle \nabla_2 \mathcal{W}(\mu, \mu')(x'), v'\rangle_{x'} \, \mathrm{d}\gamma'(x', v') \right|$$

$$\le \frac{1}{2(1-t)^2}\int |\langle H\mathcal{W}_{\gamma_t, \gamma_t'}[(x_t, v_t), (x_t', v_t')], (v_t, v_t')\rangle_{(x_t, x_t')}| \, \mathrm{d}\gamma_t(x_t, v_t)\mathrm{d}\gamma_t'(x_t', v_t')$$

$$\le \frac{1}{2(1-t)^2}L\left(\int \|v_t\|_{x_t}^2 \, \mathrm{d}\gamma_t(x_t, v_t) + \int \|v_t'\|_{x_t'}^2 \, \mathrm{d}\gamma_t'(x_t', v_t')\right)$$
(216)

$$= \frac{L}{2(1-t)^2}\big(\mathrm{W}_2^2(\mu_\gamma(t), \nu) + \mathrm{W}_2^2(\mu_\gamma'(t), \nu)\big)$$

$$= \frac{L}{2}\left(\int \|v\|_x^2 \, \mathrm{d}\gamma(x,v) + \int \|v'\|_{x'}^2 \, \mathrm{d}\gamma'(x', v')\right).$$

Then, let us bound

$$\left| \mathbb{W}(\mathbb{P}) - \mathbb{W}(\mathbb{Q}) - \iint \left\langle \int \big(\nabla_1 \mathcal{W}(\phi^{\mathcal{M}}(\gamma), \eta)(x) + \nabla_2 \mathcal{W}(\eta, \phi^{\mathcal{M}}(\gamma))\big)\mathrm{d}\mathbb{P}(\eta), v \right\rangle_x \, \mathrm{d}\gamma(x,v)\mathrm{d}\Gamma(\gamma) \right|$$

$$\le \left| \iint \big(\mathcal{W}(\phi^{\exp}(\gamma), \phi^{\exp}(\gamma')) - \mathcal{W}(\phi^{\mathcal{M}}(\gamma), \phi^{\mathcal{M}}(\gamma'))\big) \, \mathrm{d}\Gamma(\gamma)\mathrm{d}\Gamma(\gamma') \right.$$

$$\quad - \iiint \langle \nabla_1 \mathcal{W}(\phi^{\mathcal{M}}(\gamma), \eta)(x), v\rangle_x \mathrm{d}\mathbb{P}(\eta)\mathrm{d}\gamma(x,v)\mathrm{d}\Gamma(\gamma)$$

$$\quad \left. - \iiint \langle \nabla_2 \mathcal{W}(\eta, \phi^{\mathcal{M}}(\gamma'))(x'), v'\rangle_{x'} \, \mathrm{d}\mathbb{P}(\eta)\mathrm{d}\gamma'(x', v')\mathrm{d}\Gamma(\gamma') \right|$$

$$= \left| \iint \big(\mathcal{W}(\phi^{\exp}(\gamma), \phi^{\exp}(\gamma')) - \mathcal{W}(\phi^{\mathcal{M}}(\gamma), \phi^{\mathcal{M}}(\gamma'))\big) \, \mathrm{d}\Gamma(\gamma)\mathrm{d}\Gamma(\gamma') \right.$$

$$\quad \left. - \iiiint \langle \nabla_1 \mathcal{W}(\phi^{\mathcal{M}}(\gamma), \phi^{\mathcal{M}}(\gamma'))(x), v\rangle_x + \langle \nabla_2 \mathcal{W}(\phi^{\mathcal{M}}(\gamma), \phi^{\mathcal{M}}(\gamma'))(x'), v'\rangle_{x'} \, \mathrm{d}\gamma(x,v)\mathrm{d}\gamma'(x', v')\mathrm{d}\Gamma(\gamma)\mathrm{d}\Gamma(\gamma') \right|$$

$$\le \iint \left| \mathcal{W}(\phi^{\exp}(\gamma), \phi^{\exp}(\gamma')) - \mathcal{W}(\phi^{\mathcal{M}}(\gamma), \phi^{\mathcal{M}}(\gamma')) \right.$$

$$\quad \left. - \int \langle \nabla_1 \mathcal{W}(\phi^{\mathcal{M}}(\gamma), \phi^{\mathcal{M}}(\gamma'))(x), v\rangle_x \, \mathrm{d}\gamma(x,v) - \int \langle \nabla_2 \mathcal{W}(\phi^{\mathcal{M}}(\gamma), \phi^{\mathcal{M}}(\gamma'))(x'), v'\rangle_{x'} \, \mathrm{d}\gamma(x', v') \right| \, \mathrm{d}\Gamma(\gamma)\mathrm{d}\Gamma(\gamma')$$

$$\le \frac{L}{2}\left(\iint \|v\|_x^2 \, \mathrm{d}\gamma(x,v)\mathrm{d}\Gamma(\gamma) + \iint \|v'\|_x^2 \, \mathrm{d}\gamma'(x', v')\mathrm{d}\Gamma(\gamma')\right) \quad \text{by (216)}$$

$$= L\mathrm{W}_{\mathrm{W}_2}^2(\mathbb{P}, \mathbb{Q}) \quad \text{since } \Gamma \in \exp_{\mathbb{P}}^{-1}(\mathbb{Q}).$$
(217)

This allows to conclude by Definition 3.3 that

$$\nabla_{\mathrm{W}_{\mathrm{W}_2}} \mathcal{W}(\mathbb{P})(\mu) = \int \big(\nabla_1 \mathcal{W}(\mu, \nu) + \nabla_2 \mathcal{W}(\nu, \mu)\big) \, \mathrm{d}\mathbb{P}(\nu).$$
(218)

$\square$

## C.15. Proof of Proposition B.7

We note $\mathbb{P} := \mathbb{P}_{\mathbf{x}}$ and $\mu^c = \mu_{\mathbf{x}^c}$ for every $c$. Let $\mathbf{h} \in (\mathbb{R}^d)^{C \times n}$, for every $t \in \mathbb{R}$, we define $\mathbb{P}_t := \mathbb{P}_{\mathbf{x}+t\mathbf{h}}$, and for every $c$, $\mu_t^c := \mu_{\mathbf{x}^c+t\mathbf{h}^c}$, so that $\mathbb{P}_t = \frac{1}{C} \sum_{c=1}^{C} \delta_{\mu_t^c}$. We also consider the transport plan $\gamma_t^c = \frac{1}{n} \sum_{i=1}^{n} \delta_{(x_i^c, th_i^c)}$ (which satisfies $\pi_{\#}^{\mathbb{R}^d} \gamma_t^c = \mu^c$ and $\exp_{\#} \gamma_t = \mu_t^c$), and the plan $\Gamma_t = \frac{1}{C} \sum_{c=1}^{C} \delta_{\gamma_t^c}$ (which satisfies $\phi_{\#}^{\mathbb{R}^d} \Gamma = \mathbb{P}$ and $\phi_{\#}^{\exp} \Gamma = \mathbb{P}_t$).

It is not difficult to see that for $t$ small enough, for every $c$, $\gamma_t^c$ is actually optimal between $\mu^c$ and $\mu_t^c$ (that is, $\gamma_t^c \in \exp_{\mu^c}^{-1}(\mu_t^c)$), and therefore

$$\mathrm{W}_2^2(\mu^c, \mu_t^c) = \int \|v\|^2 \mathrm{d}\gamma_t^c(x, v) = \frac{t^2}{n} \sum_{i=1}^{n} \|h_i^c\|^2. \tag{219}$$

Moreover, it is also the case that for $t$ small enough, $\Gamma_t \in \exp_{\mathbb{P}}^{-1}(\mathbb{P}_t)$. Indeed, since for every $c, c'$, $\mathrm{W}_2^2(\mu^c, \mu_t^{c'}) \xrightarrow{t \to 0} \mathrm{W}_2^2(\mu^c, \mu^{c'})$ which is zero if and only if $c = c'$, it ensues that for $t$ small enough, for every $c$,

$$\mathrm{W}_2^2(\mu^c, \mu_t^c) = \min_{c'} \mathrm{W}_2^2(\mu^c, \mu_t^{c'}). \tag{220}$$

Thus, for any $\Gamma \in \Pi(\mathbb{P}, \mathbb{P}_t)$, represented by the matrix $(\Gamma_{c,c'})_{c,c'=1,\dots,C}$, we have

$$\int \mathrm{W}_2^2(\mu, \nu) \mathrm{d}\Gamma(\mu, \nu) = \sum_{c,c'=1}^{C} \mathrm{W}_2^2(\mu^c, \mu_t^{c'}) \Gamma_{c,c'} \tag{221}$$

$$\geq \sum_{c,c'=1}^{C} \mathrm{W}_2^2(\mu^c, \mu_t^c) \Gamma_{c,c'} \tag{222}$$

$$= \frac{1}{C} \sum_{c=1}^{C} \mathrm{W}_2^2(\mu^c, \mu_t^c) \tag{223}$$

$$= \frac{t^2}{Cn} \sum_{c=1}^{C} \sum_{i=1}^{n} \|h_i^c\|^2 = \iint \|v\|^2 \mathrm{d}\gamma(x, v) \mathrm{d}\Gamma_t(\gamma), \tag{224}$$

so that, by taking the minimum over $\Gamma$, we find $\iint \|v\|^2 \mathrm{d}\gamma(x, v) \mathrm{d}\Gamma_t(\gamma) \leq \mathrm{W}_{\mathrm{W}_2}(\mathbb{P}, \mathbb{P}_t)^2$. Since the reverse inequality always hold, we find that

$$\mathrm{W}_{\mathrm{W}_2}(\mathbb{P}, \mathbb{P}_t)^2 = \iint \|v\|^2 \mathrm{d}\gamma(x, v) \mathrm{d}\Gamma_t(\gamma) = \frac{t^2}{Cn} \sum_{c=1}^{C} \sum_{i=1}^{n} \|h_i^c\|^2, \tag{225}$$

and we conclude that $\Gamma_t$ is optimal, with $\mathrm{W}_{\mathrm{W}_2}(\mathbb{P}, \mathbb{P}_t) = O(t)$. Plugging $\Gamma_t$ into the definition of the WoW gradient, we find that

$$\mathbb{F}(\mathbb{P}_t) = \mathbb{F}(\mathbb{P}) + \iint \langle \nabla_{\mathrm{W}_{\mathrm{W}_2}} \mathbb{F}(\mathbb{P})(\mu)(x), x \rangle \mathrm{d}\gamma(x, v) \mathrm{d}\Gamma_t(x, v) + o(t), \tag{226}$$

that is (since $\mathbf{x} + t\mathbf{h} \in X$ for $t$ small enough),

$$F(\mathbf{x} + t\mathbf{h}) = F(\mathbf{x}) + \frac{1}{nC} \sum_{c=1}^{C} \sum_{i=1}^{n} \langle \nabla_{\mathrm{W}_{\mathrm{W}_2}} \mathbb{F}(\mathbb{P})(\mu^c)(x_i^c), h_i^c \rangle + o(t). \tag{227}$$

From the definition of the gradient, we deduce that for every $c, i$,

$$\nabla_{\mathrm{W}_{\mathrm{W}_2}} \mathbb{F}(\mathbb{P})(\mu^c)(x_i^c) = Cn \nabla_{c,i} F(\mathbf{x}). \tag{228}$$

This finishes the proof. $\qquad\square$

### C.16. Proof of Proposition B.8

Let $\mathbb{P}_t = \left(\left((1-t)\mathrm{T}_{\pi^1}^{\pi^2} + t\mathrm{T}_{\pi^1}^{\pi^3}\right)_\# \pi^1\right)_\# \Gamma$ where $\Gamma \in \Pi(\mathbb{P}, \mathbb{Q}, \mathbb{O})$, $\pi_\#^{1,2}\Gamma \in \Pi_o(\mathbb{P}, \mathbb{Q})$ a,d $\pi_\#^{1,3}\Gamma \in \Pi_o(\mathbb{P}, \mathbb{O})$. Then, we have

$$
\begin{aligned}
\mathbb{V}(\mathbb{P}_t) &= \int \mathcal{F}\left(\left((1-t)\mathrm{T}_\eta^\mu + t\mathrm{T}_\eta^\nu\right)_\# \eta\right) \, \mathrm{d}\Gamma(\eta, \mu, \nu) \\
&= \int \mathcal{F}(\mu_t) \, \mathrm{d}\Gamma(\eta, \mu, \nu) \quad \text{for } \mu_t = \exp_\eta\left((1-t)\mathrm{T}_\eta^\mu + t\mathrm{T}_\eta^\nu\right) = \left((1-t)\mathrm{T}_\eta^\mu + t\mathrm{T}_\eta^\nu\right)_\# \eta \\
&\leq (1-t)\int \mathcal{F}(\mu) \, \mathrm{d}\mathbb{Q}(\mu) + t\int \mathcal{F}(\nu) \, \mathrm{d}\mathbb{O}(\nu) - \frac{\lambda t(1-t)}{2}\int \mathrm{W}_2^2(\mu, \nu) \, \mathrm{d}\Gamma(\eta, \mu, \nu) \\
&\leq (1-t)\mathbb{V}(\mathbb{Q}) + t\mathbb{V}(\mathbb{O}) - \frac{\lambda t(1-t)}{2}\mathrm{W}_{\mathrm{W}_2}(\mathbb{Q}, \mathbb{O})^2,
\end{aligned}
\tag{229}
$$

where we used in the last two lines that $\mathcal{F}$ is $\lambda$-convex along $t \mapsto \mu_t$, and $\mathrm{W}_{\mathrm{W}_2}(\mathbb{Q}, \mathbb{O})^2 \leq \int \mathrm{W}_2^2(\mu, \nu) \, \mathrm{d}\Gamma(\eta, \mu, \nu)$. $\qquad\square$

### C.17. Proof of Proposition B.9

Let $\mathbb{P}_t = \left(\left((1-t)\mathrm{T}_{\pi^1}^{\pi^2} + t\mathrm{T}_{\pi^1}^{\pi^3}\right)_\# \pi^1\right)_\# \Gamma$ where $\Gamma \in \Pi(\mathbb{P}, \mathbb{Q}, \mathbb{O})$, $\pi_\#^{1,2}\Gamma \in \Pi_o(\mathbb{P}, \mathbb{Q})$ a,d $\pi_\#^{1,3}\Gamma \in \Pi_o(\mathbb{P}, \mathbb{O})$. Then, we have

$$
\begin{aligned}
\mathbb{W}(\mathbb{P}_t) &= \frac{1}{2}\iint \mathcal{W}\left(\left((1-t)\mathrm{T}_\eta^\mu + t\mathrm{T}_\eta^\nu\right)_\# \eta, \left((1-t)\mathrm{T}_{\eta'}^{\mu'} + t\mathrm{T}_{\eta'}^{\nu'}\right)_\# \eta'\right) \, \mathrm{d}\Gamma(\eta, \mu, \nu)\mathrm{d}\Gamma(\eta', \mu', \nu') \\
&\leq (1-t)\frac{1}{2}\iint \mathcal{W}(\mu, \mu')\mathrm{d}\mathbb{Q}(\mu)\mathrm{d}\mathbb{Q}(\mu') + t\frac{1}{2}\iint \mathcal{W}(\nu, \nu') \, \mathrm{d}\mathbb{O}(\nu)\mathrm{d}\mathbb{O}(\nu') \\
&= (1-t)\mathbb{W}(\mathbb{Q}) + t\mathbb{W}(\mathbb{O}).
\end{aligned}
\tag{230}
$$

$\qquad\square$

### C.18. Proof of Proposition B.10

Let $\mathbb{P}, \mathbb{Q}, \mathbb{O} \in \mathcal{P}_2\left(\mathcal{P}_{2,\mathrm{ac}}(\mathbb{R}^d)\right)$. Define the generalized geodesic $t \mapsto \mathbb{P}_t = \left(\left((1-t)\mathrm{T}_{\pi^1}^{\pi^2} + t\mathrm{T}_{\pi^1}^{\pi^3}\right)_\# \pi^1\right)_\# \Gamma$ where $\Gamma \in \Pi(\mathbb{P}, \mathbb{Q}, \mathbb{O})$, $\pi_\#^{1,2}\Gamma \in \Pi_o(\mathbb{P}, \mathbb{Q})$ and $\pi_\#^{1,3}\Gamma \in \Pi_o(\mathbb{P}, \mathbb{O})$. Let us show that $\mathbb{F} : \mathbb{Q} \mapsto \frac{1}{2}\mathrm{W}_{\mathrm{W}_2}(\mathbb{Q}, \mathbb{P})^2$ is convex along this curve.

To do this, first note that $\tilde{\Gamma} = \left(\pi^1, ((1-t)\mathrm{T}_{\pi^1}^{\pi^2} + t\mathrm{T}_{\pi^1}^{\pi^3})_\# \pi^1\right)_\# \Gamma \in \Pi(\mathbb{P}, \mathbb{P}_t)$. Then, we have

$$
\begin{aligned}
\mathbb{F}(\mathbb{P}_t) &= \frac{1}{2}\mathrm{W}_{\mathrm{W}_2}(\mathbb{P}_t, \mathbb{P})^2 \\
&\leq \frac{1}{2}\int \mathrm{W}_2^2\left(\mu, ((1-t)\mathrm{T}_\mu^\nu + t\mathrm{T}_\mu^\eta)_\# \mu\right) \, \mathrm{d}\Gamma(\mu, \nu, \eta).
\end{aligned}
\tag{231}
$$

Note that $\mathrm{T} = (1-t)\mathrm{T}_{\pi^1}^{\pi^2} + t\mathrm{T}_{\pi^1}^{\pi^3}$ is an OT map by Brenier's theorem since it is the gradient of a convex function (as a nonnegative weighted sum of convex functions). Thus, for $\Gamma$-almost every $(\mu, \nu, \eta)$, $\mathrm{W}_2^2\left(\mu, ((1-t)\mathrm{T}_\mu^\nu + t\mathrm{T}_\mu^\eta)_\# \mu\right) = \|(1-t)\mathrm{T}_\mu^\nu + t\mathrm{T}_\mu^\eta - \mathrm{Id}\|_{L^2(\mu)}^2$. Then, applying the parallelogram identity on the Hilbert space $L^2(\mu)$, we get

$$
\begin{aligned}
\mathbb{F}(\mathbb{P}_t) &\leq \frac{1}{2}\int \|(1-t)\mathrm{T}_\mu^\nu + t\mathrm{T}_\mu^\eta - \mathrm{Id}\|_{L^2(\mu)}^2 \, \mathrm{d}\Gamma(\mu, \nu, \eta) \\
&= \frac{1}{2}\int \|(1-t)(\mathrm{T}_\mu^\nu - \mathrm{Id}) + t(\mathrm{T}_\mu^\eta - \mathrm{Id})\|_{L^2(\mu)}^2 \, \mathrm{d}\Gamma(\mu, \nu, \eta) \\
&= \frac{(1-t)}{2}\int \|\mathrm{T}_\mu^\nu - \mathrm{Id}\|_{L^2(\mu)}^2 \, \mathrm{d}\Gamma(\mu, \nu, \eta) + \frac{t}{2}\int \|\mathrm{T}_\mu^\eta - \mathrm{Id}\|_{L^2(\mu)}^2 \, \mathrm{d}\Gamma(\mu, \nu, \eta) \\
&\quad - \frac{t(1-t)}{2}\int \|\mathrm{T}_\mu^\nu - \mathrm{T}_\mu^\eta\|_{L^2(\mu)}^2 \, \mathrm{d}\Gamma(\mu, \nu, \eta) \\
&= \frac{(1-t)}{2}\mathrm{W}_{\mathrm{W}_2}(\mathbb{P}, \mathbb{Q})^2 + \frac{t}{2}\mathrm{W}_{\mathrm{W}_2}(\mathbb{P}, \mathbb{O})^2 - \frac{t(1-t)}{2}\int \|\mathrm{T}_\mu^\nu - \mathrm{T}_\mu^\eta\|_{L^2(\mu)}^2 \, \mathrm{d}\Gamma(\mu, \nu, \eta) \\
&= (1-t)\mathbb{F}(\mathbb{Q}) + t\mathbb{F}(\mathbb{O}) - \frac{t(1-t)}{2}\int \|\mathrm{T}_\mu^\nu - \mathrm{T}_\mu^\eta\|_{L^2(\mu)}^2 \, \mathrm{d}\Gamma(\mu, \nu, \eta).
\end{aligned}
\tag{232}
$$

Finally, since $(T^\nu_\mu, T^\eta_\mu)_{\#}\mu \in \Pi(\nu, \eta)$, we also have $W_2^2(\nu, \eta) \le \|T^\eta_\mu - T^\nu_\mu\|^2_{L^2(\mu)}$. Thus, we have

$$\int \|T^\nu_\mu - T^\eta_\mu\|^2_{L^2(\mu)} \, d\Gamma(\mu, \nu, \eta) \ge \int W_2^2(\nu, \eta) \, d\Gamma(\mu, \nu, \eta) \ge W_{W_2}(\mathbb{Q}, \mathbb{O})^2, \tag{233}$$

where we applied that $\pi^{2,3}_{\#}\Gamma \in \Pi(\mathbb{Q}, \mathbb{O})$ for the last inequality. Plugging this result in (232), we get

$$\mathbb{F}(\mathbb{P}_t) \le (1-t)\mathbb{F}(\mathbb{Q}) + t\mathbb{F}(\mathbb{O}) - \frac{t(1-t)}{2} W_{W_2}(\mathbb{Q}, \mathbb{O})^2. \tag{234}$$

$\square$

## D. Additional Details and Experiments

### D.1. Minimization of the MMD

We want to minimize $\mathbb{F}(\mathbb{P}) = \frac{1}{2}\mathrm{MMD}^2(\mathbb{P}, \mathbb{Q})$ for $\mathbb{P}, \mathbb{Q} \in \mathcal{P}_2(\mathcal{P}_2(\mathbb{R}^d))$ and a kernel $K : \mathcal{P}_2(\mathbb{R}^d) \times \mathcal{P}_2(\mathbb{R}^d) \to \mathbb{R}$. Recall that $\mathbb{F}(\mathbb{P}) = \mathbb{V}(\mathbb{P}) + \mathbb{W}(\mathbb{P}) + \mathrm{cst}$ with $\mathbb{V}(\mathbb{P}) = \int \mathcal{V}(\mu) \, d\mathbb{P}(\mu)$ and $\mathcal{V}(\mu) = -\int K(\mu, \nu) \, d\mathbb{Q}(\nu)$, and $\mathbb{W}(\mathbb{P}) = \frac{1}{2} \iint K(\mu, \nu) \, d\mathbb{P}(\mu)d\mathbb{P}(\nu)$. If $K_\nu(\mu) = K(\mu, \nu)$ is a differentiable functional, then the gradient of $\mathbb{F}$ is given for all $\mathbb{P} \in \mathcal{P}_2(\mathcal{P}_2(\mathbb{R}^d)), \mu \in \mathcal{P}_2(\mathbb{R}^d)$ by

$$\begin{aligned}
\nabla_{W_{W_2}}\mathbb{F}(\mathbb{P})(\mu) &= \nabla_{W_{W_2}}\mathbb{V}(\mathbb{P})(\mu) + \nabla_{W_{W_2}}\mathbb{W}(\mathbb{P})(\mu) \\
&= \nabla_{W_2}\mathcal{V}(\mu) + (\nabla_{W_2}\mathcal{W} * \mathbb{P})(\mu) \\
&= -\int \nabla_{W_2} K_\nu(\mu) \, d\mathbb{Q}(\nu) + \int \nabla_{W_2} K_\nu(\mu) \, d\mathbb{P}(\nu).
\end{aligned} \tag{235}$$

We can choose different kernels, giving different discrepancies. We compare here different kernels based on the Sliced-Wasserstein distance (Rabin et al., 2012). Let $p \ge 1$. We recall that the Sliced-Wasserstein distance is defined between $\mu, \nu \in \mathcal{P}_2(\mathbb{R}^d)$ as

$$\mathrm{SW}_p^p(\mu, \nu) = \int_{S^{d-1}} W_2^p(P^\theta_{\#}\mu, P^\theta_{\#}\nu) \, d\sigma(\theta), \tag{236}$$

where $S^{d-1} = \{\theta \in \mathbb{R}^d, \|\theta\|_2 = 1\}$ is the sphere, $P^\theta(x) = \langle \theta, x \rangle$ and $\sigma$ is the uniform measure on the sphere. The Sliced-Wasserstein distance allows defining a Gaussian positive definite kernel $K(\mu, \nu) = e^{-\frac{1}{2}\mathrm{SW}_2^2(\mu,\nu)/h}$ (Kolouri et al., 2016; Carriere et al., 2017) and a Laplace positive definite kernel $K(\mu, \nu) = e^{-\mathrm{SW}_1(\mu,\nu)/h}$ (Meunier et al., 2022). We also propose in practice to use the Riesz SW kernel $K(\mu, \nu) = -\mathrm{SW}_2(\mu, \nu)^r$ for $r \in (0, 2)$ and inverse multiquadric kernel (IMQ) $K(\mu, \nu) = \frac{1}{\sqrt{c + \mathrm{SW}_2^2(\mu,\nu)}}$. Note however that the Riesz SW kernel is not positive definite (but conditionally positive definite), and that showing that the IMQ kernel is positive definite is an open question.

The Wasserstein gradient of the Sliced-Wasserstein distance $\mathcal{F}(\mu) = \frac{1}{2}\mathrm{SW}_2^2(\mu, \nu)$ can be computed as (Bonnotte, 2013, Proposition 5.1.7)

$$\nabla_{W_2}\mathcal{F}(\mu) = \int_{S^{d-1}} \psi'_\theta(\langle x, \theta \rangle)\theta \, d\sigma(\theta), \tag{237}$$

with $\psi_\theta$ the Kantorovich potential between $P^\theta_{\#}\mu$ and $P^\theta_{\#}\nu$, and thus $\psi'_\theta(u) = u - F^{-1}_{P^\theta_{\#}\nu}(F_{P^\theta_{\#}\mu}(u))$ for all $u \in \mathbb{R}$. For the Gaussian kernel, by the chain rule, we have $\nabla_{W_2} K_\nu(\mu) = -\frac{1}{h}e^{-\frac{1}{2}\mathrm{SW}_2^2(\mu,\nu)/h}\nabla_{W_2}\mathcal{F}(\mu)$.

In practice, the integral w.r.t. $\sigma$ is approximated using a Monte-Carlo approximation, i.e., we draw $\theta_1, \dots, \theta_L \sim \sigma$ $L$ independent directions, and approximate the Sliced-Wasserstein distance and its gradient as

$$\widehat{\mathrm{SW}}_2^2(\mu, \nu) = \frac{1}{L}\sum_{\ell=1}^L W_2^2(P^{\theta_\ell}_{\#}\mu, P^{\theta_\ell}_{\#}\nu), \quad \widehat{\nabla_{W_2}\mathcal{F}}(\mu) = \frac{1}{L}\sum_{\ell=1}^L \psi'_{\theta_\ell}(\langle x, \theta_\ell \rangle)\theta_\ell. \tag{238}$$

The Wasserstein gradient can also be computed using backpropagation as shown in Proposition A.9.

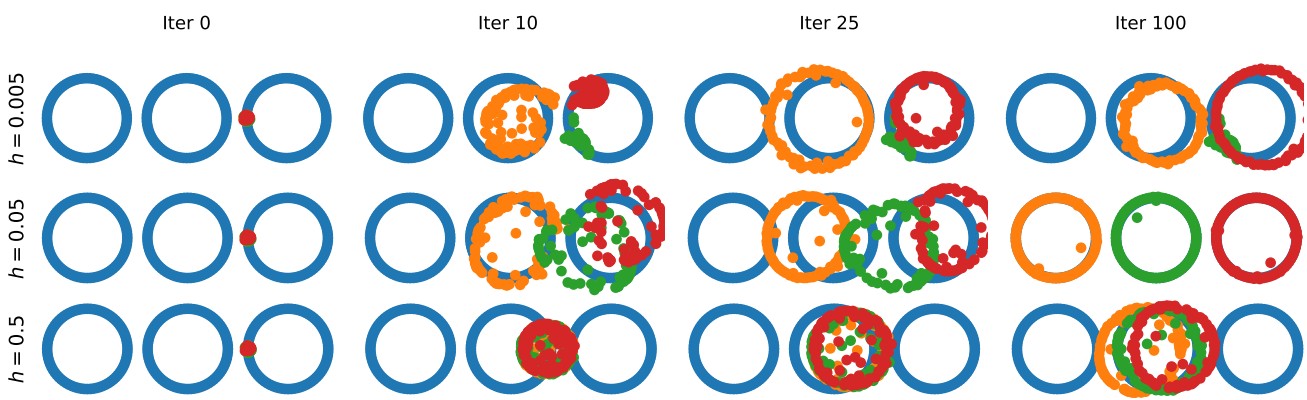

Figure 5: Gradient flow of MMD with SW Gaussian kernel $K(\mu, \nu) = e^{-\mathrm{SW}_2^2(\mu,\nu)/(2h)}$.

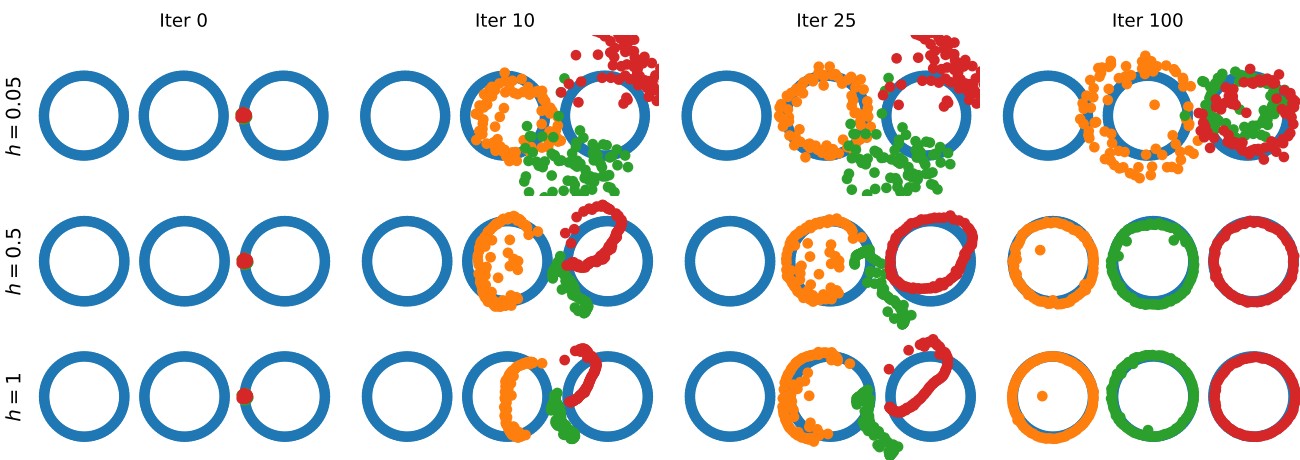

Figure 6: Gradient flow of MMD with SW Laplace kernel $K(\mu, \nu) = e^{-\mathrm{SW}_1(\mu,\nu)/h}$.

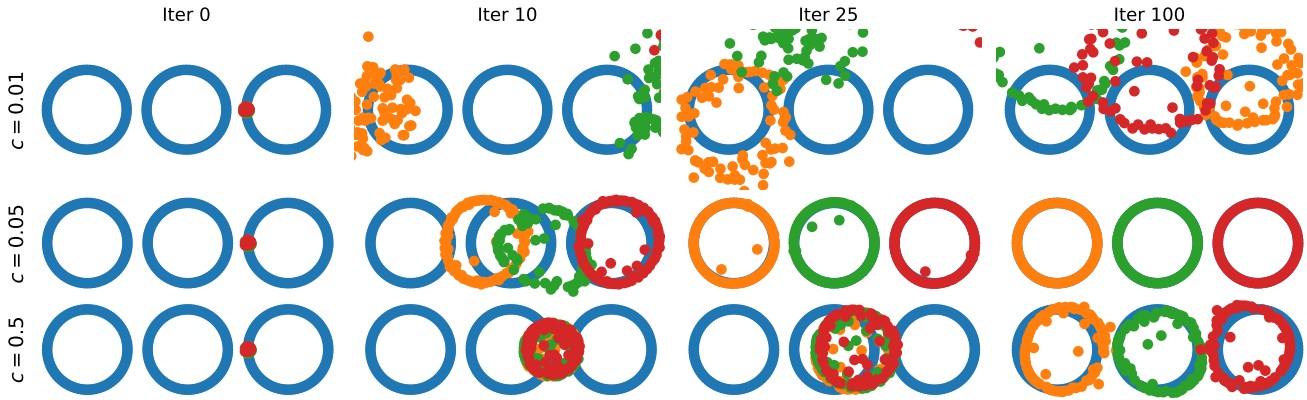

Figure 7: Gradient flow of MMD with SW IMQ kernel $K(\mu, \nu) = (c + \mathrm{SW}_2^2(\mu,\nu))^{-\frac{1}{2}}$.

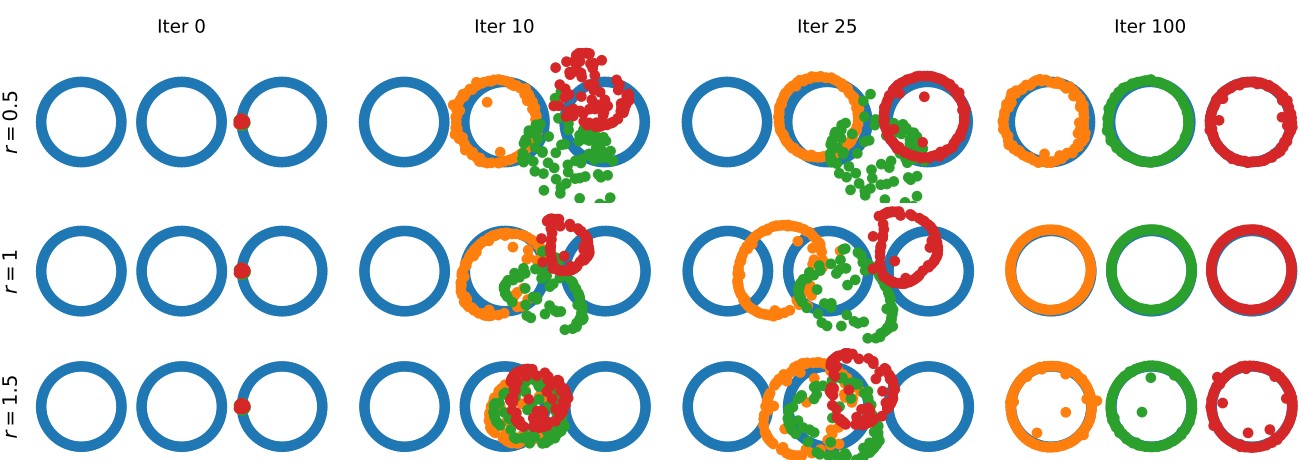

Figure 8: Gradient flow of MMD with SW Riesz kernel $K(\mu, \nu) = -\text{SW}_2(\mu, \nu)^r$.

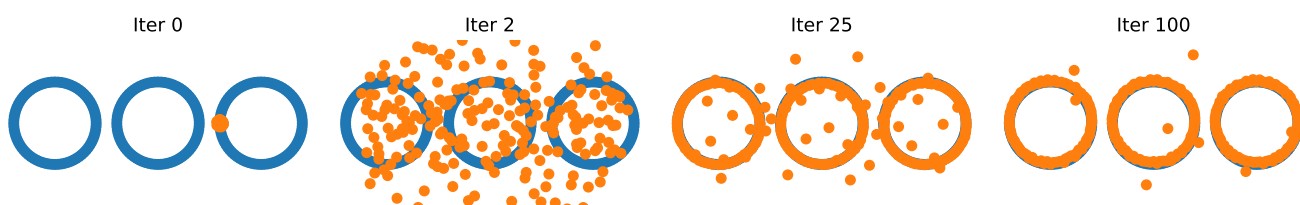

Figure 9: Gradient flow of MMD with Riesz kernel $k(x, y) = -\|x - y\|_2$.

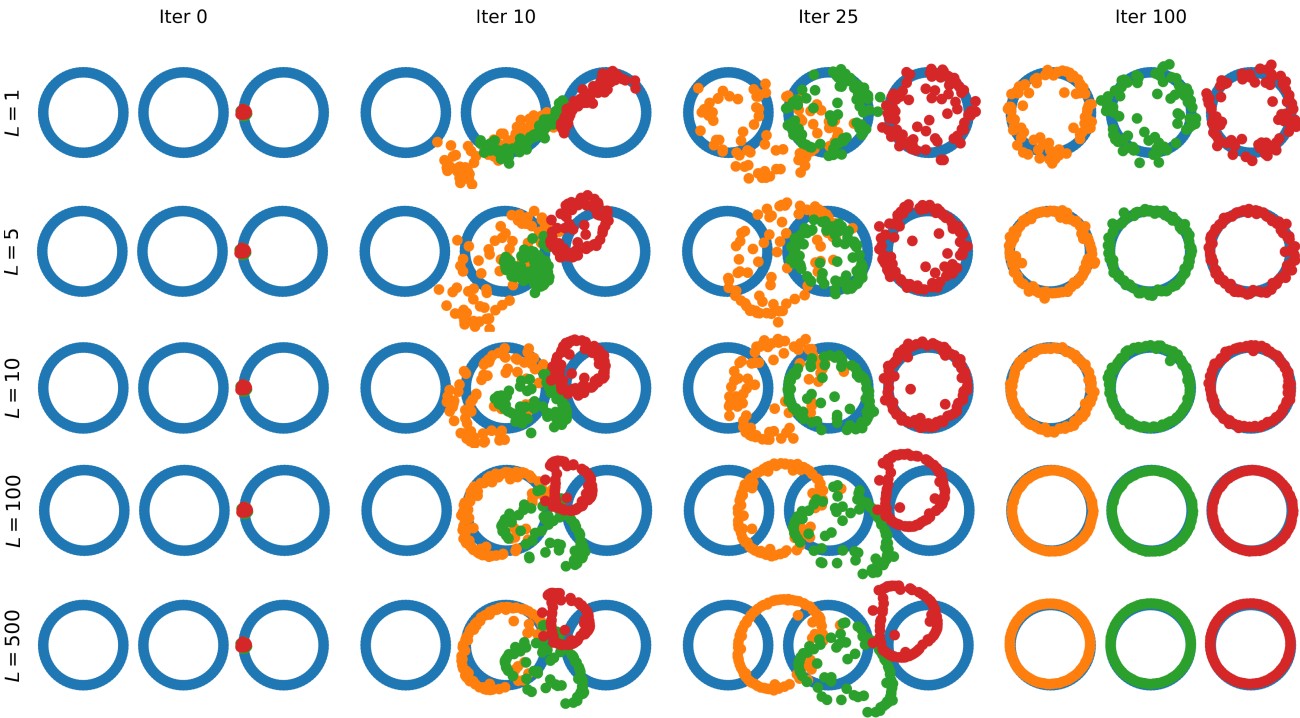

Figure 10: Ablation of the number of projections $L$ for the approximation of the Sliced-Wasserstein distance (with the SW Riesz kernel and $r = 1$).

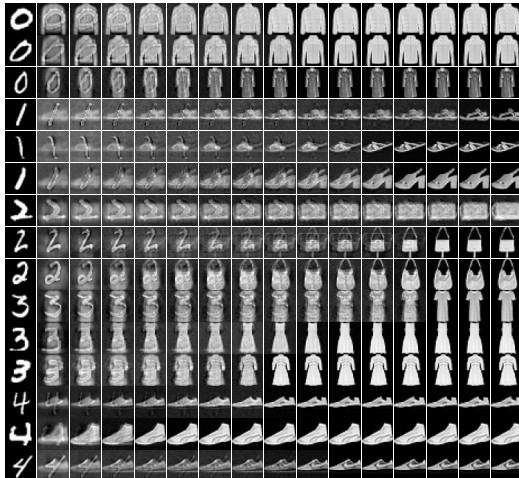 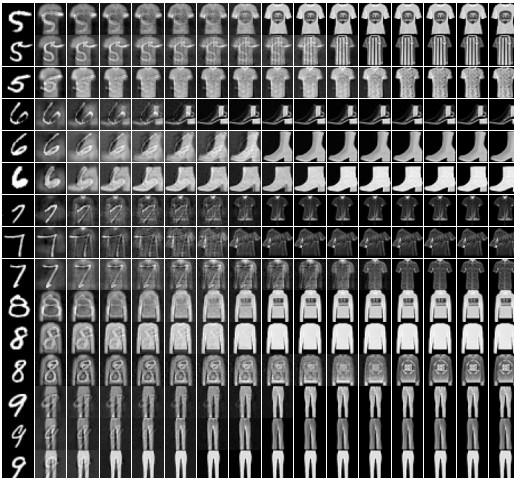

Figure 11: Images along the trajectory of the flow from MNIST to Fashion MNIST. We see that images belonging to the same class in the source dataset are flowed towards images from the same class in the target dataset.

### D.2. Ablation of Hyperparameters on Rings

Additionally to Figure 1, we compare in the following figures trajectories of the minimization of the MMD with various kernels and with different hyperparameters. To recall the setting here, the target is a mixture of rings (Glaser et al., 2021), and each ring is seen through an empirical distribution $\hat{\nu}^{c,n} = \frac{1}{n}\sum_{i=1}^n \delta_{y_i^c}$. Thus, the target is a mixture of three Dirac: $\mathbb{Q} = \frac{1}{3}\delta_{\hat{\nu}^1} + \frac{1}{3}\delta_{\hat{\nu}^2} + \frac{1}{3}\delta_{\hat{\nu}^3}$. Each distribution $\hat{\nu}^c$ contains $n = 80$ samples. We learn a distribution $\mathbb{P} = \frac{1}{3}\delta_{\mu^1} + \frac{1}{3}\delta_{\mu^2} + \frac{1}{3}\delta_{\mu^3}$, modeling each $\mu^c$ as $\mu^c = \frac{1}{n}\sum_{i=1}^n \delta_{x_i^c}$. In practice, the distributions $\mathbb{Q}$ and $\mathbb{P}$ are seen as tensors of size $(3, 80, 2)$.

To compute the gradients of the MMD, we use $L = 500$ projections, and $\tau = 0.1$ as learning rate. We plot on Figure 5 results with the Gaussian SW kernel $K(\mu, \nu) = e^{-\mathrm{SW}_2^2(\mu,\nu)/(2h)}$, on Figure 6 results with the Laplace SW kernel $K(\mu, \nu) = e^{-\mathrm{SW}_1(\mu,\nu)/h}$, on Figure 7 results with the IMQ kernel $K(\mu, \nu) = (c + \mathrm{SW}_2^2(\mu,\nu))^{-\frac{1}{2}}$ and on Figure 8 results with the Riesz SW kernel $K(\mu, \nu) = -\mathrm{SW}_2(\mu,\nu)^r$. We also add on Figure 9 a comparison with the flow of the MMD with Riesz kernel $k(x, y) = -\|x - y\|_2$ as in (Hertrich et al., 2024b), where the structure of the rings is not taken into account.

On Figure 10, we report an ablation of the the trajectories with different number of projections for the SW Riesz kernel. More precisely, we show the results for $L \in \{1, 5, 10, 100, 500\}$. This demonstrates that for low dimensional problems such as 2d rings, $L = 100$ projections already provides good results. However, the scheme is more sensitive to the number of projections in higher dimension as we show on Figure 14.

### D.3. Domain Adaptation

We first add on Figure 11 more samples of the flows of the MMD with $K(\mu, \nu) = -\mathrm{SW}_2(\mu, \nu)$ between MNIST and FashionMNIST. In this experiment, we recall that the flow starts from $\mathbb{P}_0 = \frac{1}{C}\sum_{c=1}^C \delta_{\mu^{c,n}}$, where $\mu^{c,n}$ is the uniform empirical distribution of samples belonging to the class $c \in \{1, \dots 10\}$ of MNIST, and targets $\mathbb{Q} = \frac{1}{C}\sum_{c=1}^C \delta_{\nu^{c,n}}$. We used $n = 200$ samples for each class of the datasets. The Sliced-Wasserstein distance is approximated with $L = 500$ projections. To speed up the flow, similarly as (Hertrich et al., 2024b), we add a momentum $m \in [0, 1)$, *i.e.*, at each iteration $k \geq 0$, the update for each particle $i \in \{1, \dots, n\}$ in class $c \in \{1, \dots, C\}$ is of the form

$$\begin{cases} v_{i,k+1} = \nabla_{\mathrm{W}_{\mathrm{W}_2}} \mathbb{F}(\mathbb{P}_k)(\mu_k^{c,n})(x_{i,k}^c) + mv_{i,k} \\ x_{i,k+1}^c = x_{i,k}^c - \tau v_{i,k+1}, \end{cases} \tag{239}$$

with $v_{i,0} = 0$. We choose a step size of $\tau = 0.05$ and $m = 0.9$.

Complementary to Figure 2, we see on Figure 11 that images from a same class are flowed towards images from a same class in the target dataset. To verify this intuition, we applied a domain adaptation experiment which we describe now.

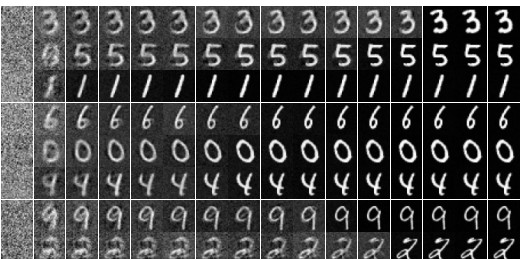 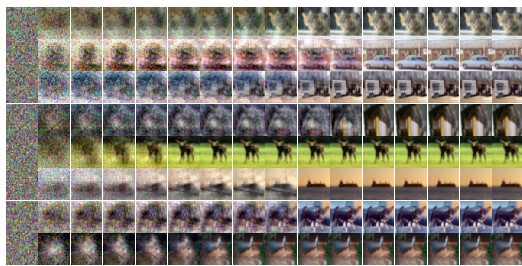

Figure 12: Samples of trajectories starting from Gaussian noise towards MNIST (**Left**) and CIFAR10 (**Right**).

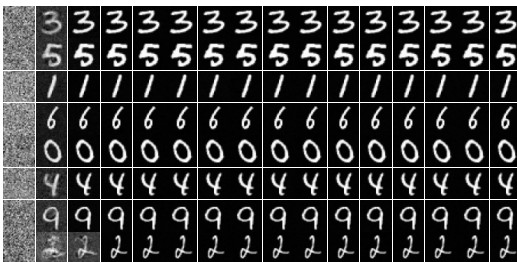 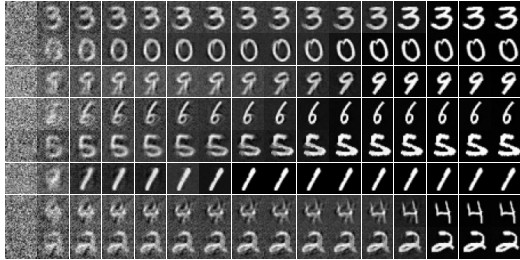

Figure 13: Samples of trajectories starting from Gaussian noise towards MNIST with momentum $m = 0.9$ (**Left**) and no momentum (**Right**). We run the flow for 100K steps, and plot samples every 6667 steps.

We first train a classifier on the training set of the MNIST dataset (using $n = 500$ samples by class). The classifier is the CNN used in the examples of the `equinox` library[4] (Kidger & Garcia, 2021). It is trained for 5000 steps with the AdamW optimizer (Loshchilov & Hutter, 2019) and a batch size of 64. After the training, it has an accuracy of 96% on the test set, and of 100% on the training set.

Then, we flow the dataset FMNIST towards MNIST by minimizing the MMD with kernel $K(\mu, \nu) = -\mathrm{SW}_2(\mu, \nu)$. We run the scheme for 500K steps with a step size of $\tau = 0.1$ and a momentum of $m = 0.9$. To match the labels of the flowed dataset with the labels of MNIST, we solve an OT problem between $\mathbb{P}$ the flowed dataset and $\mathbb{Q}$ the target dataset with the squared 2-Wasserstein distance as groundcost, *i.e.* with $\mathbb{P} = \frac{1}{C} \sum_{c=1}^{C} \delta_{\mu^{c,n}}$ and $\mathbb{Q} = \frac{1}{C} \sum_{c=1}^{C} \delta_{\nu^{c,n}}$, we solve the problem

$$\min_{\Gamma \in \Pi(\mathbb{P}, \mathbb{Q})} \left\langle \left( \mathrm{W}_2^2(\mu^{k,n}, \nu^{\ell,n}) \right)_{1 \leq k, \ell \leq C}, \Gamma \right\rangle_F \tag{240}$$

using the Python Optimal Transport library (Flamary et al., 2021).

We plot on Figure 3 the accuracy of the pretrained classifier along the flow starting from FMNIST. We observe that the accuracy converges to 100% for a sufficient number of iterations. Thus, it shows that the classes of the sources datasets are perfectly flowed towards classes of the target dataset, on which the pretrained neural network is trained, and thus has perfect accuracy.

On Figure 3, we also replicate this experiment from SVHN to CIFAR10 which are composed of $32 \times 32 \times 3$ dimensional images. The neural network used is the same convolutional network used in Appendix D.5, and is pretrained on CIFAR10 during 5000 steps with the AdamW optimizer and a batch size of 64. We use here $n = 100$ samples by class, and run the scheme for $500K$ steps with a step size of $\tau = 0.1$ and $m = 0.9$. We also observe that the accuracy converges to 100%, indicating that it also works in moderately high dimensions.

### D.4. Generative Modeling

In this experiment, we generate samples from different datasets starting from Gaussian noise.

We show on Figure 12 trajectories starting from Gaussian noise towards MNIST and CIFAR10. For both datasets, we use a

---

[4] https://docs.kidger.site/equinox/examples/mnist/

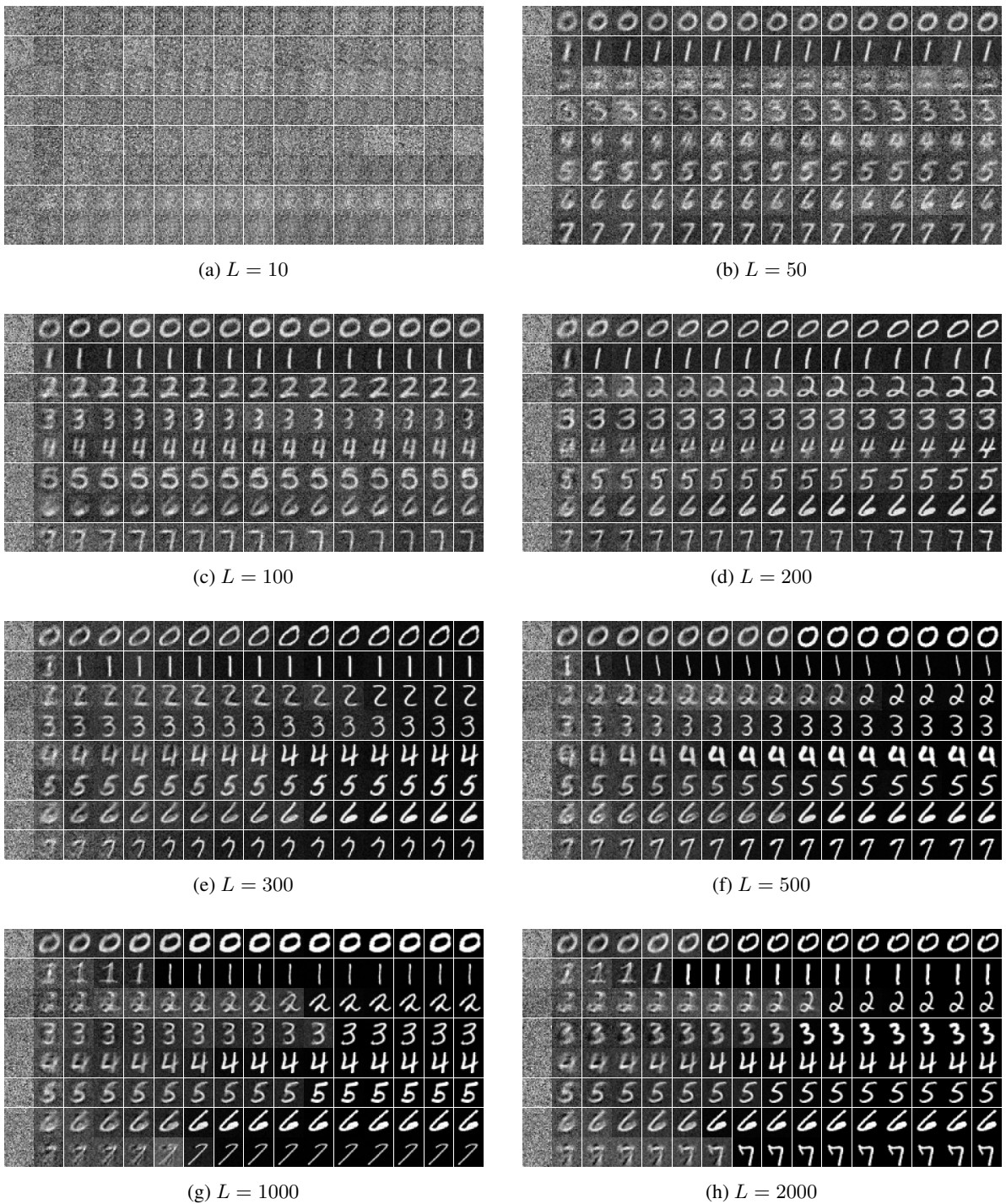

(a) $L = 10$

(b) $L = 50$

(c) $L = 100$

(d) $L = 200$

(e) $L = 300$

(f) $L = 500$

(g) $L = 1000$

(h) $L = 2000$

Figure 14: Ablation over the number of projections for generative modeling on MNIST.

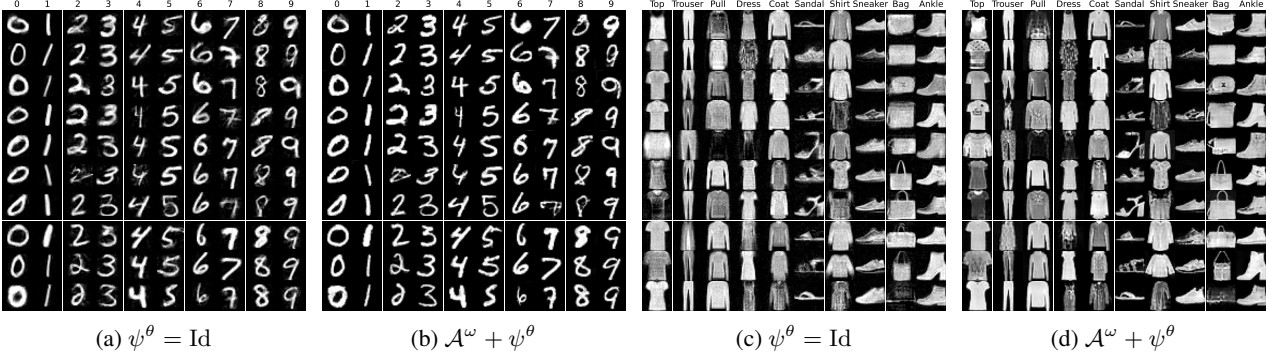

(a) $\psi^\theta = \mathrm{Id}$      (b) $\mathcal{A}^\omega + \psi^\theta$      (c) $\psi^\theta = \mathrm{Id}$      (d) $\mathcal{A}^\omega + \psi^\theta$

Figure 15: Synthetic data for the dataset distillation task on MNIST (**Left**) and FMNIST (**Right**) with or without embedding.

momentum of $m = 0.9$ and a step size of $\tau = 1$. For MNIST, we run the flow for 18K steps, and plot samples every 1200 step, while for CIFAR10, we run it for 150K steps and plot samples every 10K step. We used $n = 200$ samples for each class for MNIST and $n = 50$ for CIFAR10. We also compare trajectories on Figure 13 with using a momentum $m = 0.9$ or no momentum for MNIST, running the flow for 100K steps and showing samples every 6667 step.

In Figure 14, we present an ablation study over the number of projections used to approximate the Sliced-Wasserstein distance on the MNIST dataset (with the same setting with momentum, *i.e.* $m = 0.9$, $\tau = 1$ for 18K steps). We observe that to generate sufficiently clear images, we need at least 300 projections. This may be because a higher number of projections provides a better approximation of the gradients.

### D.5. Dataset Distillation

In this task, we aim at generating a new dataset allowing to approximate a target distribution $\mathbb{Q} = \frac{1}{C} \sum_{c=1}^{C} \delta_{\nu^{c,n}}$ with a distribution $\mathbb{P} = \frac{1}{C} \sum_{c=1}^{C} \delta_{\mu^{c,p}}$ for $p \ll n$, in order to be able to train more efficiently neural networks on it. In Table 1, we take $\mathbb{Q}$ as the MNIST and Fashion MNIST dataset, with $n = 5000$ samples by class, and $C = 10$ classes, and report the results for $p \in \{1, 10, 50\}$. We report the accuracy of a ConvNet trained on the synthetic dataset and evaluated on a test set, averaged over 5 trainings of the neural network, and 3 synthetic datasets. We use a similar architecture as (Zhao & Bilen, 2023), *i.e.* the ConvNet includes three repeated convolutional blocks, and each block involves a 128-kernel convolution layer, instance normalization layer, ReLU activation function and average pooling. This forms the backbone part of the network, and the full classifier is followed by a linear layer. For the initial distribution $\mathbb{P}_0 = \frac{1}{C} \sum_{c=1}^{C} \delta_{\mu^{c,p}}$, each $\mu^{c,p}$ is chosen as a random subset of the samples of $\nu^{c,n}$. The results reported in the column "Random" correspond to the ConvNet trained on the initial data.

Zhao & Bilen (2023) proposed to solve the problem by minimizing

$$\mathcal{F}\big((\mu^c)_c\big) = \sum_{c=1}^{C} \mathbb{E}_{\theta,\omega} \left[ \left\| \int \psi^\theta\big(\mathcal{A}^\omega(x)\big) \, \mathrm{d}(\mu^c - \nu^c)(x) \right\|^2 \right] = \mathbb{E}_{\theta,\omega} \left[ \sum_{c=1}^{C} \mathrm{MMD}_k^2\big(\psi_\#^\theta \mathcal{A}_\#^\omega \mu^c, \psi_\#^\theta \mathcal{A}_\#^\omega \nu^c\big) \right], \tag{241}$$

with linear kernel $k(x,y) = \langle x, y \rangle$, where $\mathcal{A}^\omega : \mathbb{R}^d \to \mathbb{R}^d$ is some data augmentation and $\psi^\theta : \mathbb{R}^d \to \mathbb{R}^{d'}$ with $d' \ll d$ is a randomly initialized neural network used to embed the data. This loss does not take into account the interaction between the classes and just learn any set of synthetic samples for each class. In this work, we propose to take into account the interaction between the classes, and thus minimize

$$\tilde{\mathbb{F}}(\mathbb{P}) = \frac{1}{2} \mathbb{E}_{\theta,\omega} \big[ \mathrm{MMD}_K^2(\phi_\#^{\theta,\omega} \mathbb{P}, \phi_\#^{\theta,\omega} \mathbb{Q}) \big], \tag{242}$$

with $K(\mu, \nu) = -\mathrm{SW}_2(\mu, \nu)$.

In practice, for $\psi^\theta$, we use the backbone part of the ConvNet, and for $\mathcal{A}^\omega$, we follow the same strategy as (Zhao & Bilen, 2021) (*i.e.* we sample one augmentation among color jittering, cropping, cutout, scaling and a rotation for MNIST, and also add flipping for Fashion MNIST). We optimize (241) by stochastic gradient descent over the particles, sampling one random network and one random augmentation at each step. We trained it for 20K iterations, a learning rate of $\tau = 1$ and a momentum of $m = 0.9$. In practice, we observed numerical instabilities when optimizing in the ambient space with augmentations.

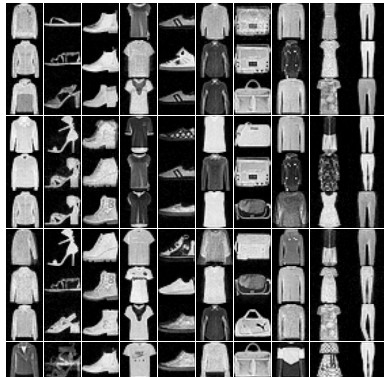 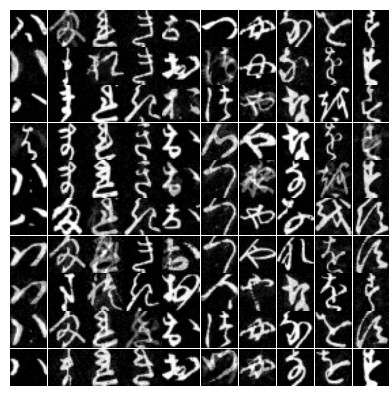 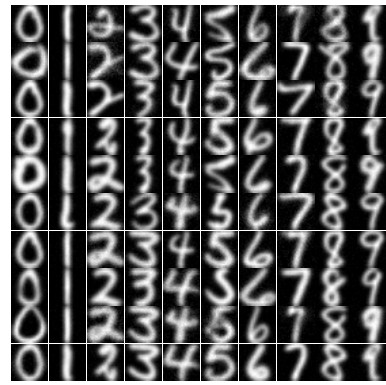

Figure 16: Examples of images output by flows for the transfer learning task with $k = 10$ for Fashion MNIST (**Left**), KMNIST (**Middle**) and USPS (**Right**).

To optimize (242), we also performed a stochastic gradient descent, sampling one random neural network and one random augmentation at each step. We used also 20K iterations, a learning rate of $\tau = 1$ and a momentum $m = 0.9$. Then, we assign the classes using an OT matching as explained in Appendix D.3. We add on Figure 15 samples learned with this loss, with and without embedding. We observe that images are slightly clearer when using an embedding. To compute the gradient of $\tilde{\mathbb{F}}$ in practice, we use autodifferentiation.

### D.6. Transfer Learning

We describe the details for the experiment of transfer learning. We recall that the target dataset is of the form $\mathbb{Q} = \frac{1}{C} \sum_{c=1}^{C} \delta_{\nu^{c,k}}$ with $\nu^{c,k}$ a uniform empirical distribution of $k$ samples of the class $c$. In Table 2, the targets datasets are Fashion-MNIST, KMNIST and USPS. Thus, $C = 10$, and we choose $k \in \{1, 5, 10, 100\}$. For the source dataset $\mathbb{P} = \frac{1}{C} \sum_{c=1}^{C} \delta_{\mu^{c,n}}$, we used the MNIST dataset with $n = 200$ samples in each class.

We augment the target dataset by flowing the samples of MNIST on the target. For MMDSW, we minimize $\mathbb{F}(\mathbb{P}) = \frac{1}{2} \text{MMD}_K^2(\mathbb{P}, \mathbb{Q})$ with kernel $K(\mu, \nu) = -\text{SW}_2(\mu, \nu)$, by running the forward scheme for 5K steps for $k \in \{1, 5, 10\}$ and 20K steps for $k = 100$, with step size $\tau = 1$ and momentum $m = 0.9$. Finally, we align the labels using an OT matching between the flowed samples $\mathbb{P}$ and the target $\mathbb{Q}$, as for the dataset distillation experiment.

We compare it with training directly on the small dataset, and with two other methods. The first one, called OTDD (Alvarez-Melis & Fusi, 2020), represents the dataset as a probability distribution on $\mathbb{R}^d \times \mathcal{P}_2(\mathbb{R}^d)$, where the labels are embedded in $\mathcal{P}_2(\mathbb{R}^d)$ by considering the conditional distribution, *i.e.*, a feature-label pair $(x, c)$ is represented as $(x, \mu^c)$. Then, they compare datasets using Optimal Transport with cost $d\big((x, c), (x', c')\big)^2 = \|x - x'\|_2^2 + \text{W}_2^2(\mu^c, \mu^{c'})$. The flow of OTDD then minimizes the OT distance with this cost, *i.e.*, the objective is $\mathcal{F}(\mu) = \frac{1}{2} \text{OTDD}(\mu, \nu)$ for $\mu, \nu \in \mathcal{P}_2(\mathbb{R}^d \times \mathcal{P}_2(\mathbb{R}^d))$, with

$$\text{OTDD}(\mu, \nu) = \inf_{\gamma \in \Pi(\mu, \nu)} \int \left( \|x - x'\|_2^2 + \text{W}_2^2(\mu^c, \mu^{c'}) \right) \, \mathrm{d}\gamma\big((x, c), (x, c')\big). \tag{243}$$

For big datasets, the conditional distributions $\mu_c$ can be approximated by Gaussian distributions. Alvarez-Melis & Fusi (2021) proposed several schemes to optimize this loss using Wasserstein gradient flows. We did not manage to replicate their results with their code. Thus, we reimplemented it with some differences. First, similarly as (Hua et al., 2023), we used an embedding in dimension 2 of the data to approximate the conditional distributions with Gaussian distributions. Thus, we model the datasets as distributions over $\mathbb{R}^d \times \mathbb{R}^2 \times S_2^{++}(\mathbb{R})$, with $S_2^{++}(\mathbb{R})$ the space of symmetric positive definite matrices. This helps avoiding memory issues and scaling to higher dimensional datasets as it reduces a lot the dimension of the samples to flow. For this embedding, we used a Principal Component Analysis (but note that we could use other embedding methods such as TSNE (Hua et al., 2023) or Multidimensional Scaling (Liu et al., 2025)). In practice, we approximate OTDD using an entropic regularization, which we compute using the Sinkhorn algorithm (Cuturi, 2013) and `ott-jax` (Cuturi et al., 2022). We optimize it using AdamW with a learning rate of $\tau = 1e^{-3}$ and run it for 5K iterations for $k \in \{1, 5, 10, 100\}$. To get the labels, we use an OT matching as in (Hua et al., 2023), which we solve using `POT` (Flamary et al., 2021). More precisely, for each class $c \in \{1, \dots, C\}$ of the target distribution, we can compute a mean $\bar{m}_c$ and a covariance $\bar{\Sigma}_c$, and a

weight $\omega_c = \frac{n_c}{n}$ with $n$ the number of samples in the target dataset, and $n_c$ the number of samples belonging to class $c$. After flowing $n$ samples, we have tuples $(x_i, m_i, \Sigma_i)_{i=1}^n$ and we want to associate to each sample a class. To do this, they propose to solve the discrete OT problem between $\mathbb{Q} = \sum_{c=1}^C \omega_c \delta_{\mathcal{N}(\bar{m}_c, \bar{\Sigma}_c)}$ and $\mathbb{P} = \frac{1}{n} \sum_{i=1}^n \delta_{\mathcal{N}(m_i, \Sigma_i)}$:

$$\min_{P \in \Pi(\mathbb{P}, \mathbb{Q})} \sum_{i=1}^n \sum_{c=1}^C P_{ic} \mathrm{W}_2^2 \big( \mathcal{N}(m_i, \Sigma_i), \mathcal{N}(m_c, \Sigma_c) \big), \tag{244}$$

and then use as distribution $\mu^n = \frac{1}{n} \sum_{i=1}^n \delta_{(x_i, y_i)}$ with $y_i = \sum_{c=1}^C c \mathbb{1}_{\{P_{ic}^* = \max \ P_i^*\}}$.

The second baseline we use is the one proposed in (Hua et al., 2023). In this work, they first observe that the Gaussian approximation for high dimensional datasets might not scale well in memory. Thus, they propose to use an embedding in a lower dimension space of the conditional distributions, before doing the Gaussian approximation. The datasets are then represented as probability distributions on $\mathbb{R}^d \times \mathbb{R}^p \times S_p^{++}(\mathbb{R})$ with $p \ll d$. Instead of using an OT cost to compare the datasets, they used the MMD with a kernel obtained as a product of Gaussian kernel. Then, they applied a Wasserstein gradient flow of the MMD (Arbel et al., 2019) to minimize it, with a Bures-Wasserstein gradient descent step (Altschuler et al., 2021) for the symmetric positive definite covariance matrix. We note that in contrast with our proposed MMD, it requires many hyperparameters to tune (3 bandwidth of Gaussian kernels and noise to add to make the flow converge). We reimplemented it in `jax` (Bradbury et al., 2018), used $p = 2$ and a Principal Component Analysis (using `scikit-learn` (Pedregosa et al., 2011)) for the lifting of the conditional distribution (instead of TSNE in (Hua et al., 2023)). The Gaussian of each class is then obtained by computing the mean and variance of each class. We used as bandwidth $h = 100$ for the feature part, $h = 50$ for the mean part and $h = 1000$ for the covariance part. We ran the flow for 20K steps with a step size of $\tau = 10$, and momentum $m = 0.9$. To get the final labels, we solved (244) as explained in the last paragraph.

In Table 2, we report the accuracy obtained by training a LeNet-5 neural network for 50 epochs with a AdamW optimizer and a learning rate of $3 \cdot 10^{-4}$. Moreover, we average the results for 5 trainings of the neural network, and 3 outputs of the flows. We add on Figure 16 examples of images returned at the end of the flow of the MMD with $K(\mu, \nu) = -\mathrm{SW}_2(\mu, \nu)$.

### D.7. Handling Different Number of Distributions between the Source and Target

Let $\mathbb{P} = \frac{1}{N} \sum_{k=1}^N \delta_{\mu^{k,n}}$ and $\mathbb{Q} = \frac{1}{M} \sum_{k=1}^M \delta_{\nu^{k,n}}$ with $M < N$. In this situation, the flow might not converge well towards the target distribution since they have a different number of Dirac. This is illustrated on Figure 17, where the target is composed of $M = 3$ rings $\nu^{k,n}$, and the source is initialized with $N = 4$ distributions, and we minimize the MMD with a Gaussian SW kernel $K(\mu, \nu) = e^{-\mathrm{SW}_2^2(\mu, \nu)/h^2}$. We see that the flow does not converge to 3 rings, as it cannot split the mass because the Wasserstein gradient descent allow only changing the position of particles.

This problem could be solved by different solutions. For instance, one could use a Wasserstein-Fisher-Rao gradient flow instead of a Wasserstein gradient flow (Gallouët & Monsaingeon, 2017). This flow can be approximated *e.g.* by using Birth death Langevin algorithms (Lu et al., 2019; 2023) where the Langevin step approximates the Wasserstein gradient flow part, and the Birth death part approximates the Fisher-Rao gradient flow part. The birth death consists at killing and duplicating randomly particles at each step. Another solution to approximate the Fisher-Rao flow is to change the weights (Yan et al., 2024).

We propose to perform the Wasserstein gradient flow, but allowing to change the weights of the particles, which is not possible for the Wasserstein gradient descent. Ideally, one would want to solve directly the JKO scheme

$$\begin{cases} \gamma_{k+1} = \mathrm{argmin}_{\gamma \in \mathcal{P}_2(\mathbb{R}^d \times \mathbb{R}^d), \ \pi_\#^1 \gamma = \mu_k} \int \|x - y\|_2^2 \, \mathrm{d}\gamma(x, y) + \tau \mathcal{F}(\pi_\#^2 \gamma) \\ \mu_{k+1} = \pi_\#^2 \gamma_{k+1}. \end{cases} \tag{245}$$

However, if we do not fix the support, it is not possible to directly solve this problem, except if we use neural networks. Note that (245) can be seen as a semi-relaxed unbalanced optimal transport problem, where the first marginal is fixed. This has been leveraged to solve the JKO scheme *e.g.* in (Choi et al., 2024).

We propose instead to alternate between a Wasserstein gradient descent step, which allows moving the particles without changing the weights, and a backward step for which we optimize over the coupling while fixing its support, which allows then to change the weights.

For simplicity, let us describe the procedure more precisely on $\mathcal{P}(\mathbb{R}^d)$. Let $\nu \in \mathcal{P}_2(\mathbb{R}^d)$ be a target distribution, and suppose at step $k$, $\mu_k = \sum_{i=1}^n \alpha_i^k \delta_{x_i}$ with $\alpha_i^k \geq 0$, $\sum_{i=1}^n \alpha_i^k = 1$. Let $D$ be a divergence we want to minimize *w.r.t.* $\nu$, *i.e.* we want

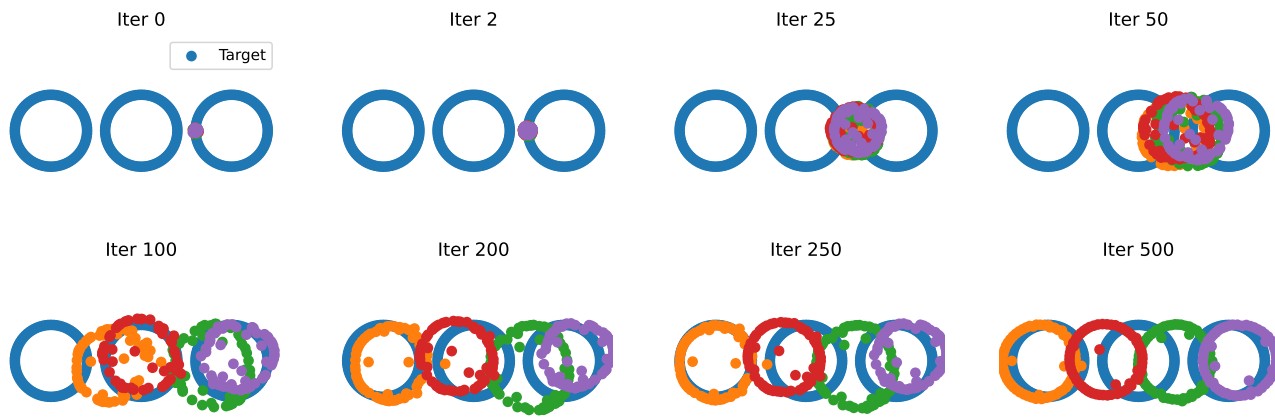

Figure 17: MMD with Gaussian SW kernel and 4 distributions flowed towards 3 rings. The flow does not converge to the 3 rings.

to minimize $\mathcal{F}(\mu) = D(\mu, \nu)$. Then, our update is

$$
\begin{cases}
\mu_{k+\frac{1}{2}} = \left(\mathrm{Id} - \tau \nabla_{\mathrm{W}_2} \mathcal{F}(\mu_k)\right)_{\#} \mu_k \\
\gamma_{k+1} = \mathrm{argmin}_{\gamma \in \mathcal{P}(\mathbb{R}^d \times \mathbb{R}^d), \mathrm{supp}(\gamma) \subset \mathrm{supp}(\mu_{k+\frac{1}{2}}) \times \mathrm{supp}(\mu_{k+\frac{1}{2}}), \, \pi^1_{\#}\gamma = \mu_{k+\frac{1}{2}}} \; \frac{1}{2} \int \|x - y\|_2^2 \, \mathrm{d}\gamma(x, y) + \tau D(\pi^2_{\#}\gamma, \nu) \\
\mu_{k+1} = \pi^2_{\#} \gamma_{k+1}.
\end{cases}
\tag{246}
$$

The first step is a regular forward step, which moves the position of the particles. The second step learns a coupling $\gamma \in \mathcal{P}_2(\mathbb{R}^d \times \mathbb{R}^d)$ which satisfies $\pi^1_{\#}\gamma = \mu_{k+\frac{1}{2}}$, and such that $\pi^2_{\#}\gamma$ is supported on the same set of particles. This step can be seen as solving a semi-relaxed Unbalanced Optimal Transport problem if the support for both distributions is the same. Suppose that $\gamma = \sum_{i,j=1}^n P_{ij} \delta_{(x_i, x_j)}$, and note $C \in \mathbb{R}^{n \times n}$ the matrix distance. Then, the second step can be rewritten as

$$
\min_{P \in \mathbb{R}_+^{n \times n}, \langle P, \mathbb{1}_{\{n \times n\}} \rangle = 1, P \mathbb{1}_n = \alpha} \langle C, P \rangle + \tau D \left( \sum_{i=1}^n [P^T \mathbb{1}_n]_i \delta_{x_i}, \nu \right).
\tag{247}
$$

For $D(\mu, \nu) = \mathrm{KL}(\mu \| \nu)$, this can be solved using the Sinkhorn algorithm for the semi-relaxed UOT problem, *i.e.* with $\varphi_1 = \iota_{\{1\}}$ (Séjourné et al., 2023). For $D = \mathrm{MMD}^2$, one can use different algorithms to solve it such as a Projected Mirror Descent or an Accelerated Gradient Descent (Manupriya et al., 2024). Here, we propose to use the half step of the Mirror Sinkhorn algorithm (Ballu & Berthet, 2023), which performs first a Mirror Descent step with Bregman potential $\phi(P) = \langle P, \log P \rangle$ (for which $P_{k+1} = \nabla \phi^*(\nabla \phi(P_k) - \tau \nabla f(P_k)) = P_k \odot e^{-\tau \nabla f(P_k)}$), and then perform a (Sinkhorn-like) projection on the constraint, *i.e.*, noting

$$
f(P) = \langle C, P \rangle + \tau \mathrm{MMD}^2 \left( \sum_{i=1}^n [P^T \mathbb{1}_n]_i \delta_{x_i}, \nu \right)
\tag{248}
$$

the objective, the algorithm becomes

$$
\begin{cases}
P'_{k+1} = P_k \odot e^{-\tau \nabla f(P_k)} \\
P_{k+1} = \mathrm{diag}\left( \alpha \oslash (P'_{k+1} \mathbb{1}_n) \right) P'_{k+1}.
\end{cases}
\tag{249}
$$

We show on Figure 18 the results on the rings experiment. We observe that the weight of the 4th ring is set to 0, and thus that the scheme converges to the target.

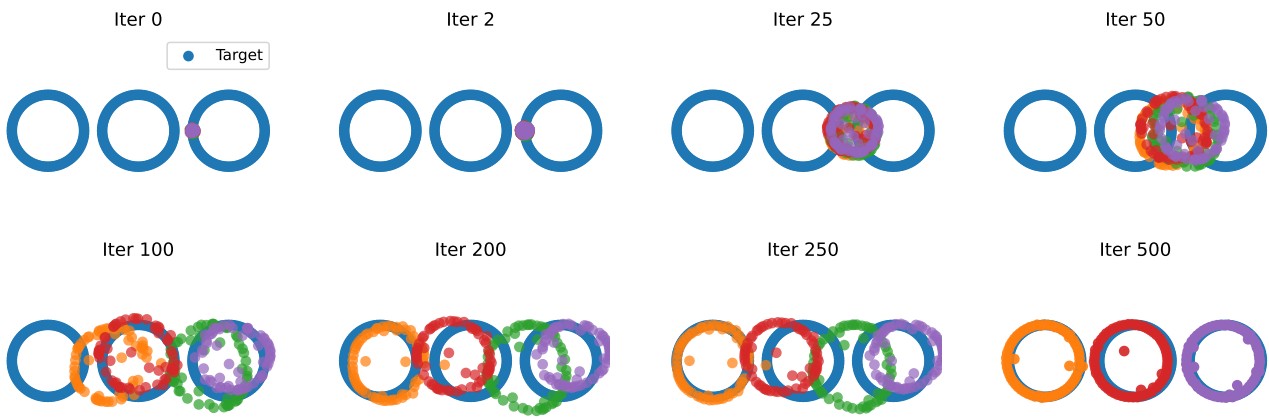

Figure 18: MMD with Gaussian SW kernel and 4 distributions flowed towards 3 rings using the proposed algorithm. The flow converges to the 3 rings by setting the weights of one of the ring to 0.

## E. Related Works

### E.1. Optimal Transport Distance for Datasets

Alvarez-Melis & Fusi (2020) first proposed to compare datasets with a dedicated discrepancy, which takes into account features and labels. They proposed to do it by representing datasets as uniform empirical distributions over $\mathbb{R}^d \times \mathcal{P}_2(\mathbb{R}^d)$, embedding the labels in $\mathcal{P}_2(\mathbb{R}^d)$ by considering the conditional distributions, *i.e.*, a feature-label pair $(x, c)$ is represented as $(x, \mu^c)$ with $\mu^c$ the distribution of samples belonging to the class $c$. They proposed to compare datasets using an optimal transport distance with cost $d\big((x,c),(x',c')\big)^2 = \|x - x'\|_2^2 + \mathrm{W}_2^2(\mu^c, \mu^{c'})$. To summarize, they consider as distance between $\mu, \nu \in \mathcal{P}_2(\mathbb{R}^d \times \mathcal{P}_2(\mathbb{R}^d))$,

$$\mathrm{OTDD}(\mu,\nu) = \inf_{\gamma \in \Pi(\mu,\nu)} \int \big(\|x - x'\|_2^2 + \mathrm{W}_2^2(\mu^c, \nu^{c'})\big) \, \mathrm{d}\gamma\big((x,c),(x,c')\big). \tag{250}$$

In practice, Alvarez-Melis & Fusi (2020) approximated the conditional distributions $\mu_c$ by Gaussians to be able to compute the Wasserstein distance in closed-form, which leads to a complexity of $O\big(Cnd^2 + C^2d^3 + n^3C^3 \log(nC)\big)$ as it requires to estimate $C$ means and covariance matrices from $n$ samples, to compute $C^2$ Bures-Wasserstein distances, and an OT problem between $Cn$ samples. The final OT problem can be approximated using an entropic regularization, which reduces the complexity to $O\big(Cnd^2 + C^2d^3 + \varepsilon^{-2}n^2C^2 \log(nC)\big)$ (Dvurechensky et al., 2018).

Liu et al. (2025) instead embedded the labels in $\mathbb{R}^d$ using a Multidimensional Scaling, and further approximated the resulting squared Wasserstein distance with a Wasserstein embedding. Bonet et al. (2025) proposed to embed the labels in a hyperbolic space, and used a Sliced-Wasserstein distance to compare distributions on the product space $\mathbb{R}^d \times \mathbb{H}$. Nguyen & Ho (2024) used a similar embedding, and a hierarchical hybrid Sliced-Wasserstein distance. More recently, Nguyen et al. (2025) introduced a sliced optimal transport dataset distance using a dedicated projection from $\mathbb{R}^d \times \mathcal{P}_2(\mathbb{R}^d)$ to $\mathbb{R}$.

Concerning the task of flowing datasets, Alvarez-Melis & Fusi (2021); Hua et al. (2023) both modeled conditional distributions as Gaussian, and solved flows on $\mathbb{R}^d \times \mathbb{R}^p \times S_p^{++}(\mathbb{R})$. More precisely, Alvarez-Melis & Fusi (2021) minimized OTDD on $\mathbb{R}^d \times \mathbb{R}^d \times S_d^{++}(\mathbb{R})$, while Hua et al. (2023) minimized an MMD over $\mathbb{R}^d \times \mathbb{R}^2 \times S_2^{++}(\mathbb{R})$ with a product of Gaussian kernels, and using an embedding on $\mathbb{R}^2$ for the conditional distributions. In contrast to these works, we encode the labels directly into the discrepancy by using a MMD on the space of probability distributions with a suitable kernel.

Alvarez-Melis & Fusi (2021) proposed several ways of minimizing $\mathcal{F}(\mu) = \mathrm{OTDD}(\mu,\nu)$ for $\mu = \frac{1}{n}\sum_{i=1}^n \delta_{(x_i,\mu^{c_i})}$. Let us note $\mu_k = \frac{1}{n}\sum_{i=1}^n \delta_{(x_{i,k},\mu_k^{c_i})}$ the dataset at step $k$, and assume $c_i \in \{1, \ldots, C\}$. For small datasets for which the Wasserstein distance between conditional distributions can be computed efficiently, they just proposed to flow the samples, *i.e.* computing $x_{i,k+1} = x_{i,k} - \tau \nabla_{x_i} \mathrm{OTDD}(\mu_k, \nu)$ and updating the conditional distributions at each step. When using the Gaussian approximation, they proposed to update the mean and covariance at each step (feature driven), or to do a gradient descent step for the $C$ means and covariances (joint-driven-fixed-label). They also considered the joint-driven-variable-

label, where they decoupled at time 0 the Gaussian, and flowed one mean $m_i$ and covariance $\Sigma_i$ by Gaussian, *i.e.*, for all $i \in \{1, \ldots, n\}$ and $k \geq 0$,

$$
\begin{cases}
x_{i,k+1} = x_{i,k} - \tau \nabla_{x_i} \text{OTDD}(\mu_k, \nu) \\
m_{i,k+1} = m_{i,k} - \tau \nabla_{m_i} \text{OTDD}(\mu_k, \nu) \\
\Sigma_{i,k+1} = \Sigma_{i,k} - \tau \nabla_{\Sigma_i} \text{OTDD}(\mu_k, \nu).
\end{cases}
\tag{251}
$$

This however requires to cluster the pairs $(m_i, \Sigma_i)$ to recover labels.

Hua et al. (2023) observed that the embedding of the conditional distribution as Gaussian can be very costly in practice for high-dimensional datasets. Thus, they first proposed to embed the features in $\mathbb{R}^2$ using TSNE, in order to embed the labels as Gaussian in $\mathbb{R}^2$, and therefore represented the datasets as empirical distributions over $\mathbb{R}^d \times \mathbb{R}^2 \times S_2^{++}(\mathbb{R})$. Then, they proposed to minimize the MMD on this space with kernel $k\big((x, m, \Sigma), (x', m', \Sigma')\big) = e^{-\|x-x'\|_2^2/h_x} e^{-\|m-m'\|_2^2/h_m} e^{-\|\Sigma-\Sigma'\|_2^2/h_\Sigma}$. For $\mu, \nu \in \mathcal{P}_2\big(\mathbb{R}^d \times \mathbb{R}^2 \times S_2^{++}(\mathbb{R})\big)$, let $\mathcal{F}(\mu) = \frac{1}{2}\text{MMD}^2(\mu, \nu) = \int V \mathrm{d}\mu + \frac{1}{2} \iint k(x, y) \, \mathrm{d}\mu(x)\mathrm{d}\mu(y)$, with $V(x) = -\int k(x, y)\mathrm{d}\nu(x)$. Its Wasserstein gradient is then for all $(x, m, \Sigma)$,

$$
\nabla_{\mathrm{W}_2}\mathcal{F}(\mu)\big((x, m, \Sigma)\big) = \nabla V\big((x, m, \Sigma)\big) + \int \nabla_1 k\big((x, m, \Sigma), (x', m', \Sigma')\big) \, \mathrm{d}\mu\big((x', m', \Sigma')\big) \in \mathbb{R}^d \times \mathbb{R}^2 \times S_2(\mathbb{R}).
\tag{252}
$$

Using the Bures-Wasserstein geometry for the covariance part, their updates are given by

$$
\begin{cases}
x_{i,k+1} = x_{i,k} - \tau [\nabla_{\mathrm{W}_2}\mathcal{F}(\mu_k)\big((x_{i,k}, m_{i,k}, \Sigma_{i,k})\big)]_1 \\
m_{i,k+1} = m_{i,k} - \tau [\nabla_{\mathrm{W}_2}\mathcal{F}(\mu_k)\big((x_{i,k}, m_{i,k}, \Sigma_{i,k})\big)]_2 \\
\Sigma_{i,k+1} = \exp_{\Sigma_{i,k}}\big(-\tau [\nabla_{\mathrm{W}_2}\mathcal{F}(\mu_k)\big((x_{i,k}, m_{i,k}, \Sigma_{i,k})\big)]_3\big),
\end{cases}
\tag{253}
$$

with $\exp_\Sigma(S) = (I_d + S)\Sigma(I_d + S)$ for $\Sigma \in S_d^{++}(\mathbb{R})$, $S \in S_d(\mathbb{R})$ the exponential map on the Bures-Wasserstein space, see *e.g.* (Altschuler et al., 2021, Appendix A.1).

## E.2. Variational Inference with Mixture of Gaussians

Lambert et al. (2022) considered to do Variational Inference with a family of Gaussian mixtures. Let's note $\mathrm{BW}(\mathbb{R}^d) \subset \mathcal{P}_2(\mathbb{R}^d)$ the Bures-Wasserstein space, *i.e.*, the space of Gaussian distributions endowed with the Wasserstein distance. Observing that there is an identification between $\mathrm{BW}(\mathbb{R}^d)$ and $\mathbb{R}^d \times S_d^{++}(\mathbb{R})$ (Chen et al., 2018; Delon & Desolneux, 2020), this amounts at solving the problem, for $\pi \in \mathcal{P}_{2,\mathrm{ac}}(\mathbb{R}^d)$,

$$
\min_{\mu \in \mathcal{P}_2(\mathbb{R}^d \times S_d^{++}(\mathbb{R}))} \text{KL}\left(\int p_\theta \mathrm{d}\mu(\theta) \| \pi\right),
\tag{254}
$$

where $p_\theta = \mathcal{N}(\cdot; m, \Sigma)$ for $\theta = (m, \Sigma) \in \mathbb{R}^d \times S_d^{++}(\mathbb{R})$. Equivalently, it can be framed as an optimization problem over $\mathcal{P}_2(\mathrm{BW}(\mathbb{R}^d))$, by solving

$$
\min_{\mathbb{P} \in \mathcal{P}_2(\mathrm{BW}(\mathbb{R}^d))} \text{KL}\left(\int \mu \, \mathrm{d}\mathbb{P}(\mu) \| \pi\right).
\tag{255}
$$

Note that the KL here is the usual Kullback-Leibler divergence, defined between $\mu, \nu \in \mathcal{P}_{2,\mathrm{ac}}(\mathbb{R}^d)$ as

$$
\text{KL}(\mu\|\nu) = \int \log\left(\frac{p_\mu(x)}{p_\nu(x)}\right) \mathrm{d}\mu(x),
\tag{256}
$$

where we note $p_\mu$ and $p_\nu$ the densities of $\mu$ and $\nu$ *w.r.t* the Lebesgue measure.

They address the problem by solving an ODE on the means and covariances, which characterizes the trajectory of the gradient flow in $(\mathcal{P}_2(\mathrm{BW}(\mathbb{R}^d)), \mathrm{W}_{\mathrm{BW}_2})$. Alternatively, they propose to solve the JKO scheme between particles by solving for all $k \geq 0$,

$$
(\theta_{k+1}^{(1)}, \ldots, \theta_{k+1}^{(n)}) = \underset{\theta^{(1)}, \ldots, \theta^{(n)}}{\arg\min} \ \mathbb{F}\left(\frac{1}{n}\sum_{i=1}^n \delta_{\mathcal{N}(m^{(i)}, \Sigma^{(i)})}\right) + \frac{1}{\tau}\mathrm{W}_{\mathrm{W}_2}^2\left(\frac{1}{n}\sum_{i=1}^n \delta_{\mathcal{N}(m^{(i)}, \Sigma^{(i)})}, \frac{1}{n}\sum_{i=1}^n \delta_{\mathcal{N}(m_k^{(i)}, \Sigma_k^{(i)})}\right).
\tag{257}
$$

We now derive the gradient of this functional using our framework, and make the connections with the formula derived in (Lambert et al., 2022, Appendix F).

**Computation of the gradient.** Let $\mathbb{P} \in \mathcal{P}_2\big(\mathcal{P}_2(\mathbb{R}^d)\big)$ and $\xi \in T_\mathbb{P}\mathcal{P}_2\big(\mathcal{P}_2(\mathbb{R}^d)\big)$. We want to do the Taylor expansion of $\mathbb{F}(\exp_\mathbb{P}(t\xi)) = \mathbb{F}\big((\mu \mapsto (\mathrm{Id} + t\xi(\mu))_{\#}\mu)_{\#}\mathbb{P}\big)$:

$$
\begin{aligned}
\mathbb{F}\big(\exp_\mathbb{P}(t\xi)\big) &= \mathrm{KL}\left(\int \mu \, \mathrm{d}\big(\exp_\mathbb{P}(t\xi)\big)(\mu) \big|\big| \pi \right) \\
&= \mathrm{KL}\left(\int (\mathrm{Id} + t\xi(\mu))_{\#}\mu \, \mathrm{d}\mathbb{P}(\mu) \big|\big| \pi \right) \\
&= \iint \log\left(\frac{\int p_{(\mathrm{Id}+t\xi(\nu))_{\#}\nu}(x)\, \mathrm{d}\mathbb{P}(\nu)}{p_\pi(x)}\right) p_{(\mathrm{Id}+t\xi(\mu))_{\#}\mu}(x)\, \mathrm{d}\mathbb{P}(\mu)\mathrm{d}x.
\end{aligned}
\tag{258}
$$

By a Taylor expansion, we can write for all $x \in \mathbb{R}^d$,

$$
p_{(\mathrm{Id}+t\xi(\mu))_{\#}\mu}(x) = p_\mu(x) + t\partial_t p_{(\mathrm{Id}+t\xi(\mu))_{\#}\mu}(x) + o(t) = p_\mu(x) - t\mathrm{div}\big(p_\mu(x)\xi(\mu)(x)\big) + o(t),
\tag{259}
$$

where we used (Villani, 2003, Theorem 5.34) for $\partial_t p_{(\mathrm{Id}+t\xi(\mu))_{\#}\mu} = -t\mathrm{div}\big(p_\mu\xi(\mu)\big)$. Plugging this in (258), we get

$$
\begin{aligned}
\mathbb{F}\big(\exp_\mathbb{P}(t\xi)\big) &= \iint \log\left(\frac{\int p_\nu(x)\mathrm{d}\mathbb{P}(\nu) - t\int \mathrm{div}\big(p_\nu(x)\xi(\nu)(x)\big)\, \mathrm{d}\mathbb{P}(\nu) + o(t)}{p_\pi(x)}\right) p_{(\mathrm{Id}+t\xi(\mu))_{\#}\mu}(x)\, \mathrm{d}\mathbb{P}(\mu)\mathrm{d}x \\
&= \iint \left(\log\left(\int p_\nu(x)\mathrm{d}\mathbb{P}(\nu)\right) - t\frac{\int \mathrm{div}\big(p_\nu(x)\xi(\nu)(x)\big)\, \mathrm{d}\mathbb{P}(\nu)}{\int p_\nu(x)\mathrm{d}\mathbb{P}(\nu)} + o(t)\right. \\
&\qquad \left. - \log p_\pi(x)\right) p_{(\mathrm{Id}+t\xi(\mu))_{\#}\mu}(x)\, \mathrm{d}\mathbb{P}(\mu)\mathrm{d}x \\
&= \iint \left(\log\left(\frac{\int p_\nu(x)\mathrm{d}\mathbb{P}(\nu)}{p_\pi(x)}\right) - t\frac{\int \mathrm{div}\big(p_\nu(x)\xi(\nu)(x)\big)\, \mathrm{d}\mathbb{P}(\nu)}{\int p_\nu(x)\mathrm{d}\mathbb{P}(\nu)} + o(t)\right) p_{(\mathrm{Id}+t\xi(\mu))_{\#}\mu}(x)\, \mathrm{d}\mathbb{P}(\mu)\mathrm{d}x.
\end{aligned}
\tag{260}
$$

Performing the Taylor expansion of the second density, we get

$$
\begin{aligned}
\mathbb{F}\big(\exp_\mathbb{P}(t\xi)\big) &= \iint \left(\log\left(\frac{\int p_\nu(x)\mathrm{d}\mathbb{P}(\nu)}{p_\pi(x)}\right) - t\frac{\int \mathrm{div}\big(p_\nu(x)\xi(\nu)(x)\big)\, \mathrm{d}\mathbb{P}(\nu)}{\int p_\nu(x)\mathrm{d}\mathbb{P}(\nu)} + o(t)\right) \\
&\qquad (p_\mu(x) - t\mathrm{div}\big(p_\mu(x)\xi(\mu)(x)\big) + o(t))\, \mathrm{d}\mathbb{P}(\mu)\mathrm{d}x \\
&= \mathbb{F}(\mathbb{P}) - t\iint \log\left(\frac{\int p_\nu(x)\mathrm{d}\mathbb{P}(\nu)}{p_\pi(x)}\right) \cdot \mathrm{div}\big(p_\mu(x)\xi(\mu)(x)\big)\, \mathrm{d}x\mathrm{d}\mathbb{P}(\mu) \\
&\quad - t\int \frac{\int \mathrm{div}\big(p_\nu(x)\xi(\nu)(x)\big)\, \mathrm{d}\mathbb{P}(\nu)}{\int p_\nu(x)\mathrm{d}\mathbb{P}(\nu)} \int p_\mu(x)\mathrm{d}\mathbb{P}(\mu)\mathrm{d}x + o(t) \\
&= \mathbb{F}(\mathbb{P}) + t\iint \left\langle \nabla\log\left(\frac{\int p_\nu(x)\mathrm{d}\mathbb{P}(\nu)}{p_\pi(x)}\right), \xi(\mu)(x)\right\rangle \, \mathrm{d}\mu(x)\mathrm{d}\mathbb{P}(\mu) + o(t).
\end{aligned}
\tag{261}
$$

We used in the last line the integration by part formula, and $\int \mathrm{div}\big(p_\nu(x)\xi(\nu)(x)\big)\, \mathrm{d}x = 0$. We can conclude that $\nabla_{\mathrm{W}_{\mathrm{W}_2}}\mathbb{F}(\mathbb{P})(\mu) = \nabla V_\mathbb{P}$, where $V_\mathbb{P}(x) = \log\big(\int p_\nu(x)\mathrm{d}\mathbb{P}(\nu)\big) - \log p_\pi(x)$.

**Computation with 1st variation.** We now verify that we would recover the same result by computing the first variation, as conjectured in Appendix B.4.

Let $\pi \in \mathcal{P}_{2,\mathrm{ac}}(\mathbb{R}^d)$, $\pi \propto e^{-V}$ with $V : \mathbb{R}^d \to \mathbb{R}$ a potential. Denote $\mathcal{F}_\pi : \mathcal{P}_{2,\mathrm{ac}}(\mathbb{R}^d) \to \mathbb{R}$, $\mathcal{F}_\pi(\mu) = \mathrm{KL}(\mu||\pi)$ for all $\mu \in \mathcal{P}_{2,\mathrm{ac}}(\mathbb{R}^d)$, and for all $\mathbb{P} \in \mathcal{P}_2\big(\mathcal{P}_{\mathrm{ac}}(\mathbb{R}^d)\big)$,

$$
\mathbb{F}(\mathbb{P}) = \mathrm{KL}\left(\int \mu \, \mathrm{d}\mathbb{P}(\mu) \big|\big| \pi\right) = \mathcal{F}_\pi\left(\int \mu \, \mathrm{d}\mathbb{P}(\mu)\right).
\tag{262}
$$

We will now derive the 1st variation of $\mathbb{F}$. First, recall that $\frac{\delta\mathcal{F}_\pi}{\delta\mu}(\mu) = 1 + \log\mu - \log\pi = 1 + \log\mu + V$. Thus, we have,

noting $\tilde{\mu} = \int \mu \, d\mathbb{P}(\mu)$, $p_{\tilde{\mu}}(x) = \int p_\mu(x) \, d\mathbb{P}(\mu)$ and $\tilde{\chi} = \int \mu \, d\chi(\mu)$,

$$
\begin{aligned}
\frac{d\mathbb{F}}{dt}(\mathbb{P} + t\chi)\big|_{t=0} &= \frac{d\mathcal{F}_\pi}{dt}\left(\int \mu \, d\mathbb{P}(\mu) + t \int \mu \, d\chi(\mu)\right)\Big|_{t=0} \\
&= \frac{d\mathcal{F}_\pi}{dt}(\tilde{\mu} + t\tilde{\chi})\big|_{t=0} \\
&= \int \frac{\delta\mathcal{F}_\pi}{\delta\mu}(\tilde{\mu})(x) \, d\tilde{\chi}(x) \quad \text{by definition of the 1st variation of } \mathcal{F}_\pi \\
&= \int \left(1 + \log p_{\tilde{\mu}}(x) - \log p_\pi(x)\right) d\tilde{\chi}(x) \\
&= \int_{\mathbb{R}^d} \left(1 + \log\left(\int p_\mu(x) \, d\mathbb{P}(\mu)\right) - \log p_\pi(x)\right) \int_{\mathcal{P}_2(\mathbb{R}^d)} p_\mu(x) \, d\chi(\mu) dx \\
&= \int_{\mathcal{P}_2(\mathbb{R}^d)} \int_{\mathbb{R}^d} \left(1 + \log\left(\int p_\nu(x) \, d\mathbb{P}(\nu)\right) - \log p_\pi(x)\right) d\mu(x) \, d\chi(\mu).
\end{aligned}
\tag{263}
$$

Therefore, the first variation of $\mathbb{F}$ at $\mathbb{P}$ is,

$$
\forall \mu \in \mathcal{P}_2(\mathbb{R}^d), \ \frac{\delta\mathbb{F}}{\delta\mathbb{P}}(\mathbb{P})(\mu) = \int_{\mathbb{R}^d} \left(1 + \log\left(\int p_\nu(x) \, d\mathbb{P}(\nu)\right) - \log p_\pi(x)\right) d\mu(x).
\tag{264}
$$

We note that this coincides with the formula of the 1st variation provided in (Lambert et al., 2022, Appendix F) (in the particular case of mixture of Gaussian).

Now, noting $V_\mathbb{P}(x) = 1 + \log\left(\int p_\nu(x) \, d\mathbb{P}(\nu)\right) - \log p_\pi(x)$, the first variation is a potential energy $\frac{\delta\mathbb{F}}{\delta\mathbb{P}}(\mathbb{P})(\mu) = \int V_\mathbb{P} \, d\mu$. Thus, the gradient of $\mathbb{F}$ at $\mathbb{P} \in \mathcal{P}_2(\mathcal{P}_2(\mathbb{R}^d))$ is obtained by the conjecture in Appendix B.4 as, for all $\mu \in \mathcal{P}_2(\mathbb{R}^d)$ $x \in \mathbb{R}^d$,

$$
\nabla_{\mathrm{W}_{\mathrm{W}_2}} \mathbb{F}(\mathbb{P})(\mu)(x) = \nabla_{\mathrm{W}_2} \frac{\delta\mathbb{F}}{\delta\mathbb{P}}(\mathbb{P})(\mu)(x) = \nabla V_\mathbb{P}(x).
\tag{265}
$$

This is well the same formula obtained by computing the Taylor expansion.

If we want to compute the gradient on $\mathcal{P}_2(\mathrm{BW}(\mathbb{R}^d))$, we can take the Bures-Wasserstein gradient of the first variation instead of the Wasserstein gradient. Since it is a potential energy, by (Diao et al., 2023, Lemma 3.1), for any $\mathbb{P} \in \mathcal{P}_2(\mathrm{BW}(\mathbb{R}^d))$ and $\mu \in \mathrm{BW}(\mathbb{R}^d)$, $x \in \mathbb{R}^d$,

$$
\nabla_{\mathrm{W}_{\mathrm{BW}}} \mathbb{F}(\mathbb{P})(\mu)(x) = \nabla_{\mathrm{BW}} \frac{\delta\mathbb{F}}{\delta\mathbb{P}}(\mathbb{P})(\mu)(x) = \int \nabla V_\mathbb{P} \, d\mu + \left(\int \nabla^2 V_\mathbb{P} \, d\mu\right)(x - m_\mu),
\tag{266}
$$

with $m_\mu = \int x \, d\mu(x)$. Since the tangent space of the Bures-Wasserstein space is of the form $T_\mu \mathrm{BW}(\mathbb{R}^d) = \{x \mapsto m + S(x - m_\mu), \ m \in \mathbb{R}^d, S \in S_d(\mathbb{R})\}$ (see *e.g.* (Diao et al., 2023, Appendix A)), then we can identify the mean and covariance part of the gradient as $(\int \nabla V_\mathbb{P} \, d\mu, \int \nabla^2 V_\mathbb{P} \, d\mu)$, which coincides well with the formula derived in (Lambert et al., 2022, Appendix F).

Lambert et al. (2022) experimented with $\mathbb{F}$ in practice by evolving Gaussian particles. Note however that they observed that, even though the KL divergence is (geodesically) convex in $\mathcal{P}_2(\mathbb{R}^d)$ for $V$ convex, $\mathbb{F}$ is not convex in $\mathcal{P}_2(\mathrm{BW}(\mathbb{R}^d))$ as the negative entropy is not.

Also related, Huix et al. (2024) considered optimizing the KL over mixtures of Gaussian, but with fixed covariance observing that the objective can be seen as the KL between a mollified distribution and the target. They minimized it using Wasserstein gradient flows over the means of each mixture. Moreover, their scheme in that case can be seen as a particular case of the one of (Lambert et al., 2022), as described in (Huix et al., 2024, Appendix B).

Also to solve Variational Inference problems, Lim & Johansen (2024) considered the family $q_{\theta,\mu} = \int k_\theta(\cdot|z) d\mu(z)$ with parametric kernels $k_\theta$ satisfying $\int k_\theta(x|z) dx = 1$ for all $z \in \mathbb{R}^{d_z}$. They solved this problem by minimizing the KL divergence with a regularizer, using a gradient flow over $\mathbb{R}^{d_\theta} \times \mathcal{P}_2(\mathbb{R}^{d_z})$. Rønning et al. (2025) considered a mixture family of the form $q(x|\mu_m) = \frac{1}{m} \sum_{\ell=1}^m k(x|z_\ell)$ with $\mu_m = \frac{1}{m} \sum_{\ell=1}^m \delta_{z_\ell}$, and minimized an ELBO using the Stein Variational Gradient Descent.

