# OpenReview forum: "Flowing Datasets with Wasserstein over Wasserstein Gradient Flows"
_ICML.cc/2025/Conference — ICML 2025 oral_

### Official Review · Reviewer_yRk5 · 2025-03-06

**Overall Recommendation:** 5

**Summary:**

This work presents a novel theoretical framework for handling probability distributions over probability distributions. The authors begin by rigorously establishing the Wasserstein over Wasserstein (WoW) distance metric from a functional analysis perspective, providing a mathematical foundation for measuring distances between distributions of distributions. Building upon this metric structure, they develop a comprehensive theory of gradient flows, incorporating maximum mean discrepancy (MMD) with Sliced-Wasserstein based kernels as a tractable objective functional. The theoretical framework is extensively validated through diverse experimental scenarios. These include transformations between synthetic distributions (demonstrated through a three-ring scenario), domain adaptation tasks, dataset distillation, and transfer learning applications. To ensure accessibility and reproducibility, the authors provide detailed mathematical derivations, comprehensive background material, and thorough theoretical analyses. This includes explicit connections to optimal transport theory, probability theory, functional analysis, and variational inference, making the work both theoretically rigorous and practically applicable.

**Claims And Evidence:**

The claims made in this submission are thoroughly supported by both theoretical foundations and empirical evidence. Specifically:

1. Theoretical Rigor:
- The mathematical framework is developed with precise definitions and rigorous derivations
- All theoretical claims are supported by formal proofs.

2. Empirical Validation:
- The experimental results comprehensively validate the theoretical framework across multiple scenarios:
  * Synthetic distributions (three-ring transformation)
  * Domain adaptation tasks
  * Dataset distillation
  * Transfer learning applications
- Each experiment provides quantitative metrics or qualitative visualizations that convincingly demonstrate the method's effectiveness

3. Technical Soundness:
- The implementation details are thoroughly documented
- The experimental protocols are clearly specified and reproducible


In conclusion, the submission maintains a high standard of scientific rigor, with all claims well-supported by both theoretical analysis and experimental results.

**Essential References Not Discussed:**

The paper does not currently discuss related works on semi-implicit variational inference, which are similar to the proposed approach in appendix E.2. For example:
1. Variational Bayes with Stein Mixture Inference
2. Particle Semi-Implicit Variational Inference

**Experimental Designs Or Analyses:**

I have examined the experimental designs and suggest two specific improvements:

1. Ring Transformation Experiments:
- Include comparisons with relevant baselines:
  * KALE flow [1]
  * Kernel KL divergence [2]

---

References:
1. KALE Flow: A Relaxed KL Gradient Flow for Probabilities with Disjoint Support
2. Statistical and Geometrical properties of regularized Kernel Kullback-Leibler divergence

**Methods And Evaluation Criteria:**

Yes, the proposed methods and evaluation criteria are well-designed and appropriate for demonstrating the effectiveness of the proposed approach.

**Other Comments Or Suggestions:**

1. Presentation: Including a visualization of the gradient for the deep learning backend pipeline in Section 4.1 could improve clarity and make the simulation workflow easier to understand.

2. Adding an MP4 or a GIF for three ring case could more effectively showcase the superiority of the proposed approach in the supplementary zip file.

**Other Strengths And Weaknesses:**

## Strengths:
1. The paper is realted to ICML conference.
2. The experimental results are sufficient.

## Weaknesses:
1. The limitations and future research directions are not given explictly.
2. The experiments can be conducted on the recommender system scenario for a wider impact.
3.  This work mainly focus on the MMD functional, the reviewer doubts that how to implement such approach for $f$-divergence, which is of great importance to design sampling algorithm.

**Questions For Authors:**

1. **How does the proposed method handle scenarios where the labels are not categorical, such as in transfer learning between regression tasks**?
2. Why the authors mainly consider the sliced wasserstein distance rather than the vanilla wasserstein distance or sinkhorn distance?
3. The loss curve in the `Rings.ipynb` appears to fluctuate. How can the convergence of the proposed approach be effectively validated?
4. Is it possible to extend the proposed approach to constrained support? For example, Dirichlet distribution on the manifold.

**Relation To Broader Scientific Literature:**

Yes, the key contributions of the paper are foundational and have potential implications for broader applications of the proposed approach.

**Theoretical Claims:**

As an applied mathematics researcher, I have carefully examined most of the theoretical derivations and proofs in the manuscript. While the mathematical framework is generally sound and well-justified (for me), I would like to raise several technical concerns that merit further discussion:

1. Domain Specification of the Functional:
- On page 3, the authors define $\mathcal{F}:\mathcal{P}_2(\mathcal{M})\to\mathbb{R}$
- For most machine learning applications, particularly those involving distances or discrepancy measures, shouldn't the codomain be specifically $\mathbb{R}^+ \cup$ {0}?

2. Treatment of Higher-Order Terms:
- The derivations frequently approximate higher-order terms as $o$ terms
- While this is a common practice in differential geometry, its applicability to $\text{W}_{\text{W}_2}$ warrants further justification
- My understanding is that the finite second moment property of Wasserstein space might justify this approximation, but is this operation still suitable for WoW?

3. Continuity Equations in WoW Framework:
- An interesting theoretical extension would be the formulation of continuity equations within the $\text{W}_{\text{W}_2}$ framework for functional $\mathcal{F}$. Specifically, for $\mathcal{F}[\rho]$ in Wasserstein space, we may define $\frac{\partial \rho}{\partial \tau}=-\nabla\cdot(\rho\nabla\frac{\delta \mathcal{F}[\rho]}{\delta \rho})$. Can we have similar a PDE in WoW ?

---

> ### Author Rebuttal · Authors · 2025-03-28
>
> Thank you for your appraisal and positive comments on our paper. We address your comments below.
>
> **Domain Specification of the Functional**
>
> For most of ML applications where we aim at minimizing distances, we agree that the codomain is $\mathbb{R}_+\cup \\{+\infty\\}$. Nonetheless, the differential structure holds also for $\mathbb{R}$ as codomain, which may cover e.g. potential energies with $V$ negative.
>
> **Higher-Order Terms**
>
> The $o(W_{W_2})$ notation can be defined using the Landau notation. The main difficulty when using these terms in the article is whether we can integrate them when using the Taylor expansion. This can be handled using regularity hypothesis as bounded Hessian.
>
> **Continuity Equations in WoW Framework**
>
> We expect a similar continuity equation $\partial\_t \mathbb{P}\_t = \mathrm{div}(\mathbb{P}\_t \nabla_{W\_{W\_2}}\mathbb{F}(\mathbb{P}\_t))$. In Proposition 3.7, we took a first step by showing that AC curves satisfy a continuity equation. However, to derive the equation of the WoW gradient flow, we would need to show that the minimizing movement scheme converges towards the right equation (see e.g. (Santambrogio, 2017)), and to define an appropriate notion of divergence operator on this space, e.g. the one proposed in (Schiavo, 2020). We leave these questions for future work.
>
> **Comparisons with relevant baselines.**
>
> Thank you for these suggestions. KALE flows and the flows of the KKL divergence are not designed to optimize over $\mathcal{P}(\mathcal{P}(\mathbb{R}^d))$. In Figure 4, we showed an example of the flow of the MMD with Riesz kernel, which is designed on $\mathcal{P}(\mathbb{R}^d)$ as KALE flows and the KKL divergence.
>
> **Related works.**
>
> Thank you for pointing to us these related works which we were not aware of.
>
> **Limitations and future research directions.**
>
> We will add a paragraph in the revised version of the paper indicating future research directions (e.g. using other kernels, f-divergences or continuous labels) and limitations (e.g. theory on compacts manifolds, lack of continuity equation for the flow, dimension of xps).
>
> **This work mainly focus on the MMD functional.**
>
> We focus on the MMD functional as it can be decomposed as a potential and an interaction energy, which we know how to differentiate in the WoW space (see Section 4.2).
>
> Using this approach for f-divergences could be of great interest to do e.g. sampling. We plan to tackle this in future works, but we note that it is not straightforward, as we would need to identify a base measure w.r.t which the measures need to be AC (see e.g. (Schiavo, 2020)), and then compute the WoW gradient of these functionals.
>
> **Adding a GIF for 3 ring case.**
>
> Thank you for this suggestion. We will add https://ibb.co/5qgjhgC.
>
> **1.Labels not categorical?**
>
> Handling continuous labels is an interesting extension of our method. This can be viewed as conditioning with continuous labels, akin to an infinite mixture. More formally, we could define it as $\mathbb{P}=\int \delta_{\mu_y}\mathrm{d}\lambda(y)$ with $\mu_y$ the conditional distribution given the label $y$ and $\lambda$ a distribution over the labels. For a discrete distribution $\lambda$, this would recover the discrete case we use in the paper. In practice,  we would need to discretize $\lambda$.
>
> **2. Why the authors consider the SW distance rather than wasserstein or sinkhorn?**
>
> We consider SW for two reasons:
> 1. It is much faster to compute than the Wasserstein or Sinkhorn distance (complexity of $O(Ln\log n)$ for SW versus $O(n^3\log n)$ for Wasserstein and $O(n^2\log n/\varepsilon^2)$ for Sinkhorn, see e.g. [1]).
> 2. SW allows to define valid positive definite kernels [2], which is not the case for the Wasserstein distance [1].
>
> [1] Peyré, G., & Cuturi, M. Computational optimal transport: With applications to data science. Foundations and Trends in Machine Learning, 2019.
>
> [2] Kolouri, S., Zou, Y., & Rohde, G. K. Sliced Wasserstein kernels for probability distributions. IEEE Conference on Computer Vision and Pattern Recognition. 2016.
>
> **3. The loss curve in the Rings.ipynb appears to fluctuate.**
>
> The loss curve may fluctuate as SW is estimated through a Monte-Carlo approximation. We observed empirically that for a sufficient number of iterations and with well chosen hyperparameters, it converges well and it fluctuates around small values, close to the minimum.
>
> **4. Is it possible to extend the proposed approach to constrained support?**
>
> The proposed flow might be combined with methods to constrain the support such as Mirror Descent, which has been recently extended to Wasserstein gradient flows in [1], or using barrier methods as in [2].
>
> [1] Bonet, C., Uscidda, T., David, A., Aubin-Frankowski, P. C., & Korba, A. Mirror and preconditioned gradient descent in wasserstein space. NeurIPS 2024.
>
> [2] Li, L., Liu, Q., Korba, A., Yurochkin, M., & Solomon, J. Sampling with mollified interaction energy descent. ICLR 2023.

---

> > ### Comment · Reviewer_yRk5 · 2025-04-03
> >
> > Thank you for the detailed and thoughtful response to my questions. I truly appreciate the effort you have invested in improving the submission and the careful consideration given to the feedback. In light of these updates, *I have reassessed the paper and decided to increase my score*, as I now believe this work makes a stronger contribution to the fields of gradient flow and data mining. Additionally, *I hope to see future work or further experiments exploring non-categorical labels*.

---

> > > ### Author Response · Authors · 2025-04-05
> > >
> > > Thank you for raising the score and again for your positive comments!

---

### Official Review · Reviewer_LUYC · 2025-03-11

**Overall Recommendation:** 5

**Summary:**

The paper proposes a framework for optimizing functions over probability measure spaces  of probability measures. The approach is based on Wasserstein over Wasserstein gradient flows. The main contribution is a theoretical definition of this flow. The author also introduces objectives that are tractable within this framework. The approach is validated on synthetic datasets and small vision datasets for distribution flows, domain adaptation and dataset distilaltion.

**Claims And Evidence:**

Claim 1: Flowing datasets can be represented as probability distributions over probability distributions.

Claim 2: A differential structure exists in the space of Wasserstein distances over Wasserstein distance.

These claims are supported by a rigorous mathematical foundation, utilizing optimal transport, Riemannian structures, and geodesics.

Claim 3:
This approach can be applied to distribution flows, domain adaptation, and dataset distillation.

This claim is supported by experimental results (see next section).

**Essential References Not Discussed:**

no.

**Experimental Designs Or Analyses:**

The experimental design and analysis are well-suited to the problem, except for the limitation in problem size.

**Methods And Evaluation Criteria:**

The approach is first illustrated on a synthetic dataset (Three Rings). The qualitative results in this case are convincing.
Then, the approach is validated on several tasks:
- Domain Adaptation: Qualitative results of data flowing from MNIST to other datasets are presented. A classification task is then proposed, where a network is pretrained on a dataset, and accuracy is measured as the data flows from the initial dataset to another one (with aligned classes).
- Dataset Distillation: The WoW gradient flow is used to generate a condensed dataset that maintains high classification accuracy.
- Transfer Learning: The WoW gradient flow is also applied to transfer learning, generating a condensed dataset that preserves high classification accuracy.

For all experiments, the quantitative results support the initial claims and are convincing. However, there are some limitations:
- The qualitative results (Figures) are not entirely convincing and do not clearly suggest a continuous flow from one dataset to another.
- The results are limited to very small datasets and toy problems.

**Other Comments Or Suggestions:**

Typos :
- In this experiment, we generate sample from different datasets starting from Gaussian noise -> In this experiment, we generate **samples** from different datasets starting from Gaussian noise

**Other Strengths And Weaknesses:**

Strengths:
- The paper is very rigorous and mathematically strong.
- Introducing a tractable gradient flow for probability distributions over probability distributions, even if limited to small problems, is an important contribution.

Weakness:
- The experiments are restricted to very small datasets.

**Questions For Authors:**

How can the method be applied to larger-scale problems?

**Relation To Broader Scientific Literature:**

The paper is well-situated within the current literature, and the authors demonstrate a strong understanding of the state of the art in the field.

**Theoretical Claims:**

I reviewed the proofs, but I am not expert enough to verify their correctness.

---

> ### Author Rebuttal · Authors · 2025-03-28
>
> Thank you for your appraisal and positive comments on our paper. We answer your comments below.
>
> **The qualitative results (Figures) are not entirely convincing and do not clearly suggest a continuous flow from one dataset to another.**
>
> We observed empirically that the flow goes very fast from the source dataset towards the neighborhood of the target dataset, but takes more time to converge towards clean images of the target dataset. This might explain why the flows do not appear very continuous, e.g. on Figure 2, as the discretization in time is linear. On Figure 9, we reported a finer discretization, which shows slightly better the behaviour.
>
> We will also add in the zip supplementary materials the following gif showing the evolution of the rings (https://ibb.co/5qgjhgC).
>
>
> **How can the method be applied to larger-scale problems?**
>
>
> As the computational complexity of computing the MMD with a Sliced-Wasserstein kernel is in $O(C^2 Ln(\log n + d))$ for datasets with $C$ classes, and $n$ samples in each class, we believe that we can scale this algorithm to larger datasets with more classes and more samples by class.
>
>
>
> The bottleneck will probably be for higher dimensional datasets. We tried on a (relatively) higher dimensional dataset CIFAR10 (see Figure 10 in Appendix D.4), and observed that the flow scales well. However, it required a lot more optimization steps to converge (150K steps for CIFAR10 against 18K steps for MNIST) with the same number of Monte-Carlo samples.
>
> To scale to higher dimensional datasets, there could be several solutions. On one hand, one can try to get the number of projections $L$ bigger to get a better approximation of the Sliced-Wasserstein distance, or use better approximations using e.g. control variates as in [1]. We could also use faster optimization algorithms to try to accelerate the speed of convergence or other variants of the Sliced-Wasserstein distance more adapted to images, e.g. using convolution as projections as in [2,3]. We leave these investigations for future works.
>
> We also replicated the domain adaptation experiment (Figure 3) by flowing from the dataset SVHN to CIFAR10, which are both dataset of shape 32x32x3, see https://ibb.co/C3B3Ffty. It demonstrates that it still works in moderate higher dimension. We will put it in Figure 3 in place of the KMNIST to MNIST example.
>
>
> [1] Leluc, R., Dieuleveut, A., Portier, F., Segers, J., & Zhuman, A. Sliced-Wasserstein estimation with spherical harmonics as control variates. International Conference on Machine Learning (2024).
>
> [2] Nguyen, Khai, and Nhat Ho. "Revisiting sliced Wasserstein on images: From vectorization to convolution." Advances in Neural Information Processing Systems 35 (2022).
>
> [3] Du, C., Li, T., Pang, T., Yan, S., & Lin, M. "Nonparametric generative modeling with conditional sliced-Wasserstein flows." International Conference on Machine Learning (2023).
>
>
> **Typos**
>
> Thank you, we corrected the typo.

---

> > ### Comment · Reviewer_LUYC · 2025-04-02
> >
> > I thank the authors for their precise answer, which confirm my recommendation.

---

> > > ### Author Response · Authors · 2025-04-05
> > >
> > > Thank you for your response and for your positive comments!

---

### Official Review · Reviewer_asNc · 2025-03-14

**Overall Recommendation:** 4

**Summary:**

This paper introduces a framework for optimizing functionals over probability measures of probability measures by leveraging the Riemannian structure of this space to develop Wasserstein over Wasserstein (WoW) gradient flows. It provides a theoretical foundation for these flows and a practical implementation using Forward Euler discretization. The paper also proposes a functional objective, which is based on Maximum Mean Discrepancy with a Sliced-Wasserstein kernel, enabling computationally efficient gradient flow simulation. The method is applied to dataset modeling, treating datasets as mixtures of probability distributions corresponding to label-conditional distributions.

**Claims And Evidence:**

The claim Riemannian structure of WoW space enables gradient flow dynamics is proved in Appendix and the effectiveness of SW-based MMD for dataset flows is validated empirically.

The claim of tractable implementation lacks runtime analysis or complexity comparisons (e.g., runtime, number of projections, number of measures) with alternatives like Sinkhorn, especially for large-scale datasets when authors claim to be tractable.

**Essential References Not Discussed:**

A related work on sliced Wasserstein kernel can be useful to discuss "Unbiased Sliced Wasserstein Kernels for High-Quality Audio Captioning", Luong et al.

**Experimental Designs Or Analyses:**

No ablation study on the number of SW projections L (fixed to 100).

Comparisons are limited to OTDD and basic baselines. Recent methods like Dataset Condensation with Gradient Matching are absent from Table 1.

The paper criticizes prior work for assuming Gaussian class-conditional distributions but not including an ablation study where the target dataset has non-Gaussian class distributions.

**Methods And Evaluation Criteria:**

The methodology is sound, employing well-established optimal transport techniques and MMD functionals. The evaluation is based on classification accuracy improvements, which are a reasonable measure for transfer learning and dataset distillation tasks.   The use of SW kernels is suitable as SW distances avoid the O(N^2) cost of Wasserstein.  However, additional baselines and comparisons with alternative dataset adaptation methods would strengthen the evaluation.

**Other Comments Or Suggestions:**

No

**Other Strengths And Weaknesses:**

Strengths: The paper introduces a novel and theoretically grounded approach to dataset adaptation, demonstrating strong empirical results.

Weaknesses: Experiments are only on *NIST datasets. Including more datasets on natural images could make the paper stronger e.g., CIFAR10, Imagenet.

**Questions For Authors:**

Can the method scale to larger datasets beyond MNIST variants?

The experiments use fixed hyperparameters (e.g., L=500 projections for SW, \tau=0.1 step size). How sensitive is the method to choices of L, \tau, momentum , and kernel bandwidth? Are there guidelines for tuning these in practice?

The WoW gradient flow involves nested Wasserstein computations. How does the computational cost of your method scale with the number of classes  and samples per class? Could you provide a comparison of efficiency (runtime and memory usage) against methods like OTDD on large-scale datasets?

The paper assumes compactness and connectivity of the manifold for theoretical results. How do these assumptions impact practical applications where data may lie on non-compact spaces?

**Relation To Broader Scientific Literature:**

The paper builds on a rich literature of optimal transport, gradient flows, and dataset adaptation. It extends previous works on dataset dynamics in probability space by introducing a hierarchical structure through the WoW distance. The connection to prior work on Wasserstein gradient flows and MMD functionals is well-discussed.

**Theoretical Claims:**

Proposition 3.7 assumes the base space M is compact, which is restrictive for unbounded domains. The authors do not address how their results extend to non-compact cases.

The continuity equation (Eq. 3.6) relies on the strong assumption that the velocity field vt is Lipschitz.

The class alignment via 1-NN and majority vote seems heuristic.

---

> ### Author Rebuttal · Authors · 2025-03-26
>
> Thank you for reading the paper and for your feedback. We answer your comments below. Please do not hesitate if you have other questions.
>
> **The continuity equation relies on the strong assumption that the velocity field vt is Lipschitz.**
>
> In Proposition 3.7, we show that if $(\mathbb{P}\_t)\_t$ is an absolutely continuous curve, then we have $\|v_t\|_{L^2(\mathbb{P}_t)}\le |\mathbb{P}'|(t)$ and the continuity equation. We do not assume that the velocity field is Lipschitz.
>
> **The class alignment via 1-NN and majority vote seems heuristic.**
>
> Another solution less heuristic to assign the classes is to compute an OT map between $\mathbb{P}=\frac{1}{C}\sum_{c=1}^C \delta_{\mu_c}$ and the target $\mathbb{Q}=\frac{1}{C}\sum_{c=1}^C \delta_{\nu_c}$ with the 2-Wasserstein distance as ground cost. We replicated the results of Figure 3 with this method, and observe that it gives similar results. We will fix this in the revision.
>
>
> **Recent methods like Dataset Condensation with Gradient Matching (DC) are absent from Table 1.**
>
> We chose to compare with methods that only optimize a distance to generate new samples, as these are the are the most comparable approaches, and we do not claim to be state of the art for dataset distillation. Thus, we compared in Table 1 with Distribution Matching (DM), introduced in [1]. Notably, [1] already compares DM with DC (see their Table 1), showing that their performances are comparable.
>
> [1] Zhao, B., & Bilen, H. Dataset condensation with distribution matching. WACV 2023.
>
> **Ablation study where the target dataset has non-Gaussian class distributions.**
>
> As shown in Table 2, our method outperforms in most settings prior works that assume Gaussian class distributions, suggesting that avoiding this assumption leads to better performance on image datasets.
>
> **Related work.**
>
> Thank you for pointing to us this work, which seems relevant.
>
> **Can the method scale to larger datasets beyond MNIST variants?**
>
> Please find the answer in the response of Reviewer LUYC.
>
> **How sensitive is the method to choices of L, tau, momentum , and kernel bandwidth?**
>
> The method is sensitive to the kernel bandwidth when using the Gaussian SW kernel, as we showed in Figure 5 of Appendix D.2. In practice, we use the Riesz SW kernel, which does not require to tune a bandwidth.
>
> As we use a gradient descent method, taking the step size too big will make the scheme diverge.
>
> We use momentum to accelerate the convergence of the scheme, but we notice that it still converges when using no momentum, but slower, see Figure 11 in Appendix D.4.
>
> In low dimension, the method is not very sensitive to the number of projections $L$ (see e.g. the Figure https://ibb.co/SX2gtntd for the ring experiment). In higher dimension such as for MNIST, it is more sensitive, and a relatively big number of projections improves the convergence as it provides a better approximation of the gradient. We will add the following Figure (https://ibb.co/pr2MwMLH) showing results for different values of $L$ for the generation of MNIST samples.
>
> **How does the computational cost of your method scale with the number of classes and samples per class?**
>
> In our experiments, we focused on the minimization of the MMD with a kernel based on the Sliced-Wasserstein distance, which can be computed in $O(Ln\log n)$ between $\mu_n=\frac{1}{n}\sum_{i=1}^n \delta_{x_i}$ and $\nu_n=\frac{1}{n}\sum_{j=1}^n \delta_{y_j}$. For $C$ classes with each class having $n$ samples, the MMD has therefore a total runtime complexity of $O(C^2 Ln(\log n + d))$. Moreover, note that the pairwise Sliced-Wasserstein distances can be computed in parallel.
>
> In contrast, OTDD requires first to compure $C^2$ pairwise Wasserstein distances which has a complexity of $O(C^2 n^3 \log n)$ if they are computed between the empirical distributions, and of $O(C^2 (nd^2 + d^3))$ if they are computed using the Gaussian approximation. Then, it requires to compute a second OT problem between the $nC$ samples with cost $d\big((x,c), (x,c')\big) = \|x-x'\|^2 + W_2^2(\mu_c,\nu_{c'})$, which has a complexity in $O(n^3C^3\log(nC))$ and can be reduced to $O(n^2C^2\log(nC)/\varepsilon)$ using the entropic regularized problem.
>
> We compared the runtime for the transfer learning experiment in https://ibb.co/4R0GtczT.
>
>
> **The paper assumes compactness and connectivity of the manifold for theoretical results.**
>
> We assume compactness and connectivity of the manifold for theoretical purpose, as we notably rely on the seminal results of [1]. Of course, this is not always the case in practical applications, and thus we acknowledge that there is still a gap here to make the theoretical derivations to hold on non-compact spaces. Nonetheless, it is reasonable to assume that the data lie on a compact space.
>
> [1] Schiavo, L. D. A Rademacher-type theorem on L2-Wasserstein spaces over closed Riemannian manifolds. Journal of Functional Analysis, 2020.

---

> > ### Comment · Reviewer_asNc · 2025-04-03
> >
> > I would like to thank the authors for the detailed rebuttal. I raised the score to 4 since my questions are addressed. I suggest the authors to include the discussion with reviewers in the revision.

---

> > > ### Author Response · Authors · 2025-04-05
> > >
> > > Thank you for raising the score. We will add the elements discussed with the reviewers in the revised version.

---

### Decision · Program_Chairs · 2025-05-01

**Decision:**

Accept (oral)

**Comment:**

Through motivating applications like domain transfer and distillation, the novel task of dataset flow, where data is itself a measure is considered i.e., flow involving measures of measures. Firstly, the WOW metric is defined, which is nothing but the Wasserstein whose ground metric is another Wasserstein. It geometry is formally studied. Then, JKO flows under this metric are formally studied. simulations on benchmarks show the efficacy of the methodology.

All the reviewers acknowledge that the problem setting is novel and interesting. The technical contribution is enough for a publication.